# *Notch1* mutations drive clonal expansion in normal esophageal epithelium but impair tumor growth

Emilie Abby[1], Stefan C. Dentro [1,2,9], Michael W. J. Hall [1,3], Joanna C. Fowler[1], Swee Hoe Ong [1], Roshan Sood [1], Albert Herms [1,4], Gabriel Piedrafita [5,6], Irina Abnizova[1], Christian W. Siebel[7], Moritz Gerstung [2,9], Benjamin A. Hall [8] & Philip H. Jones [1,3] ✉

*NOTCH1* mutant clones occupy the majority of normal human esophagus by middle age but are comparatively rare in esophageal cancers, suggesting *NOTCH1* mutations drive clonal expansion but impede carcinogenesis. Here we test this hypothesis. Sequencing *NOTCH1* mutant clones in aging human esophagus reveals frequent biallelic mutations that block *NOTCH1* signaling. In mouse esophagus, heterozygous *Notch1* mutation confers a competitive advantage over wild-type cells, an effect enhanced by loss of the second allele. Widespread *Notch1* loss alters transcription but has minimal effects on the epithelial structure and cell dynamics. In a carcinogenesis model, *Notch1* mutations were less prevalent in tumors than normal epithelium. Deletion of *Notch1* reduced tumor growth, an effect recapitulated by anti-NOTCH1 antibody treatment. *Notch1* null tumors showed reduced proliferation. We conclude that *Notch1* mutations in normal epithelium are beneficial as wild-type *Notch1* favors tumor expansion. NOTCH1 blockade may have therapeutic potential in preventing esophageal squamous cancer.

Aging tissues accumulate somatic mutations[1–4]. Some mutations confer a competitive advantage on progenitor cells, which may form mutant clones that colonize normal tissue. These clonal expansions are often associated with mutations linked to cancer and may represent the first step in malignant transformation[4]. However, the under-representation of *NOTCH1* mutants in esophageal cancer compared with normal aging epithelium suggests *NOTCH1* mutations may inhibit malignant transformation[2,5].

NOTCH1 is a cell surface receptor composed of an extracellular domain (NEC) and a transmembrane and cytoplasmic subunit (NTM), interacting noncovalently through the negative regulatory region

(NRR; Extended data Fig. 1a)[6,7]. The NRR comprises three Lin12-Notch repeats (LNR) and a heterodimerization domain (HD) that inhibits NOTCH1 activation in the absence of ligand[8]. Ligands bind to conserved epidermal growth factor (EGF) repeats in the NEC. This results in proteolytic cleavage events releasing the intracellular domain (NICD), which translocates to the nucleus and alters target gene transcription[8]. In the esophagus, NOTCH1 protein is expressed in proliferating cells and regulates both development and adult tissue maintenance (Extended data Fig. 1a,b)[9].

Different studies have suggested that *NOTCH1* is a tumor suppressor or conversely may promote esophageal carcinogenesis[10–12]. Here we

[1]Wellcome Sanger Institute, Hinxton, UK. [2]European Molecular Biology Laboratory, European Bioinformatics Institute, Cambridge, UK. [3]Department of Oncology, University of Cambridge, Cambridge, UK. [4]Department of Biomedical Sciences, Faculty of Medicine, University of Barcelona, Barcelona, Spain. [5]Department of Biochemistry and Molecular Biology, Complutense University, Madrid, Spain. [6]Epithelial Carcinogenesis Group, Spanish National Cancer Research Centre (CNIO), Madrid, Spain. [7]Department of Discovery Oncology, Genentech, South San Francisco, CA, USA. [8]Department of Medical Physics and Biomedical Engineering, University College London, London, UK. [9]Present address: Artificial Intelligence in Oncology (B450), Deutsches Krebsforschungszentrum, Heidelberg, Germany. ✉e-mail: pj3@sanger.ac.uk

investigate how *NOTCH1* mutants colonize the epithelium, their impact on tissue maintenance and their effect on esophageal carcinogenesis[2,4].

## *NOTCH1* mutant clones in human esophagus

Deep targeted sequencing studies have revealed numerous *NOTCH1* mutants in human esophagus but have not visualized clones and resolved which *NOTCH1* mutation(s) or copy number alterations they carry[2,4]. To achieve this, histological sections of normal epithelium from elderly donors were immunostained for NOTCH1 (Fig. 1a). Positive and negative staining areas were microdissected and targeted sequencing for 322 genes associated with cancer was performed (Fig. 1b). A total of 247 protein-altering somatic variants were identified across 86 samples from six donors aged 43–78. The predominant mutant genes were *NOTCH1*, *TP53* and *NOTCH2* (refs. [2,4]; Supplementary Tables 1–3 and Supplementary Note). Near clonal *NOTCH1* mutations with an average variant allele frequency (VAF) of 0.36 were detected in 81% (70/86) of samples (Fig. 1c,d). Ninety-three percent (25/27) of negative staining areas carried nonsense, essential splice mutations or indels in *NOTCH1* with copy neutral loss of heterozygosity (CNLOH) of the *NOTCH1* locus (human GRCh37−chr9:139,388,896−139,440,238) or a further mutation, likely to disrupt the second *NOTCH1* allele (Fig. 1d,e). Fifty-nine percent (35/59) of positively stained samples carried a missense *NOTCH1* mutation and most of these had either CNLOH or a second mutation (Fig. 1d–f, Extended data Fig. 1c,d and Supplementary Table 4). Overall, most samples (73%, 51/70) had likely biallelic *NOTCH1* alterations (Fig. 1f). To test if the mutations disrupted NOTCH1 function, we stained consecutive sections from additional donors for NOTCH1 protein and NICD1, which is detectable in the nucleus during active signaling (Fig. 1g,h, Extended data Fig. 1a and Supplementary Table 5)[13]. The proportion of epithelium with active NOTCH1 decreased with age (Kendall's tau-b = −0.67, P = 0.014). In older donors, in whom *NOTCH1* mutations are common, NOTCH1− areas were associated with NICD1 loss. We also found occasional NOTCH1+ NICD1− areas, consistent with the presence of missense mutant proteins that reach the cell membrane but lack signaling activity (Fig. 1g,h). NICD1+ and NICD1− areas were histologically undistinguishable, with no significant differences in tissue thickness, cell density or the expression of the proliferation marker Ki67 (Extended data Fig. 1e–h). We conclude that many *NOTCH1* mutant clones in aging human esophagus carry biallelic alterations that disrupt signaling.

## *Notch1* mutations increase clonal fitness

To investigate how *NOTCH1* mutant clones colonize normal epithelium, we tracked the fate of *Notch1* mutant clones in transgenic mice using lineage tracing. Mouse esophageal epithelium consists of layers of keratinocytes. Proliferation is restricted to progenitor cells in the basal layer (Extended data Fig. 2a). Differentiating cells cease dividing, leave the basal layer and migrate toward the epithelial surface where they are shed. Progenitor division is linked to the exit of a nearby differentiating

cell from the basal layer, ensuring basal cell density is kept constant[14]. Dividing progenitors generate either two progenitor daughters, two differentiating daughters or one cell of each type. In wild-type tissue, the probabilities of each progenitor outcome are balanced, generating equal proportions of progenitor and differentiated cells, maintaining cellular homeostasis (Extended data Fig. 2a)[15,16]. Mutations that alter progenitor fate leading to excessive production of progenitors drive mutant clone growth[17,18].

For lineage tracing, we generated *AhCre^ERT Rosa26^floxedYFP Notch1^flox* triple transgenic (YFPCreNotch1) mice. These animals carry a conditional *Notch1* allele and a genetic labeling system. An inducible *Cre* recombinase (*AhCre^ERT*) was used to delete one or both conditional *Notch1* alleles in *Notch1^wt/flox* or *Notch1^flox/flox* animals and induce a separate conditional yellow fluorescent protein (YFP) reporter allele (*Rosa-26^floxedYFP*)[15,19]. YFP was expressed in recombined epithelial cells and their progeny (Extended data Fig. 2b,c). This model was induced at low dose to recombine scattered single basal cells (clonal induction) or at a higher level to recombine a large proportion of basal cells (high induction) (Extended data Fig. 2c,d).

Excision of the *Notch1* allele and expression of the YFP reporter at the *Rosa26* locus can occur in combination or separately, resulting in *Notch1* mutant or wild-type cells expressing YFP or not (Extended data Fig. 2c,d). We confirmed the recombination status of exon 1 of *Notch1* of wild type and fully recombined *Notch1^+/−* and *Notch1^−/−* esophageal epithelium. *Notch1* mRNA and protein expression was halved in *Notch1^+/−* and abolished in *Notch1^−/−* cells compared with wild-type keratinocytes (Extended data Fig. 3a–h and Supplementary Table 6). We then performed genetic lineage tracing by inducing recombination in scattered single progenitors in *YFPCreNotch1^+/+*, *YFPCreNotch1^+/flox* or *YFPCreNotch1^flox/flox* mice. YFP-expressing clones were detected by imaging sheets of epithelium stained for YFP and NOTCH1 (Fig. 2a). YFP+ *Notch1^+/−* or YFP+ *Notch1^−/−* clones were identified from reduced intensity or absence of NOTCH1 immunostaining, respectively, a method validated by detecting *Notch1* recombination in microdissected clones (Fig. 2b, Extended data Fig. 3i–n and Supplementary Note).

The number and location of cells in YFP-expressing clones of each genotype were determined by 3D confocal imaging. The size of YFP+ *Notch1^+/−* clones was substantially increased compared to wild-type YFP+ *Notch1^+/+* clones at all time points. YFP+ *Notch1^−/−* clones were larger still (Fig. 2b–d, Extended data Fig. 3i,j and Supplementary Table 7). To examine the cellular mechanisms underlying mutant clonal expansion, we used short-term cell tracking by labeling cycling cells with the S phase probe 5-ethynyl-2′-deoxyuridine (EdU).

We first counted the proportion of basal cells positive for EdU at 1 h after labeling, which measures the fraction of cells in S phase (Fig. 2e,f). This value was similar for cells within *Notch1^+/−* clones and wild-type cells distant from clones (Fig. 2g and Supplementary Table 8). Within *Notch1^−/−* mutant clones, the proportion of EdU+ basal cells was

**Fig. 1 | *NOTCH1* mutant clones in human esophageal epithelium. a,** Cyrosection of human esophagus. NOTCH1 (green) stains basal and lower suprabasal layer cells, expression is lost in regions of the esophagus. F-actin, magenta; Pa, papillae. Dotted line indicates epithelial submucosal boundary. Image representative of three donors. Scale bar, 100 μm. **b,** Protocol for **c**–**f**. Cryosections were stained for NOTCH1. Contiguous NOTCH1+ and NOTCH1− staining areas were microdissected and sequenced. **c,** Representative images from **b** for donor PD40290. NOTCH1 is red, DNA is blue. Upper labels show sample identification (Id) and NOTCH1 staining status (positive, + or negative, −) for each sample. Lower labels show nonsynonymous *NOTCH1* mutations and VAF and indicate CNLOH if detected. Only mutations with VAF > 0.1 are displayed. Mutation effects are color coded (indel_splicing, gray; missense, blue; nonsense, red). Dashed lines delineate the epithelium and submucosa (white) and borders of sequenced samples (yellow). Solid lines separate the two images of the adjacent regions. Scale bars, 250 μm. **d,** Results from **b**, showing NOTCH1 staining, donor identification, *NOTCH1* mutation calling, CNLOH affecting

*NOTCH1* locus and number of *NOTCH1* mutations per sample (*n* = 86 samples from six donors aged 43–78 years). **e,** Proportion of missense, nonsense, indel/splicing or intronic/silent *NOTCH1* mutations in NOTCH1+ and NOTCH1− samples. Number of *NOTCH1* mutations for each group is shown in brackets. **f,** Proportion of *NOTCH1* mutant samples carrying monoallelic or biallelic *NOTCH1* alterations in each donor. 'Biallelic with second mutation' category includes samples without CNLOH, carrying at least two mutations with VAF ≥ 0.15. Numbers in brackets are total number of *NOTCH1* mutated samples per donor. **g,** NOTCH1 (green, upper panel) and NICD1 (red, lower panel) staining in successive sections of epithelium from an aged donor. ITGA6 (magenta) marks the basal cells. DNA is blue. Inset shows basal and lower suprabasal cells (white rectangles). Dashed lines delineate staining pattern. Images representative of six middle-aged and elderly donors. Scale bar, 100 μm. **h,** Proportion of tissue positive or negative for NOTCH1 and NICD1 in donors aged 20–78 years (total section length 4774–17988 μm per donor, *n* = 9 donors). Id, identification. See Supplementary Tables 1–5.

marginally lower than in wild-type cells (Fig. 2h). We conclude neither *Notch1*[+/−] nor *Notch1*[−/−] clonal expansion results from an increase in mutant cell division rate compared with wild-type cells.

A 48 h EdU experiment labeled S phase cells and tracked the fate of the two cells generated by the subsequent mitosis over the following 48 h. The pair of labeled cells may remain in the basal layer, or

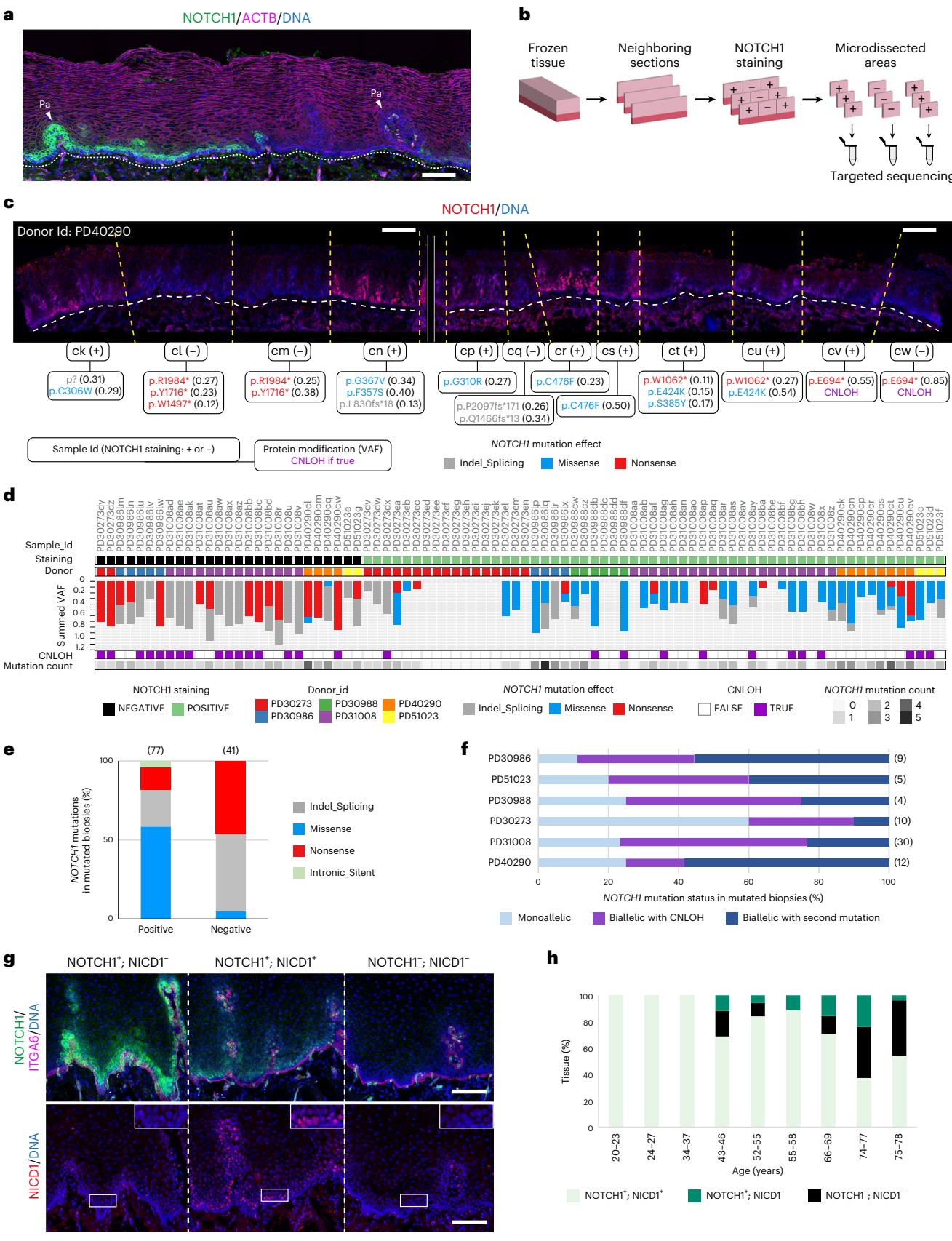

one or both may differentiate and exit the basal layer (Fig. 2i,j). The ratio of EdU-labeled suprabasal cells to the total EdU-labeled cells reflects the rate of production of differentiating cells in the basal layer and their stratification into the suprabasal layers. In *Notch1*[+/−] and *Notch1*[−/−] clones, this ratio is decreased, consistent with a tilt in mutant progenitor cell fate, so that more progenitors and fewer differentiating daughters are produced per average cell division (Fig. 2k). Strikingly, adjacent to *Notch1*[−/−] clones, there was an increase in the suprabasal EdU[+]:total EdU[+] cell ratio in the wild-type cells at the clone margin compared with wild-type cells further from the mutant clone (Fig. 2j,l). This, along with a small decrease in the proportion of wild-type S phase cells at the clone edge, indicates that wild-type cells adjacent to the clone exit the cell cycle, differentiate and exit the basal layer at an increased rate, a phenomenon also reported in previous studies of Notch inhibited keratinocytes interacting with wild type cells (Fig. 2h)[18,20].

These observations explain the increased fitness of *Notch1*[−/−] over *Notch1*[+/−] clones. Cell density was similar in both mutant genotypes and wild-type areas, suggesting that the linkage between cell division and the exit of a nearby differentiating cell from the basal layer is maintained (Fig. 2m,n). Within this constraint, the driving of wild-type cell differentiation and stratification permits *Notch1*[−/−] cell division at the clone edge, accelerating clonal expansion (Fig. 2o).

These observations were integrated into a Wright–Fisher style quantitative model in which fit mutant clones expand until they collide with other mutant clones of similar fitness, at which point they revert to neutral competition[21]. We fitted this model to the clone size data. The inferred fitness for *Notch1*[+/−] clones was higher than that of wild-type cells and the inferred fitness of *Notch1*[−/−] clones markedly greater than that of heterozygous clones (Extended data Fig. 4a–d, Video 1 and Supplementary Note).

## *Notch1* haploinsufficiency enables epithelial colonization

Clones generated by the transgenic deletion of *Notch1* alleles may not reflect the behavior of *Notch1* mutants that appear during aging. We therefore investigated spontaneous *Notch1* mutant clones in control *YFPCreNotch1*[+/+] mice, and the heterozygous epithelium of highly induced *YFPCreNotch1*[+/flox] animals. Both strains were aged before immunostaining the epithelium for NOTCH1 (Fig. 3a). The area of epithelium stained negative for NOTCH1 increased progressively to 12% of *Notch1*[+/+] and 78% of *Notch1*[+/−] epithelium by 65 weeks (Fig. 3b,c and Supplementary Table 9). Widespread loss of NICD1 staining was seen in aged *Notch1*[+/−] tissue (Extended data Fig. 5a,b). These observations

suggest that, as in humans, *Notch1* mutants colonize the aging mouse esophagus and that selection is enhanced in *Notch1*[+/−] epithelium.

To localize potential clones, we stained for NOTCH1 and the YFP reporter. Aging *Notch1*[+/−] epithelium contained multiple ovoid areas of homogenous NOTCH1 staining, positive or negative for YFP but far larger than most YFP labeled clones (Fig. 3d,e). These were suggestive of clonal expansion. A total of 246 such 'expanded' areas along with typical 'nonexpanded' regions were dissected and underwent targeted sequencing for 73 Notch pathway and cancer-related genes (Supplementary Tables 1, 10 and 11). We analyzed for CNLOH and mutations with VAF ≥ 0.2, as below this threshold mutations were considered unlikely to drive clonal expansion. Nintey-seven percent (180/185) of the 'expanded' areas had either *Notch1* protein-altering mutations with VAF ≥ 0.2 or CNLOH involving the *Notch1* locus (GRCm38−chr2:26,457,903-26,503,822). In contrast, only 2 of 61 nonexpanded areas carried *Notch1* mutations and none had *Notch1* CNLOH (Fig. 3f, Extended data Fig. 5c,d and Supplementary Table 10). Only a few mutations in other genes were found, some may have been passengers within a *Notch1* mutant clone. Ninety-four percent (169/180) of expanded areas with *Notch1* altering events carried only a single event (about 50% one *Notch1* protein-altering mutation and the remainder CNLOH) with an average VAF 0.44, consistent with them being clones carrying spontaneous changes affecting the nonrecombined *Notch1* allele (Fig. 3e–g, Supplementary Tables 10, 11 and Extended data Fig. 5c,d). Among clones carrying a *Notch1* mutation, 85% of those stained positive for NOTCH1[+] harbored missense mutations while NOTCH1 negatively stained clones carried mainly indel/splicing (51%) or nonsense mutations (46%) (Fig. 3g). Overall, these results were consistent with findings in aging human esophagus (Fig. 1).

To test the impact of missense *Notch1* mutations, we used an ex vivo functional assay (Extended data Fig. 5e–j and Supplementary Table 10)[22]. *Notch1*[+/−] tissues in Fig. 3e–g were incubated with ethylenediaminetetraacetic acid (EDTA) at 37 °C before fixation. This promotes NOTCH1 cleavage and nuclear migration of NICD without ligand binding (Extended data Fig. 1a)[22]. Some NOTCH1[+] clones displayed nuclear staining, but others did not (Extended data Fig. 5e). Nuclear staining clones were enriched in missense mutations in the ligand binding site, EGF repeats 8–12, whereas non-nuclear staining clones were enriched mutations in the LNR repeats of the NRR domain (Extended data Fig. 5g,h, *P* = 0.001, Chi-square test). Most of ligand binding domain mutations had highly destabilizing properties, consistent with disrupting ligand binding, a process bypassed in the EDTA assay (Extended data Fig. 5i)[23,24]. The NRR domain mutants were clustered in the LNR1 and

---

**Fig. 2 | Lineage tracing of *Notch1* mutant clones. a**, Protocol. *YFPCreNotch1*[+/+], *YFPCreNotch1*[+/flox] and *YFPCreNotch1*[flox/flox] mice were induced at clonal density. YFP[+] *Notch1* wild type (+/+) and YFP[+] *Notch1* mutant clones (+/− or −/−) were imaged at several time points. **b**, *xy* plane basal layer view at 4 weeks p.i. of wild type, *Notch1*[+/−] and *Notch1*[−/−] clones stained for NOTCH1, magenta, YFP, green and DNA, blue. White dashed lines delineate mutant clones. Scale bars: 30 μm. **c,d**, Basal (**c**) and suprabasal (**d**) cells per clone following induction of *Notch1*[+/−] (left panel) or *Notch1*[−/−] (right panel) compared to *Notch1*[+/+] clones. Lines show median and quartiles. *n* mice (clones) for +/+ at 10 d, 2 weeks, 4 weeks, 9 weeks and 13 weeks, respectively: 3 (206)/3 (155)/3 (143)/3 (132)/3 (126). *n* mice (clones) for +/− at 10 d, 4 weeks, 9 weeks and 13 weeks, respectively: 5 (84)/4 (97)/4 (68)/7 (107). *n* mice (clones) for −/− at 10 d, 2 weeks, 4 weeks, respectively: 6 (68)/3 (69)/9 (63). Two-tailed Mann–Whitney test of mutant against +/+ at each time point. **e**, Protocol. *YFPCreNotch1*[+/flox] and *flox/flox* mice were clonally induced, and S phase cells labeled with EdU, 1 h precollection (red). **f**, EdU[+] cells were counted inside clones (green), in wild-type cells adjacent to clones (orange) or distant from clones (beige). **g,h**, Ratio of EdU[+]:total basal cells in YFP[+] *Notch1*[+/−] (**g**) or *Notch1*[−/−] (**h**) mutant clones (YFP[+]; +/− or −/−), in wild type cells at clone edges (edge +/+) or distant from clones (distant +/+). (Mean ± s.e.m., each dot represents a mouse; **g**, *n* = 4830; 1584; 4607 cells in distant +/+; edge +/+; YFP[+] +/− clones from four mice; **h**, *n* = 3967; 1036; 4279 cells in distant +/+; edge +/+; YFP[+] −/− clones from four mice). One-way RM ANOVA; adjusted *P* values from Tukey's multiple

comparisons test against distant[+/+]. **i**, Protocol. Mice were clonally induced and EdU injected 48 h before collection. Labeled cells, red, reveal division outcomes. **j**, *Z* plane (side) views of projected confocal z stacks of YFP[+] *Notch1*[+/−] clone 13 weeks p.i. (left), and YFP[+] *Notch1*[−/−] clone 4 weeks (p.i. right) from (**i**). NOTCH1 (magenta); YFP (green); EdU (gray); DNA (blue). Yellow dashed lines show clone edges. Orange arrow shows differentiating cell adjacent to clone. Images representative of clones in 3 YFPCre*Notch1*[+/flox] and 5 *YFPCreNotch1*[flox/flox] mice. Scale bars: 30 μm. **k,l**, Protocol as in **i**. EdU[+] suprabasal/total EdU[+] cells in YFP[+] *Notch1*[+/−] (**k**), YFP[+] *Notch1*[−/−] (**l**) mutant clones (YFP[+]; +/− or −/−), in wild type cells at clone edges (edge +/+) or distant from (distant +/+) clones. (Mean ± s.e.m., each dot represents a mouse; **k**, *n* = 471; 300; 525 EdU[+] cells in distant +/+; edge +/+; YFP[+] +/− clones from three mice; **l**, *n* = 1304; 723; 1318 EdU[+] cells in distant +/+; edge +/+; YFP[+] −/− clones from five mice). One-way RM ANOVA; adjusted *P* values, Tukey's multiple comparisons test against distant[+/+]. **m,n**, Basal cell density in mutant clones (+/− in **m**, −/− in **n**) and in respective distant wild-type areas (distant +/+). (Mean ± s.e.m., each dot represents a mouse. *n* = 3 mice in **m**, *n* = 6 mice in **n**). Two-tailed paired Student's *t*-tests. **o**, Mechanism of *Notch1* mutant clone expansion. Mutant cell divisions produce more progenitors than differentiating cells on average. Neighboring wild-type cells stratify at the edge of *Notch1*[−/−] mutant clones, allowing accelerated mutant clone expansion. P.i., postinduction. Nb, number. RM, repeated measures; w, weeks. See Supplementary Tables 7 and 8.

LNR2 domains (Extended data Fig. 5j)[25]. In contrast, *NOTCH1* activating mutations in human T cell acute lymphoblastic leukemia (T-ALL) (https://cancer.sanger.ac.uk/cosmic) cluster in the HD domain of the NRR and promote NEC cleavage without ligand interaction

(two-sided Fisher exact test comparing mutation counts in the LNR1-2 and LNR3-HD subregions of the NRR, $P = 1.48 \times 10^{-10}$, Extended data Fig. 5j)[26,27]. These observations suggest that esophageal NRR domain mutations may prevent the cleavage of NOTCH1. We conclude that in

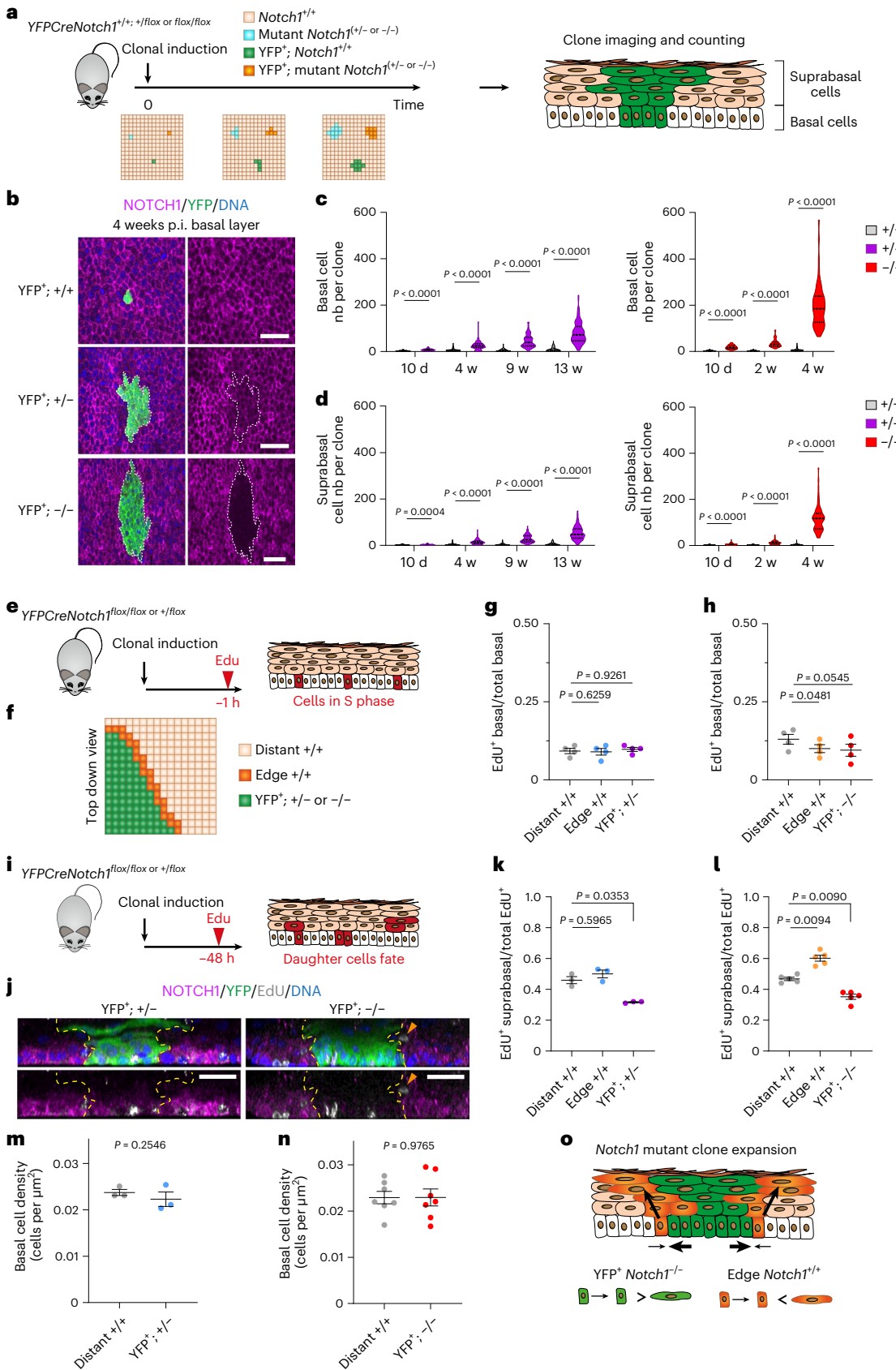

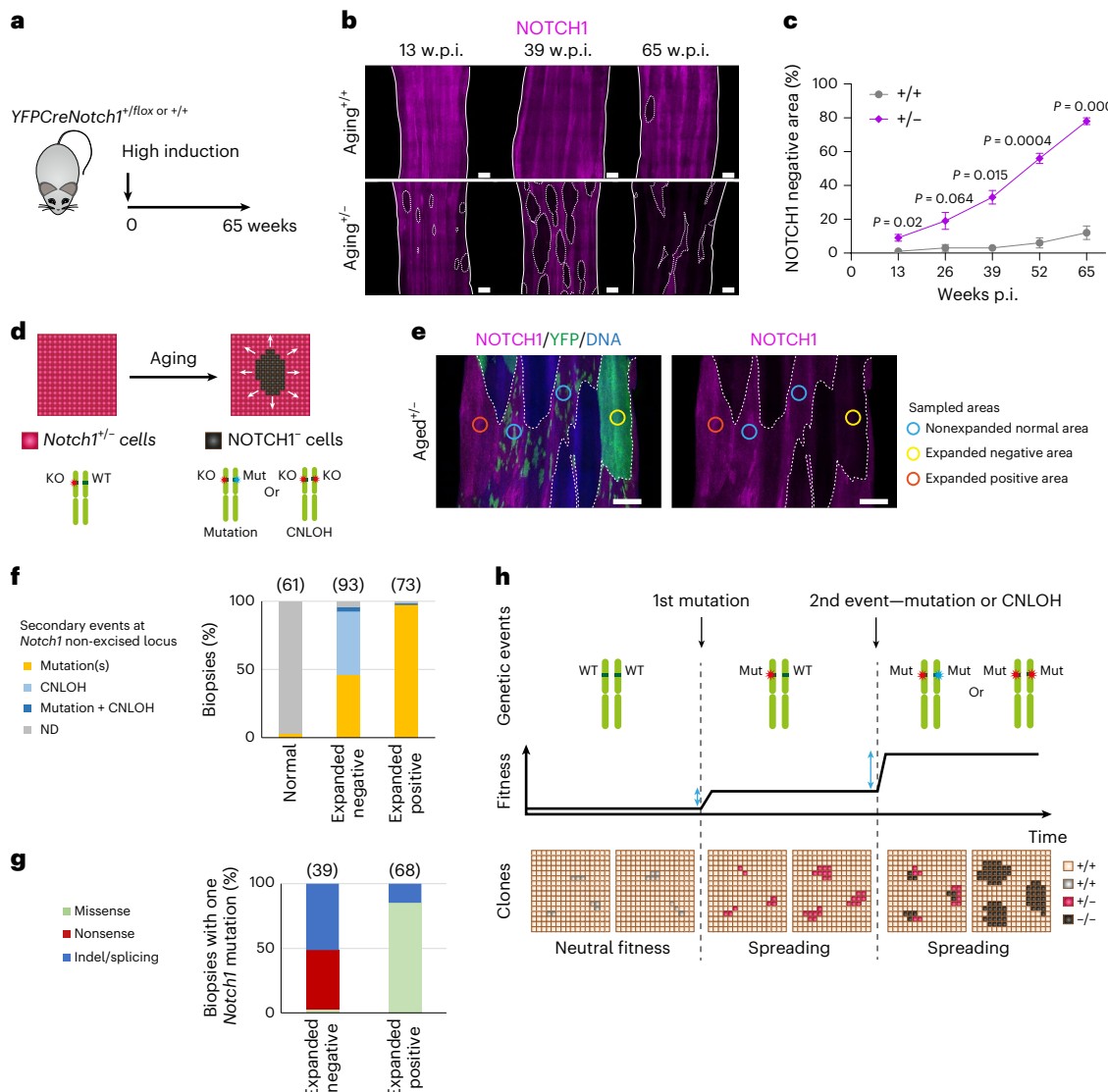

**Fig. 3 | *Notch1* mutants colonize aging esophageal epithelium.**
**a**, *YFPCreNotch1*[+/flox] and *YFPCreNotch1*[+/+] mice were induced at a high level and aged for 65 weeks. **b**, Representative NOTCH1 staining in esophageal epithelium of aging *YFPCreNotch1*[+/+] and *YFPCreNotch1*[+/flox] mice at the indicated time points. White dashed lines delineate negative areas and solid lines delineate tissue edges. Images representative of three mice per time point. Scale bars: 500 μm **c**, Percentage of NOTCH1[−] area increases with age in *Notch1*[+/+] (Kendall's tau-b correlation = 0.56, *P* = 0.0062) and *Notch1*[+/−] (Kendall's tau-b correlation = 0.91, *P* = 8.3 × 10[−6]) esophagi (Mean ± s.e.m., *n* = 3 mice per time point). *P* values shown are from two-sided Welch's *t* test. **d**, Schematic of *Notch1*[+/−] cells (purple cells) showing the spontaneous appearance of expanding NOTCH1[−] cells (black) with aging, possibly caused by genetic events affecting the *Notch1* locus. **e**, Highly induced *YFPCreNotch1*[+/flox] mice were aged 54–78 weeks old, when esophageal epithelium was collected and stained for NOTCH1 (magenta), YFP (green) and DNA (blue). Expanding areas devoid or fully stained with YFP appeared

distinct from normal-appearing areas marked with a patchwork of small YFP[+] clones. Expanded NOTCH1[−] (yellow) and NOTCH1[+] (orange) areas and normal-appearing areas (blue) were isolated for targeted sequencing (*n* = 246 biopsies from ten mice). Colored circles show the sampled areas. White dashed lines delineate negative areas. Scale bars: 500 μm. **f**, Proportion of normal appearing, expanded NOTCH1[−] and expanded NOTCH1[+] biopsies with *Notch1* mutations or CNLOH. **g**, Proportion of NOTCH1[−] and NOTCH1[+] areas carrying a secondary missense, nonsense or indel/splicing *Notch1* mutation. For **f** and **g**, *n* samples are shown in brackets, redundant samples, defined as biopsies sharing the same mutation and separated by <1 mm were counted once (*n* = 227 unique biopsies in total). **h**, Model of colonization by *Notch1* clones. Clonal fitness increases from monoallelic and biallelic *Notch1* mutation resulting in a selective pressure (blue arrows) for biallelic gene alterations. p.i., postinduction, w.p.i., weeks postinduction. WT, wild type. KO, knock-out allele lacking *Notch1* exon 1. Mut, mutation. ND, none detected. See Supplementary Tables 9–11.

heterozygous epithelium, most spontaneous mutants disrupt NOTCH1 function, conferring a fitness advantage over neighboring cells.

Collectively these observations reveal that haploinsufficiency is key for the normal esophagus to be colonized so effectively by *Notch1* mutants. Neutral mutants do not colonize the tissue[15,28]. Loss of one allele biases mutant progenitor cell fate toward the production of progenitors, increasing the likelihood that mutant clones will expand and persist in the epithelium (Extended data Fig. 4e,f and Video 2). *Notch1* inactivated cells have a further increased fitness so that subclonal

loss of the second allele within a persisting heterozygous clone will generate cells that outcompete both *Notch1*[+/+] and *Notch1*[+/−] neighbors (Fig. 3h). This model explains the high prevalence of clones with *NOTCH1* mutation and CNLOH in aging human esophagus.

## *Notch1*[−/−] epithelium has minimal phenotype

Epithelium lacking functional NOTCH1 might be expected to have a cellular phenotype. To explore the effects of *Notch1* loss in the mouse esophagus, we first performed bulk RNA sequencing (RNA-seq) on

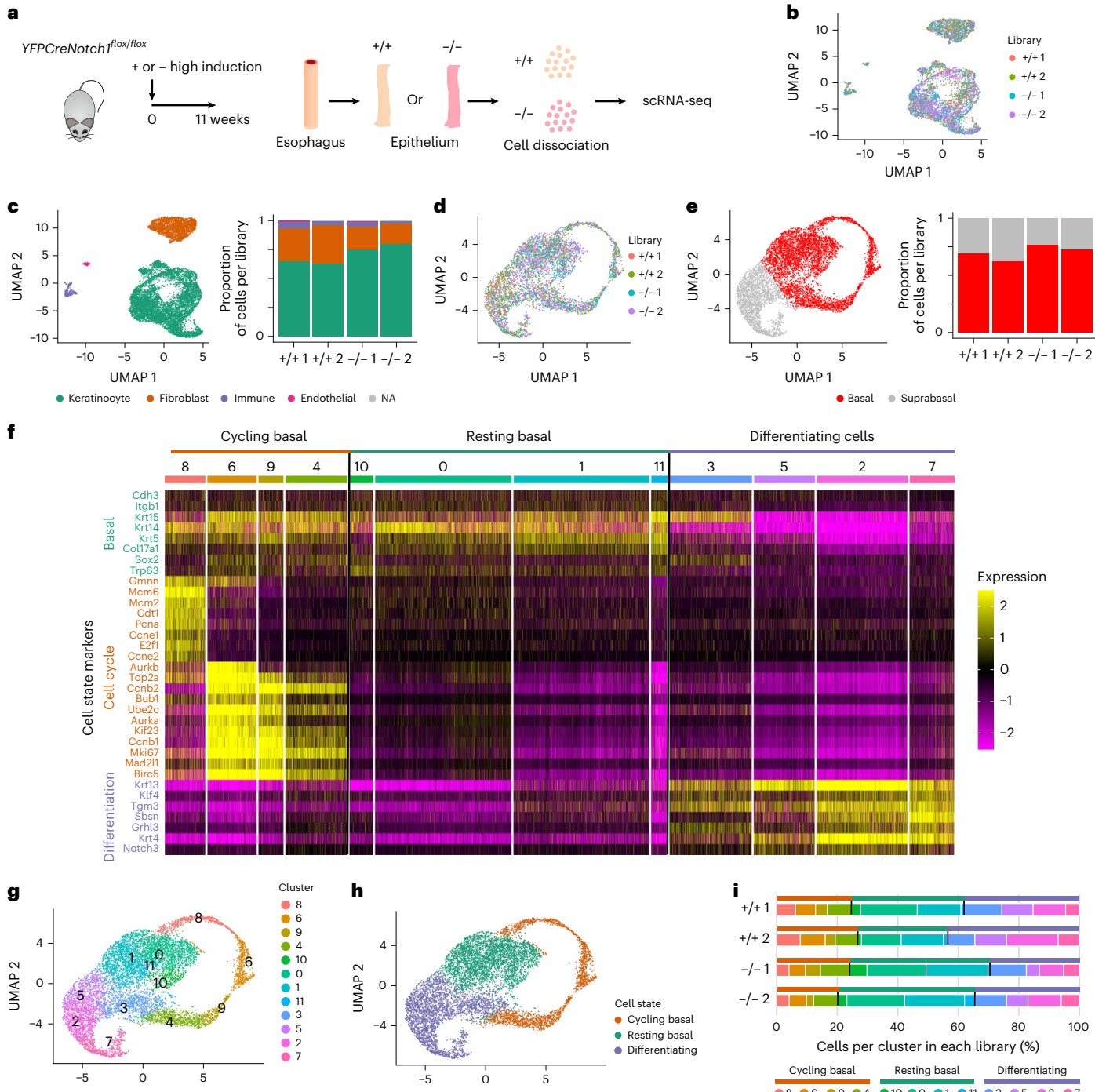

**Fig. 4 | *Notch1* loss does not alter tissue composition or cell dynamics.**
**a**, *YFPCreNotch1^flox/flox* mice were highly induced and aged for 11 weeks, allowing the mutant cells to completely occupy the esophageal epithelium. Controls were uninduced *YFPCreNotch1^flox/flox* mice (+/+). Esophageal epithelium was dissociated and sequenced. **b**, UMAP plot shows an overlay of 1,500 cells from each library ($n$ = 2 mice per genotype; +/+1, $n$ = 2,454; +/+2, $n$ = 3,194; −/−1, $n$ = 1,929; −/−2, $n$ = 5,534). **c**, Left, UMAP plot showing cell types identified via scRNA-seq. Right, stacked bar chart shows the proportion of cell types per library. NA, not available. **d**, UMAP plot shows an overlay of 1,400 cells annotated as keratinocytes from each library (+/+1, $n$ = 1,555; +/+2, $n$ = 1,932; −/−1, $n$ = 1,403; −/−2, $n$ = 3,919). Milo test shows no significant difference in local cell density through UMAP space (Supplementary Note). **e**, Left, UMAP plot of keratinocytes. Right, stacked bar

chart shows the estimated proportion of keratinocytes per library belonging to the basal or suprabasal layers (Supplementary Note). **f**, Heat map showing Seurat processed expression values in the keratinocyte population for representative marker genes of basal cells, cell cycle, and differentiation for the 11 clusters shown in **g** (marker list from ref. [32]). Clusters are grouped in three different cell states: cycling basal, resting basal and differentiating cells. **g**, UMAP plot of keratinocytes representing cell clusters based on Seurat analysis pipeline via the Leiden algorithm. **h**, UMAP plot of keratinocytes showing cycling basal (orange), resting basal (green) and differentiating (purple) cell states based on clusters and differentiation markers analysis performed in **f** and **g**. **i**, Stacked bar charts show the proportion of keratinocytes per cell state (upper bar) and per cluster (lower bar) in each library. See Supplementary Table 16.

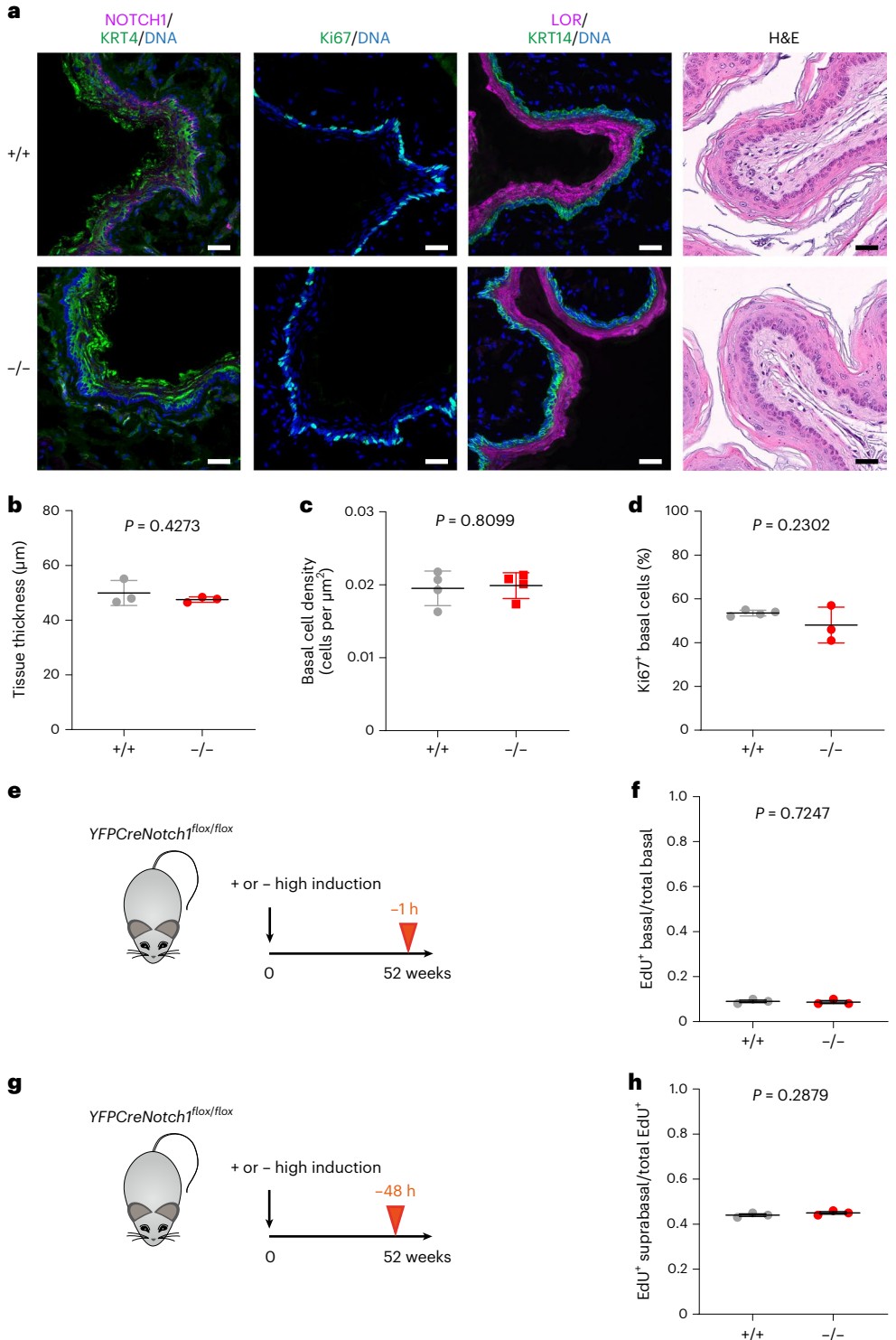

**Fig. 5 | Differentiation and homeostasis in aged *Notch1* mutant mouse tissue.**
**a**, *YFPCreNotch1flox/flox* mice were induced at high dose so mutant cells rapidly covered the esophageal epithelium (−/−). Uninduced *YFPCreNotch1flox/flox* mice were used as wild-type controls (+/+). Mice were aged as in **e** and **g** and tissue was collected. After sectioning, tissue was stained for basal cell marker KRT14, NOTCH1, proliferation marker Ki67, differentiation markers KRT4 and LOR and with H&E. Images are representative of three mice of each genotype. Scale bars, 30 μm. **b**, Thickness of the epithelium was measured on H&E scanned sections (mean ± s.e.m., each dot represents a mouse, *n* = 3 mice). Two-tailed unpaired Student's *t* test. **c**, Epithelium basal cell density was measured on whole-mount tissue. (Mean ± s.e.m., each dot represents a mouse, +/+, *n* = 4097 cells from four mice; −/−, *n* = 3964 cells from four mice). Two-tailed unpaired Student's *t* test. **d**, Proportion of proliferative basal cells was measured on sections stained for

Ki67, KRT14 and DAPI. (Mean ± s.e.m., each dot represents a mouse, +/+, *n* = 1548 cells from four mice; −/−, *n* = 1129 cells from three mice). Two-tailed unpaired Student's *t* test. **e,f**, Highly induced or uninduced control *YFPCreNotch1flox/flox* mice were aged for 52 weeks and injected with EdU 1 h before collection (**e**). Ratio of EdU+ basal cells on total number of basal cells was calculated (**f**) (mean ± s.e.m., each dot represents a mouse, +/+, *n* = 2754 cells from three mice; −/−, *n* = 2565 cells from three mice). Two-tailed, unpaired Student's *t* test. **g,h**, Highly induced or uninduced control *YFPCreNotch1flox/flox* mice were aged for 52 weeks and injected with EdU 48 h before collection (**g**). Ratio of EdU+ suprabasal cells. (Mean ± s.e.m., each dot represents a mouse; +/+, *n* = 2687 EdU+ cells from three mice; −/−, *n* = 2201 EdU+ cells from three mice). Two-tailed unpaired Student's *t* test. See Supplementary Table 18.

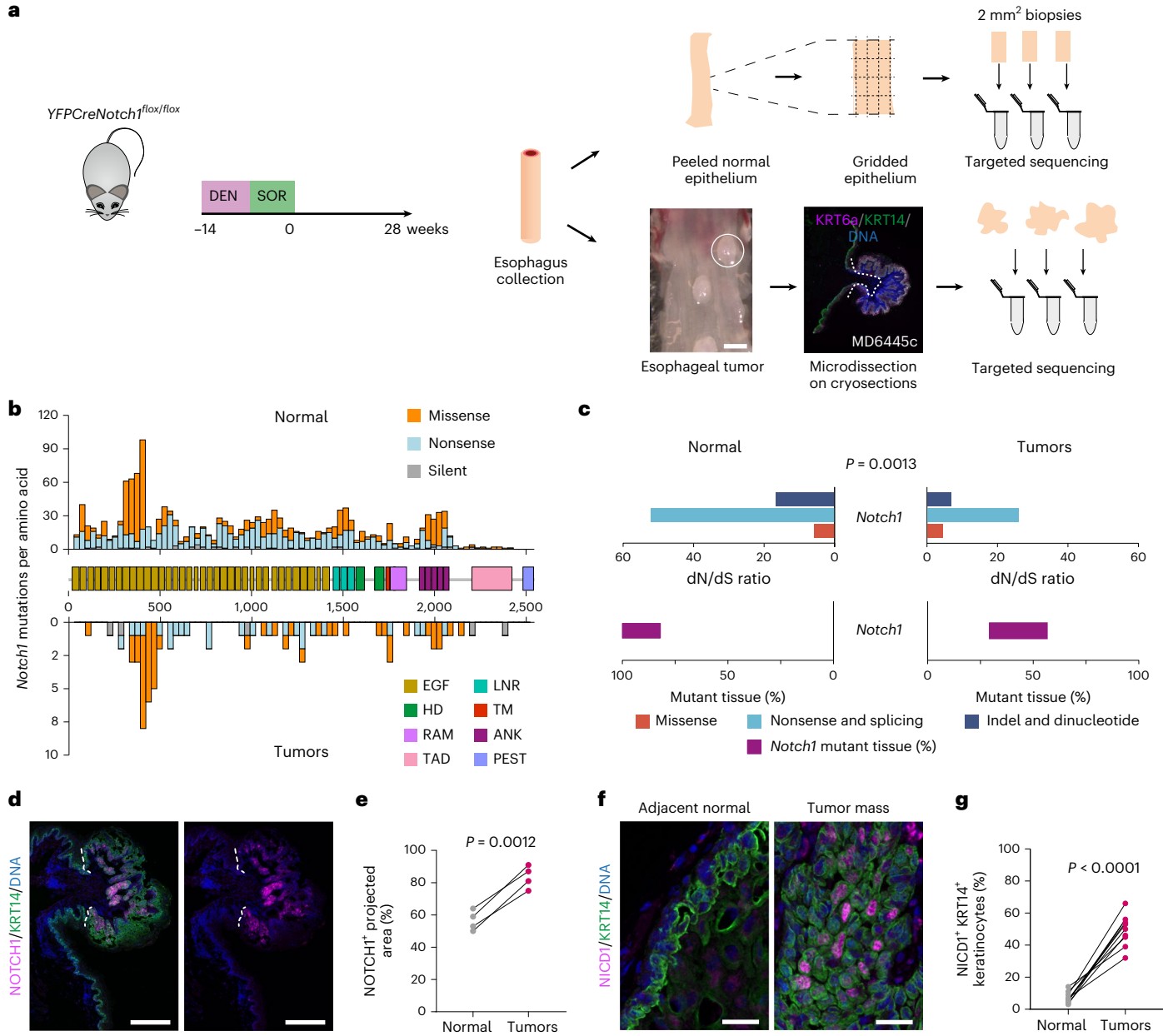

**Fig. 6 | Tumors retain functional *Notch1* in carcinogenesis. a,** Uninduced *YFPCreNotch1^flox/flox* mice were treated with DEN and SOR. Tissue was collected 28 weeks after treatment. Tumors were dissected from underlying submucosa and normal epithelium was cut into a gridded array of 2 mm² samples before targeted sequencing. Scale bar, 1 mm. **b,** Number of *Notch1* mutations per amino acid is plotted by NOTCH1 protein domains in normal gridded biopsies (upper) and tumors (lower) from *Notch1* wild type mice (normal, *n* = 115 biopsies from six mice; tumors, *n* = 17 biopsies from seven mice). Domains: EGF-like repeats, LNR, HD, TM, transmembrane, RAM, RBP-J-associated module, ANK, ankyrin repeats, TAD, trans-activation domain, PEST, rich in proline, glutamate, serine and threonine. **c,** dN/dS ratio for *Notch1* mutations (top plot) and proportion of *Notch1* mutant tissue in normal epithelium (purple bars) (*n* = 115 biopsies from six mice) and tumors (*n* = 17 biopsies from seven mice). Two-tailed *P* value,

likelihood ratio test of dN/dS ratios[2]. **d,** Representative NOTCH1 (magenta) and KRT14 staining (green) in tumors and surrounding tissue, DNA is blue. Image typical of 10 tumors from six animals. White dashed lines delineate tumor from adjacent normal tissue. Scale bars, 250 μm. **e,** Proportion of NOTCH1⁺ staining area in normal epithelium and tumors from the same control animals (each dot represents a mouse, *n* = 40 tumors from four mice). Two-tailed paired Student's *t* test. **f,** Representative images showing nuclear NICD1 (magenta) in keratinocytes (KRT14, green) inside a tumor in comparison to the normal adjacent tissue. DNA is blue. Image typical of 10 tumors from six animals. Scale bars, 25 μm. **g,** Proportion of KRT14⁺ keratinocytes with nuclear NICD1 staining in tumors and surrounding epithelium in the same sections (each dot represents a tumor, *n* = 10 tumors from six mice). Two-tailed paired Student's *t* test. See Supplementary Tables 19–23.

peeled epithelium from wild type, and highly induced, fully colonized *Notch1^+/−* and *Notch1^−/−* esophagus (Extended data Fig. 3f–h and Extended data Fig. 6a–e). In comparison with wild-type tissue, 20 genes in *Notch1^+/−* and 227 genes in *Notch1^−/−* esophagus were differentially expressed (*P* adjusted <0.05, Extended data Fig. 6b–d and

Supplementary Tables 12,13). These included the *Notch1*-regulated genes *Igfbp3* and *Sox9* (Supplementary Table 14)[18,29,30]. Gene set enrichment analysis (GSEA) showed that transcripts of genes involved in DNA replication were downregulated in *Notch1^−/−* colonized epithelium (Extended data Fig. 6e and Supplementary Table 15).

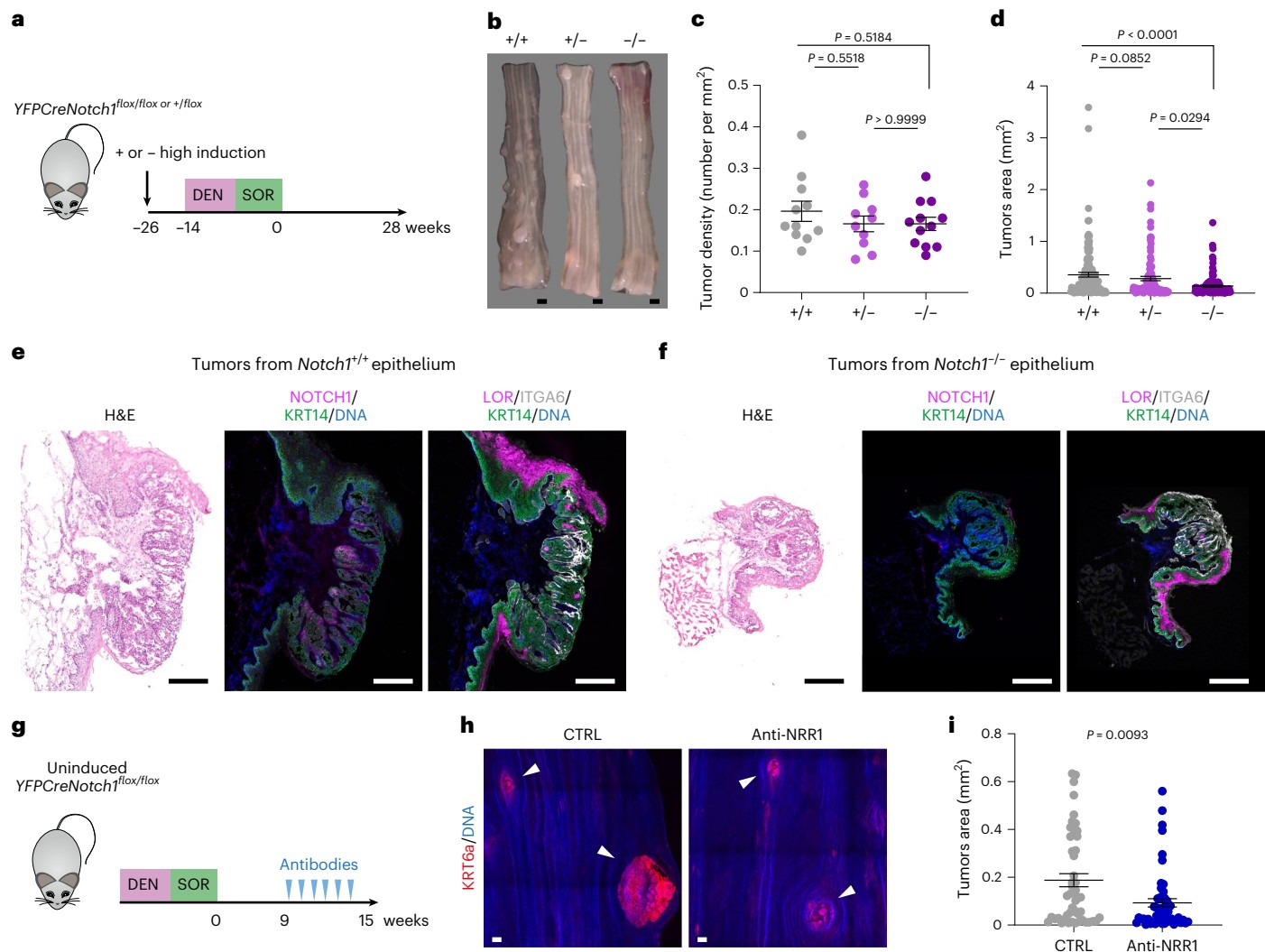

**Fig. 7 | Tumor growth is reduced by *Notch1* inactivation. a**, Highly induced *YFPCreNotch1*^flox/flox or +/flox^ (+/−) and *YFPCreNotch1*^flox/flox^ (−/−) mice or uninduced control (+/+) mice were treated with DEN and SOR and aged for 28 weeks. For **b**–**d**, *Notch1*^+/+^, $n = 11$; *Notch1*^+/−^ $n = 10$; *Notch1*^−/−^, $n = 12$. **b**, Representative images of esophagi for each genotype. Scale bar, 1 mm. **c**, Tumor density per genotype. Mean ± s.e.m., each dot represents a mouse. One-way ANOVA; adjusted *P* values from Tukey's multiple comparisons test. **d**, Tumor areas per genotype. Mean± s.e.m., each dot represents a tumor. Kruskal–Wallis test; adjusted *P* values from Dunn's multiple comparisons test. **e**,**f**, Tumors from *Notch1*^+/+^ (**e**) and *Notch1*^−/−^ (**f**) epithelium were sectioned and stained for H&E (left panel), for keratinocyte progenitor marker Keratin 14 (KRT14, green), and NOTCH1 (magenta) (middle panel) or keratinocyte differentiation marker Loricrin (LOR, magenta) and progenitor markers ITGA6 (gray) and KRT14 (green) (right panel). DNA is blue. Images representative of $n = 19$ tumors from *Notch1*^+/+^ and $n = 13$ tumors from *Notch1*^−/−^ epithelium. Scale bars, 250 μm. **g**, Uninduced *YFPCreNotch1*^flox/flox^ mice (+/+) were treated with DEN/SOR and aged for 9 weeks. Mice were treated with anti-NOTCH1 NRR1.1E3 or with CTRL for 6 weeks before collection. **h**, Representative tumors marked by KRT6a staining (red) are shown with white arrowheads in esophageal epithelium from control and anti-NRR1.1E3 treated mice. Scale bars: 100 μm. **i**, Quantification of tumor area (mean ± s.e.m., each dot represents a tumor, $n = 4$ mice per group). *P* values from two-tailed Mann–Whitney test. Data are shown in Supplementary Table 24.

To phenotype fully colonized *Notch1*^−/−^ epithelium, we performed single-cell RNA-seq (scRNA-seq) on highly induced *YFPCreNotch1*^flox/flox^ and uninduced control mouse esophagus (Fig. 4a–i, Extended data 7a–k and Supplementary Table 16). After filtering out poor-quality cells, a total of 13,111 cells remained for analysis, from two biological replicates per genotype (Fig. 4b and Supplementary Note). The proportions of keratinocytes, fibroblasts, immune and endothelial cells were similar in both genotypes, confirmed by staining esophageal sections (Fig. 4c and Extended data Fig. 7b–d)[31]. Keratinocytes showed no significant difference in density in Uniform Manifold Approximation and Projection (UMAP) space between the two genotypes (Fig.4d and Supplementary Note). The analysis revealed a continuum of keratinocyte cell states, from progenitors expressing *Krt14* to differentiating cells expressing *Krt4* or *Tgm3* to cornified cells expressing *Lor* (Extended data Fig. 7h–k).

We used these markers to discriminate basal and suprabasal cells in UMAP space, finding similar proportions of both populations in control and *Notch1*^−/−^ epithelium (Fig. 4e and Supplementary Note). In a further analysis, we assigned keratinocytes to cycling basal, resting basal or differentiating cells, finding no substantial differences between genotypes[32] (Fig. 4f–i and Supplementary Note).

To validate the scRNA-seq findings, we performed a cell-tracking assay. Mice with *Notch1*^−/−^ esophageal epithelium and littermate controls were injected EdU and 5-bromo-2′-deoxyuridine (BrdU) at 48 h and 1 h, respectively, before collection (Extended data Fig. 7l). Staining for EdU revealed the fate of S phase cells over the following 48 h, BrdU^+^ cells were currently in S phase. Cells were also stained for phospho-Histone H3 (pHH3), a G2/M phase marker (Extended data Fig. 7m). The ratio of suprabasal EdU^+^:total EdU^+^ cells reflecting the

generation of differentiating cells and their stratification, the proportion of BrdU[+] basal cells and the percentage of pHH3[+], BrdU[−] basal cells were all similar in wild type and *Notch1*[−/−] epithelium, consistent with the scRNA-seq findings (Extended data Fig. 7n–r, Supplementary Table 17 and Supplementary Note).

We also examined the epithelium in induced *YFPCreNotch1*[flox/flox] mice and control littermates that were aged 52 weeks. Tissue thickness, basal cell density and expression of the differentiation markers KRT14, KRT4 and LOR and the proliferation marker Ki67 were similar in both genotypes, (Fig. 5a–d and Supplementary Table 18). Pulse labeling and short-term lineage tracing for 48 h with EdU confirmed no significant difference in the proportion of S phase cells or in the stratification of differentiating cells, respectively, between *Notch1*[−/−] and wild-type esophagus (Fig. 5e–h).

We conclude that once *Notch1*[−/−] cells have occupied the epithelium, their behavior reverts toward that of wild-type cells so that tissue integrity is maintained.

## *Notch1* loss slows tumor growth

Next, we explored the role of *Notch1* in esophageal carcinogenesis. We began by treating *YFPCreNotch1* wild-type mice with the mutagen diethylnitrosamine (DEN), and sorafenib (SOR), a protocol that generates high-grade dysplastic lesions[33]. Tissue was collected after aging 28 weeks (Fig. 6a). Deep targeted sequencing of 73 cancer-associated and Notch pathway genes was performed on macroscopic tumors and a gridded array of normal epithelium (Fig. 6a and Supplementary Tables 19–22).

The normal epithelium contained a high density of clones carrying protein-altering mutations. To determine which genes conferred a clonal advantage, we calculated the ratio of silent to protein-altering mutations in each gene, dN/dS[3,34]. Mutant genes under positive selection with a dN/dS ratio substantially above 1 (*q* < 0.05) were the Notch pathway genes *Notch1, Notch2* and *Adam10*, plus *Fat1, Trp53* and *Arid1a*, all of which are selected in normal human esophagus along with *Ripk4* and *Chuk* (Supplementary Table 21)[2,21].

In tumors, the most prevalent mutant gene was the known mouse esophageal tumor driver *Atp2a2*, which is not selected in normal epithelium (Extended data Fig. 8a,b and Supplementary Tables 19–22)[35,36]. Protein-altering *Notch1* mutations were under weaker selection and less prevalent in tumors than in the adjacent epithelium (Fig. 6b,c, Extended data Fig. 8a,b and Supplementary Tables 19–22). Immunostaining confirmed more cells stained positive for NOTCH1 and NICD1 in tumors than in normal tissue (Fig. 6d–g and Supplementary Table 23). These findings parallel observations in humans and indicate *Notch1* wild-type cells are more likely to contribute to tumors than those carrying *Notch1* mutations[2,5].

Next, we used a high induction protocol to delete one or both alleles in the entire esophageal epithelium of *YFPCreNotch1*[flox/flox] and *YFPCreNotch1*[+/flox] mice before DEN and SOR treatment. Uninduced littermates were used as controls (Fig. 7a). The density of tumors was similar in all three genotypes, arguing *Notch1* is not required for tumor initiation (Fig. 7b,c and Supplementary Table 24). However, tumors were significantly smaller in *Notch1*[−/−] epithelium, in which immunostaining confirmed the loss of *Notch1* expression and function (Fig. 7d–f and Supplementary Table 24). Immunostaining for markers of differentiation (LOR, ITGA6 and KRT14) showed multiple layers of undifferentiated keratinocytes in lesions of both genotypes. Markers of apoptosis (cleaved caspase 3), endothelial cells (CD31) and immune cells (CD45) were also similar in tumors from *Notch1*[−/−] and *Notch1*[+/+] epithelium (Fig. 7e,f and Extended data Fig. 8d,e). CDH1 loss contributes to tumorigenesis[37]. Tumors from *Notch1*[+/+], but not *Notch1*[−/−], esophagus displayed focal loss of CDH1 expression (Extended data Fig. 8f–h and Supplementary Table 24).

These observations argue that *Notch1* favors tumor growth. To test this hypothesis, we treated wild-type mice with a NOTCH1 function blocking antibody (anti-NRR1.1E3)[38]. The antibody reduced levels of cleaved NOTCH1 in esophageal epithelium, abolished nuclear NICD1 immunostaining and altered levels of multiple transcripts encoding *Notch1* loss of function markers (Extended data Fig. 9a–e, Extended data Fig. 6d and Supplementary Table 25). Anti-NRR1.1E3 also reduced the expansion of *Notch1*[−/−] clones in clonally induced *YFPCreNotch1*[flox/flox] mice by inhibiting NOTCH1 signaling in wild type cells (Extended data Fig. 9f–i and Supplementary Table 25). Wild-type mice were given DEN and SOR, tumors allowed to develop for 9 weeks and anti-NRR1.1E3 or control antibody given for 6 weeks (Fig. 7g). Anti-NRR1.1E3 significantly reduced tumor size compared with control, indicating NOTCH1 signaling favors the growth of established lesions (Fig. 7h,i and Supplementary Table 24).

To understand how *Notch1* loss alters tumor growth, we sequenced tumors from *Notch1*[−/−] epithelium, finding they share the same driver mutation, *Atp2a2*, (6/7 tumors), as the tumors from *Notch1*[+/+] epithelium (17/17 tumors) (Extended data Fig. 8a–c and Supplementary Tables 20 and 26)[35,36]. Comparison of transcriptomes of tumors and adjacent normal tissue showed an upregulation of transcripts encoding genes linked with DNA replication, cell cycle and RNA processing and downregulation of mRNAs associated with lipid metabolism in tumors of both genotypes (Fig. 8a–c, Extended Data Fig. 10a,b and Supplementary Tables 27 and 28). These changes are consistent with the reported effects of *Atp2a2* mutation on keratinocytes[35,36,39,40]. Comparison of tumors from *Notch1*[+/+] and *Notch1*[−/−] epithelium revealed DNA replication and cell-cycle-associated transcripts were significantly downregulated in *Notch1*[−/−] tumors (Fig. 8d–f, Extended data Fig. 10c,d and Supplementary Tables 29 and 30). Furthermore, the proportion of cycling cells expressing pHH3 and CCNB1 within KRT14[+] cells was reduced in tumors from *Notch1*[−/−] compared to *Notch1*[+/+] esophagus

**Fig. 8 | Cell division is decreased in tumors from *Notch1*–/– esophagus.**
**a**, *Notch1*[+/+] and *Notch1*[−/−] normal esophageal tissue and tumors (Fig. 7a) were RNA sequenced. *Notch1*[+/+]: *n* = 11 epithelial samples from seven mice, *n* = 8 *Notch1*[+/+] tumors from four mice; *Notch1*[−/−] *n* = 10 epithelial samples from seven mice, *n* = 6 *Notch1*[−/−] tumors from five mice. **b**, MA plots showing differentially expressed genes (red, *q* < 0.05, DESeq2 analysis, two-sided Wald test with Benjamini−Hochberg correction), red, in *Notch1*[+/+] and *Notch1*[−/−] tumors versus normal epithelium. Zero-fold change shown by red dotted line. **c**, −log$_{10}$ (*P* value) of top Gene ontology biological processes (GOBP) in tumors versus normal epithelium in *Notch1*[+/+], gray, *Notch1*[−/−], red genotypes (Supplementary Tables 27 and 28). **d**, MA plots showing differentially expressed genes (red, *q* < 0.05, DESeq2 analysis, two-sided Wald test with Benjamini−Hochberg correction), in tumors from *Notch1*[−/−] versus *Notch1*[+/+] esophagus. Red dotted line, zero-fold change. **e**, GSEA of tumors from *Notch1*[−/−] versus *Notch1*[+/+] esophagus, DNA replication gene set shown (normalized enrichment score, NES = −2.48, false discovery rate, FDR *q*-value = 0.0, Supplementary Table 29). **f**, Transcript per million values of cell cycle and DNA replication transcripts selected from GSEA in tumors from *Notch1*[+/+] and *Notch1*[−/−] esophagus. Mean ± s.e.m., *n* = 8 tumors from *Notch1*[+/+] esophagus and *n* = 6 from *Notch1*[−/−] esophagus. Two-tailed unpaired Student's *t*-test. **g**, Representative images of *n* = 8 tumors from *Notch1*[+/+] and *n* = 9 tumors from *Notch1*[−/−] esophagus. KRT14 (green), pHH3 (gray). DNA, blue. Scale bars, 30 μm. **h**, Percentage of pHH3[+], KRT14[+] keratinocytes within tumors from *Notch1*[+/+] and *Notch1*[−/−] esophagus. Mean ± s.e.m., each dot represents a tumor, +/+: *n* = 8 tumors from 4 mice; −/−: *n* = 9 tumors from 7 mice. Two-tailed unpaired Student's *t*-test. **i**, Representative images from *n* = 8 tumors each from *Notch1*[+/+] and *Notch1*[−/−] esophagi, KRT14 (green), phospho-ERK1/ERK2 (p-ERK, magenta), DNA (blue). Insets, magnified areas indicated by white squares. Scale bars, 30 μm. **j**, Normalized mean intensity of fluorescence for p-ERK (left) and total ERK (t-ERK, right) in KRT14[+] cells in tumors from *Notch1*[+/+] and *Notch1*[−/−] esophagi relative to adjacent normal tissue. Mean ± s.e.m., each dot represents a tumor. +/+: *n* = 8 tumors from four mice; −/−: *n* = 8 tumors from seven mice. Two-tailed unpaired Student's *t* test. **k**, In tumors lacking *Notch1*, signals downstream of mutant *Atp2a2* are disrupted, cell division reduced, and tumor growth slows. A.U., arbitrary unit. See Supplementary Tables 24 and 27–30.

(Fig. 8g,h, Extended data Fig. 10e,f and Supplementary Table 24). Finally, as RAS/MEK/ERK signaling is activated in *Atp2a2* mutant cells, we measured phospho-ERK1/ERK2 and total ERK1/ERK2 staining finding a significant decrease of the former in tumors from *Notch1*[−/−] compared to *Notch1*[+/+] epithelium (Fig. 8i,j and Supplementary Table 24)[36,40]. These findings are consistent with attenuated signaling downstream of mutant *Atp2a2* in tumor cells lacking *Notch1* (Fig. 8k).

## Discussion

These results shed light on the disparity in the prevalence of *NOTCH1* mutations in normal esophageal epithelium and tumors[2,5]. Mutations reducing the function of one *Notch1* allele confer a competitive advantage on mutant progenitors, making it likely they will form persistent, expanding clones. As the heterozygous mutant population grows, the probability that the remaining allele will be lost increases. When this happens, it

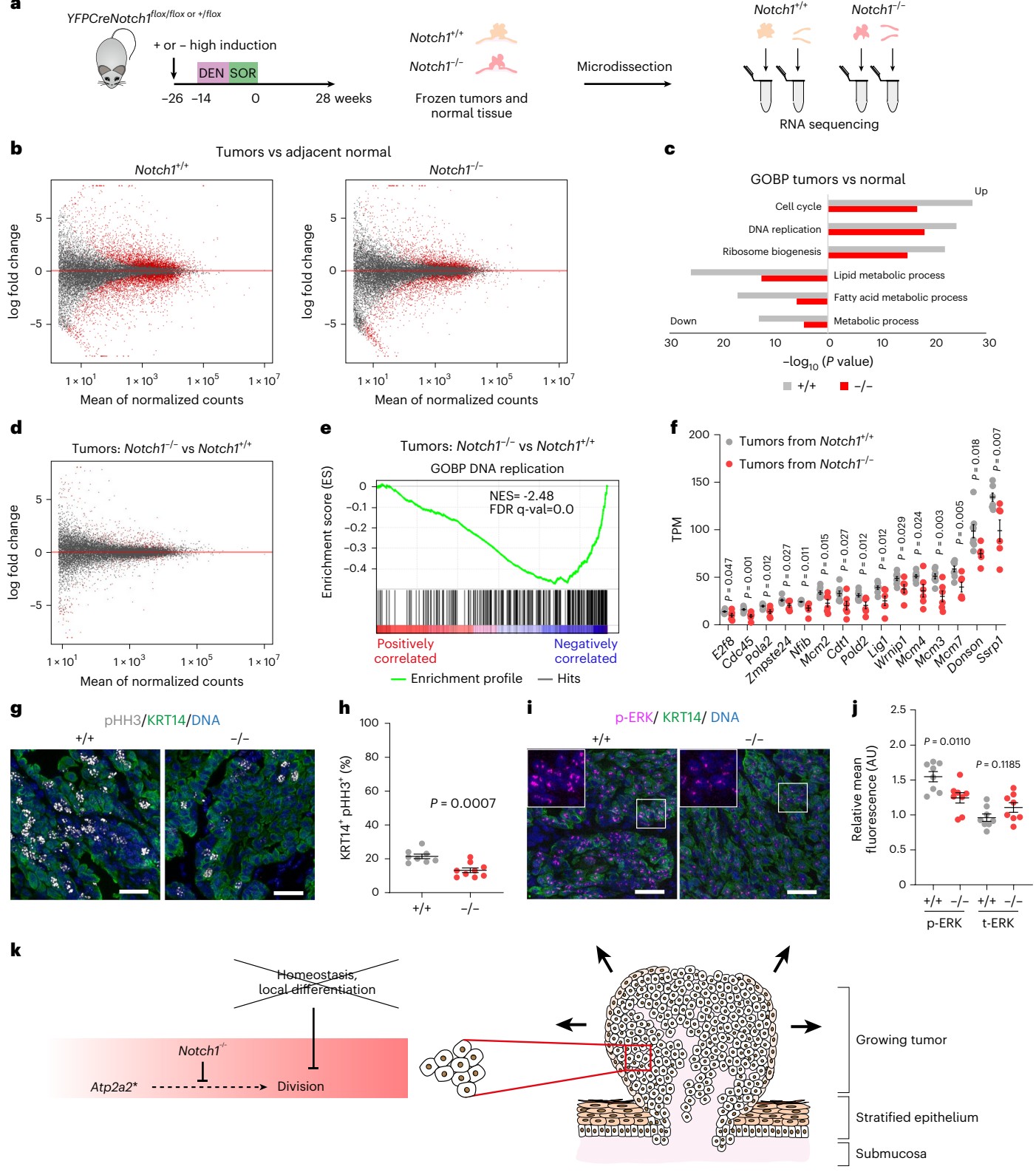

**g** pHH3/KRT14/DNA

**i** p-ERK/KRT14/DNA

confers a further increase in fitness (Fig. 3h). By driving wild-type cell differentiation, *Notch1* null cells at the clone margins can divide, resulting in extensive colonization of the epithelium (Fig. 2o). This mechanism explains how clones with biallelic *NOTCH1* disruption dominate normal human esophagus. Such 'supercompetition' also occurs in the intestine where *Apc* mutant intestinal stem cells drive the differentiation of their wild-type neighbors to colonize the intestinal crypt[41].

Once an area has been colonized by biallelic *Notch1* mutants, the phenotype of mutant cells reverts toward that of wild-type cells. This reversion toward a near-normal cell state explains the normal appearance of aged human esophageal epithelium despite NOTCH1 signaling being disrupted in most of the tissue.

In *Atp2a2* mutant tumors, the constraint that links cell division to the exit of differentiating cells from the basal cell layer to maintain cellular homeostasis does not operate. In this context, the faster cells divide, the faster the lesion will expand. As loss of *Notch1* slows the cell division rate, $Notch1^{-/-}$ lesions are smaller than wild-type tumors (Fig. 8k).

Might these findings be relevant to humans? Over 90% of human esophageal squamous cell carcinoma (ESCC) retain one or more wild-type copies of *NOTCH1* but develop from epithelium where a high proportion of cells have biallelic *NOTCH1* disruption, arguing wild-type *NOTCH1* favors ESCC development. What of the subset of ESCC that does have biallelic *NOTCH1* disruption?[5] One possibility is that NOTCH1 loss, in association with multiple other genomic alterations, promotes transformation in these cases. Alternatively, it is plausible that the *NOTCH1* alterations in these tumors are 'passengers', carried over from normal tissue with the requirement for wild-type *NOTCH1* in carcinogenesis bypassed by other genome changes.

*Notch1* illustrates how inactivating mutations in the same gene can drive clonal expansion in normal tissue but impair tumor growth. This is due to the differences in cell dynamics between wild-type normal tissue and a mutated tumor. Our results highlight the potential of NOTCH1 blockade in reducing the growth of premalignant tumors. NOTCH1 inhibitors are in clinical development, and investigation of their potential in esophageal neoplasia seems warranted.

## Online content

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

## Methods

### Human samples

**Ethical approval.** The study protocol was ethically reviewed and approved by the UK National Research Ethics Service Committee East of England−Cambridge South, Research Ethics Committee under protocol reference 15/EE/0152 NRES.

**Collection.** Esophageal tissue was obtained from deceased organ donors. Written Informed consent was obtained from the donor's relatives. A sample of mid-esophagus was removed, placed in University of Wisconsin (UW) organ preservation solution (Belzer UW Cold Storage Solution, Bridge to Life) and flash frozen in tissue freezing medium (Leica, 14020108926)[2].

**Immunostaining.** Triplet 10 μm serial cryosections fixed with 4% paraformaldehyde for 10 min were stained for NOTCH1/KRT4/ITGA6/DNA (section 1), NICD1/KRT14/DNA (section 2), and NOTCH1/KRT14/ITGA6/DNA (section 3) and imaged (see Histology and Confocal microscopy sections). Corresponding areas in each section were identified. Contiguous regions staining positive or negative for NOTCH1 or nuclear NICD1 were identified and their length was measured using Volocity 6 software (Perkin Elmer). For morphological analysis of NICD1+ and NICD1− areas, sections were stained for NICD1/KRT14/DNA or NICD1/Ki67/DNA. Epithelial thickness, cell counting and density measurement were performed using Volocity 6 software (Perkin Elmer).

**DNA sequencing.** Sampling, library preparation, targeted sequencing processing and analysis are detailed in Supplementary Note.

### Animals

All experiments were ethically reviewed under and conducted in accordance with the UK Home Office Project Licenses 70/7543, P14FED054 and PF4639B40. Both male and female adult mice of 10−16 weeks of age at the start of the experiments were used. Animals were housed in individually ventilated cages and fed on standard chow. Mice were maintained at SPOF health status. B6.129 × 1-*Notch1*[tm2Rko/GridJ] mice were purchased from the Jackson Laboratory and crossed with *Rosa26*[floxedYFP] and *AhCre*[ERT] to generate *YFPCreNotch1* triple mutant mice (Extended data Fig. 2b−d)[15,19,42]. C57BL/6J wild-type mice were also used as indicated.

### qPCR recombination assay

**Design of the assay.** Specific primer sets were designed to analyze excision of the floxed exon 1 of *Notch1* by *Cre* recombinase (Extended data Fig. 3c). Primer set A allows intragenic normalization using the nonfloxed *Notch1* exon 3; primer set B measures the disappearance floxed exon 1 with recombination; primer set C specifically detects exon 1 recombination (primer sequences are provided in Supplementary Table 32). Quantitative PCR on genomic DNA was carried out using specific primers and SYBR Green master mix (Thermo Fisher Scientific, 4309155) according to the manufacturer's instructions in a StepOnePlus Real-Time PCR System (Thermo Fisher Scientific, 4376600). Relative qPCR expression was calculated using delta−delta Ct method, a wild type or *Notch1*[−/−] reference sample was used within the same assay for set B or set C, respectively. Validation of the linearity of the recombination assay was performed against a standard curve reproducing different recombination rates with Exon 1/Exon 3 ratios of 1, 0.75, 0.5, 0.25 and 0. The standard curve was made using diluted genomic DNA from the esophagus of highly induced and fully recombined *Notch1*[−/−] mice (as verified by qPCR, staining and protein assay) and from *Notch1* wild-type tissue.

**Recombination status in highly induced tissues.** Genomic DNA was extracted from large pieces of freshly peeled epithelium using either AllPrep DNA/RNA mini kit (Qiagen) or QIAamp DNA micro kit (Qiagen,

56304) and qPCR assay was performed using set B. Full recombination of the esophageal epithelium will reduce the Exon 1/Exon 3 ratio to zero in induced *Notch1*[−/−] mice and halve it in induced *Notch1*[+/−] mice compared to wild-type mice.

**Detection of the recombined allele in microdissected fixed tissue.** Clonally induced tissues were fixed and stained for NOTCH1 and YFP at 4 weeks postinduction for *YFPCreNotch1*[flox/flox] mice and 13 weeks postinduction for *YFPCreNotch1*[+/flox] mice. NOTCH1 detection and intensity measurement were used to resolve *Notch1*[−/−] and *Notch1*[+/−] clones, respectively (Supplementary Note). Putative clonal and control areas were then microdissected from the esophageal epithelium. Clonal microdissection was carried out under a Fluorescent Stereo Microscope Leica M165 FC (Leica) using 0.25 mm diameter punch (Stoelting, 57391) as shown in Extended data Fig. 3k−n. gDNA from the microbiopsies was extracted using Arcturus PicoPure DNA extraction kit (Applied Biosystems, 11815-00) following the manufacturer's instructions. gDNA extracted from fixed tissue is fragmented, altering the linearity of the qPCR assay. Therefore, set C rather than Set B was used to determine the recombination status of the microbiopsies as specific detection of the recombined allele above background noise was sufficient to conclude on a reliable discrimination of mutant clones. Nonetheless, on average recombined exon 1 detection increased two folds in *Notch1*[−/−] clones compared to *Notch1*[+/−] clones.

### RT-qPCR assay

RNA extractions were performed on peeled mouse esophageal epithelium as described in the RNA-seq method section (Supplementary Note). Total RNA was measured using Qubit RNA BR Assay Kit (Thermo Fisher Scientific, Q10211). cDNA synthesis of 500 ng total RNA was performed using QuantiTect Reverse Transcription Kit (Qiagen, 205313). RT-qPCR was performed with Taqman Fast Advanced Master Mix (Thermo Fisher Scientific, 4444557) on StepOnePlus Real-Time PCR System (Thermo Fisher Scientific, 4376600) and analyzed using StepOne Software v2.3. Relative qPCR expression to *Gapdh* housekeeping gene was calculated using delta−delta Ct method. The Taqman assays used for quantification are shown in Supplementary Table 32.

### Immune capillary electrophoresis

RLT Plus lysates with Complete Protease Inhibitor (Roche, 11836170001) homogenized as described in the 'RNA-seq' section (Supplementary Note) were passed through the RNA binding column from the AllPrep DNA/RNA Mini kit (Qiagen) and the flow through was collected for protein precipitation. For precipitation, nine volumes of ice-cold pure Ethanol were mixed with the lysates before storage overnight at −80 °C. Precipitates were spun for 30 min at 20,000g at 4 °C, pellets were dried and solubilized progressively with 5% Sodium dodecyl sulfate in 100 mM TEAB solution (Sigma-Aldrich, T7408). Total protein quantification was performed using Pierce BCA Protein Assay Kit (Thermo Fisher Scientific, 10678484). Immune capillary electrophoresis was performed using Wes Simple (ProteinSimple) following manufacturer's instructions and analyzed using Compass for SW version 4.1.0. Primary antibodies were the following: anti-NOTCH1 targeting C terminus of the protein (Cell signaling, 3608); anti-NOTCH2 targeting C terminus of the protein (Cell signaling, 5732); anti-α-Tubulin (Cell signaling, 2125).

### Whole-mount preparation of mouse esophagus

**Tissue preparation.** Mouse esophagus was opened longitudinally and the muscle layer was removed with forceps. For lineage tracing and EdU/BrdU experiments, tissue was incubated for 15 min in Dispase I (Roche, 04942086001), diluted at 1 mg ml⁻¹ in PBS before separating the epithelium with fine forceps. For all other immunostaining experiments (including long-term antibody treatment), tissue was incubated for 2 h 15 min to 3 h in 5 mM EDTA at 37 °C before peeling the epithelium. The epithelium was then flattened and fixed in 4% paraformaldehyde

for 1 h 15 min at room temperature under agitation, washed in PBS and stored in PBS at 4 °C[21].

**Whole-mount immunostaining.** Whole-mount tissues were stained as previously described[43]. Tissues were incubated for 1 h in staining buffer (0.5% BSA, 0.25% fish skin gelatin, 0.5% Triton X-100 and 10% donkey serum in PHEM). This blockage step was followed by incubation with primary antibodies (Supplementary Table 31) in staining buffer overnight at room temperature, three washes of 30 min with 0.2% Tween-20 in PHEM and incubation with secondary antibodies (Supplementary Table 31) in staining buffer for 3 h at room temperature. After further washes, tissues were incubated for an hour at room temperature with 1 µg ml⁻¹ DAPI or 0.5 µM Sytox Blue solution (Biolegend, 425305) to stain cell nuclei and mounted using Vectashield mounting media (Vector Laboratories, H-1000).

### Histology

#### Hematoxylin and eosin staining (H&E).
H&E was either performed on 10 µm cryosectioned tissue processed as described below or on 5 µm paraffin-embedded tissue sections. Before paraffin embedding, esophageal tissue was collected and fixed in 4% paraformaldehyde for at least 2 h before undergoing progressive dehydration in Tissue-Tek VIP 6 AI tissue processor (Sakura). Slides were then scanned at objective ×20 using NanoZoomer S60 Digital slide scanner (Hamamatsu).

#### Immunostaining on esophageal sections.
Esophageal tissue was flash frozen in tissue freezing medium (Leica, 14020108926). Ten micrometer transverse sections were fixed with 4% paraformaldehyde for 10 min, blocked in staining buffer (0.5% BSA, 0.25% fish skin gelatin, 0.5% Triton X-100 and 10% donkey serum in PHEM) and stained with primary and secondary antibodies for 3 h to overnight at room temperature (Supplementary Table 31). PHEM washes were performed between incubations. Before NICD1 staining, sections were incubated 20 min in 50 mM Glycine/PBS solution. Finally, tissues were incubated for an hour at room temperature with 1 µg ml⁻¹ DAPI or 0.5 µM Sytox Blue solution (Biolegend, 425305) to stain cell nuclei and mounted in Vectashield mounting media (Vector Laboratories, H-1000). For Extended data Fig. 3b, freshly collected esophagus was fixed in 4% PFA for 2 h and embedded in 4% low-melting agarose. Hundred micrometer thick Vibratome (Leica) sections were cut permeabilized for 1 h and stained as for whole mounts.

### Confocal microscopy

Immunofluorescence images were acquired on a Leica TCS SP8 confocal microscope using ×10, ×20 or ×40 objectives. Typical settings for acquisition were optimal pinhole, line average 3 and 4, and scan speed 400–600 Hz and a resolution of 1024 × 1024 pixels. Visualization and image analysis were performed using Volocity 6 Image Analysis software (PerkinElmer).

### Lineage tracing using a YFP reporter

To induce recombination at *Notch1* and *Rosa26* loci, *YFPCreNotch1* mice were injected intraperitoneally (i.p.) with β-Naphthoflavone (BNF, MP Biomedicals, 156738) and tamoxifen (TAM, Sigma-Aldrich, N3633). To induce recombination at clonal level in the esophageal epithelium, mice were treated with BNF (80 mg kg⁻¹) and TAM (0.125 mg)[42]. Excision of the *Notch1* allele and expression of the YFP reporter at the *Rosa26* locus can occur in the cells either in combination or separately, resulting in four different populations of cells, *Notch1* mutant or not, and expressing YFP or not. Peeled epithelium was stained for NOTCH1, YFP and DNA, imaged with confocal microscopy at ×40 objective and YFP⁺ basal and suprabasal cells were counted. In induced *Notch1^{flox/flox}* tissue, YFP + clones expressing NOTCH1 were categorized as wild type (+/+), while the clones without detectable expression of NOTCH1 were classified as *Notch1^{-/-}*. For example, in *YFPCreNotch1^{flox/flox}* induced mice at

4 weeks postinduction, we observed 67 ± 2% of YFP⁻; *Notch1^{-/-}* clones; 6 ± 1% YFP⁺; *Notch1^{-/-}* clones and 27 ± 3% of YFP⁺; *Notch1^{+/+}* clones (data obtained from three mice). In induced *Notch^{+/flox}* tissue, YFP⁺ wild type (+/+) and YFP⁺ *Notch1^{+/-}* clones were distinguished using NOTCH1 staining measurement (Supplementary Note).

### EdU lineage tracing

EdU incorporates during replication in the proliferating cells located in the basal layer of the esophageal epithelium. EdU i.p. injection at 10 µg was performed either at 1 h or 48 h before tissue collection. Tissue was processed and EdU was detected in whole mount using Click-iT EdU imaging kit (Life Technologies, C10338 or C10340).

### EdU/BrdU lineage tracing

Mice were injected with EdU i.p. injection at 10 µg 48 h before tissue and with BrdU i.p. at 1 mg 1 h before collection to label cells in S phase. Tissue was processed as in 'Whole-mount sample preparation of mouse esophagus'. For immunostaining, tissue was first incubated for 30 min in permeabilization buffer (0.5% BSA, 0.25% fish skin gelatin, 0.5% Triton X-100 in PHEM) followed by 20 min at 37 °C in DNAse buffer containing 500 units DNAse under 500 rpm agitation (NEB, M0303L). Tissue was washed three times in PBS for 20 min. Samples were then processed as described in 'whole-mount immunostaining'. EdU was detected in whole mount using Click-iT EdU imaging kit (Life Technologies, C10338). BrdU was detected using primary antibody anti-BrdU (Abcam, ab6326). PHH3 was detected using conjugated Alexa Fluor 647 Anti-Histone H3 (phospho S10) antibody (Abcam, ab196698) (Supplementary Table 31).

### Aging experiments

*YFPCreNotch1* mice between 10 and 16 weeks of age were injected i.p. with BNF at 80 mg kg⁻¹ and TAM at 1 mg for *Notch1^{+/+}* and *Notch1^{+/flox}* and at 0.25 mg for *Notch1^{flox/flox}* and aged up to 78 weeks old. *Notch1^{+/+}* or noninduced mice were used as wild-type controls as indicated. A lower dose of Tamoxifen was used for the *YFPCreNotch1^{flox/flox}* mice to minimize the recombination of the *Notch1* allele in the corneal epithelium, possibly leading to corneal opacification and keratinization[44].

### Projected NOTCH1 stained area quantification

To quantify the percentage of NOTCH1⁺ or NOTCH1⁻ area in the entire esophageal epithelium or the projected surface of NOTCH1⁻ clones, whole-mount esophageal epithelia were prepared and stained for NOTCH1 and counterstained with DAPI or Sytox Blue as described in the dedicated sections. The entire epithelium was imaged using a high-precision motorized stage coupled to a Leica TCS SP8 confocal microscope. Typical settings for the acquisition of multiple z stacks were optimized 2.41 µm z step size, zoom ×1, optimal pinhole, line average 4, scan speed 400–600 Hz and a resolution of 1024 × 1024 pixels using an ×10 HC PL Apo CS Dry objective with a 0.4 numerical aperture. Images were processed using Volocity 6 software. To measure their projected surface area, we used the 'extended focus' visualization mode on the Volocity software. Regions of interest (ROI) were defined with ROI tool allowing surface area measurement. NOTCH1 staining was automatically detected based on the defined intensity and minimum object size.

### Carcinogen treatment

Mice were induced with BNF/TAM at a dose that allowed full coverage of the tissue with the mutant *Notch1* heterozygous or homozygous cells within 3 months. *YFPCreNotch1^{+/flox}* mice were injected i.p. on two consecutive days with BNF at 80 mg kg⁻¹ and TAM at 1 mg and YFPCre*Notch1^{flox/flox}* were injected once with BNF at 80 mg kg⁻¹ and TAM at 0.25 mg. Noninduced *YFPCreNotch1* mice were used as wild-type controls. Mice were then treated with DEN (Sigma-Aldrich, N0756) in sweetened drinking water (40 mg l⁻¹) for 24 h, 3 d a week for 8 weeks[18,21].

SOR (LC Chemicals, S8502) was then administered at 50 mg kg⁻¹ (5 μl of 10 mg ml⁻¹ solution per gram bodyweight) by i.p. injection on alternate days during 6 weeks, for a total of 21 doses[33].

Mice were aged for 28 weeks after the last dose of SOR and esophageal tissue was collected. Macroscopic images of unpeeled tissue were obtained under Leica M80 zoom stereomicroscope with Leica Plan ×1.0 Objective M-Series 10450167 coupled with Leica DFC295 Camera (Leica Microsystems). Macroscopic tumors were removed and flash frozen. Esophageal tissue was whole-mount immunostained for KRT6 and DNA. The projected area of lesions was determined using ROI tool in Volocity 6 software[45].

### Antibody treatment validation

Antibody validation was performed in two steps. First, C57BL/6J were injected i.p. at a high dose with rat anti-Notch1 NRR hybridoma Clone 1E3.19.1 (anti-NRR1.1E3, Genentech) at 25 mg kg⁻¹ (*n* = 3 mice per group). Antibody Ragweed:9652 10D9.W.STABLE mIgG2a (CTRL, Genentech) was used as control at 25 mg kg⁻¹. Three days later, esophageal tissue was collected and processed for RT-qPCR assay and protein quantification as described in the 'RT-qPCR' and 'Immune Capillary Electrophoresis assay' (ICE) sections. At the protein level, cleaved transmembrane/intracellular regions (NTM + NICD) of NOTCH1 and NOTCH2 were quantified by ICE to determine the specificity of anti-NRR1.1E3 for NOTCH1. NICD1 staining of cryosections confirmed the absence of active NOTCH1 in anti-NRR1.1E3 treated tissue. At the transcriptional level, RT-qPCR for *Notch1* loss of function markers *Igfbp3, Tgm2, Gli2, Adam8* and *Sox9* was performed. Second, *YFPCreNotch1*^flox/flox^ mice were induced at clonal level, and starting a week later, were treated weekly with antibodies for 3 weeks. Epithelial whole mounts were then stained for NOTCH1. NOTCH1 blockade would be expected to neutralize the competitive advantage of *Notch1*⁻/⁻ clones over wild-type cells, reducing clonal expansion[21]. Anti-NRR1.1E3 at 10 mg kg⁻¹ offered the greatest reduction in NOTCH1⁻ clone size. *n* = 4 mice injected with Ragweed controls at 10 mg kg⁻¹; *n* = 3 mice given NRR1.1E3 at 10 mg kg⁻¹; *n* = 2 mice given NRR1.1E3 10 mg kg⁻¹ loading dose (LO: week 1, 10 mg kg⁻¹; week 2, 7.5 mg kg⁻¹; week 3, 5 mg kg⁻¹); *n* = 1 mouse given NRR1.1E3 5 mg kg⁻¹; and *n* = 1 mouse given NRR1.1E3 5 mg kg⁻¹ loading (LO: week 1, 5 mg kg⁻¹; week 2, 4 mg kg⁻¹; week 3, 3 mg kg⁻¹). Antibody NRR1.1E3 did not cause weight loss or other adverse effects.

### Immunotherapeutic treatment

To analyze the effect of NOTCH1 neutralizing antibody on tumor growth, uninduced *YFPCreNotch1*^flox^ (wild type) mice were first treated with DEN and SOR and aged for 9 weeks to allow the tumors to start growing before starting a treatment with anti-NRR1.1E3 (Genentech) at 10 mg kg⁻¹ or with Ragweed control (Genentech) at 10 mg kg⁻¹ (*n* = 4 mice per group), once a week for 6 weeks. One week after the last dosage, tissue was collected and processed for macroscopic and microscopic quantification of the projected areas of the tumors using Volocity 6 software as described in 'carcinogen treatment' section.

### Cell density in mouse esophagus

Density of the basal cells was measured on whole-mount stained tissue, imaged at ×40 objective using Leica TCD SP8 confocal microscope (see 'confocal microscopy'). DAPI⁺ or Sytox Blue⁺ basal nuclei were quantified per area. For *Notch1* mutant clones and control areas, analysis was performed in seven to nine clones and paired areas from three *YFPCreNotch1*^+/flox^ mice at 13 weeks postinduction and in three to seven clones and paired areas from seven *YFPCreNotch1*^flox/flox^ mice at 4 weeks postinduction. For aged mouse tissue, analysis was performed at three to six random positions of the tissue, *n* = 4 mice per genotype.

### Cell counting and epithelial thickness

Epithelial thickness was quantified in cryosections stained with H&E with NanoZoomer Digital Pathology software (NDP.view2,

Hamamatsu). Measurements were performed at 18–23 positions and averaged for each mouse (*n* = 3 mice). For cell counting, stained sections were imaged by confocal microscopy and analyzed with Volocity 6 software (Perkin Elmer). Ki67⁺ basal cells were counted at three different positions per animal and averaged for each mouse (*n* = 3–4 mice). For tumor cells in G2/M phase, cryosections were stained for pHH3, CCNB1, KRT14 and DNA (*n* = 8 tumors from four wild-type mice; *n* = 9 tumors from seven mutant mice). For NICD1, sections of tumors and adjacent normal tissue were stained for NICD1, KRT14 and DNA. The proportion of KRT14⁺ keratinocytes expressing nuclear NICD1 inside the tumor mass and in the adjacent normal epithelium was quantified (*n* = 10 tumors from six mice).

### Fluorescence intensity quantification in tumors and normal tissue

Esophageal sections carrying tumors and adjacent normal tissue stained for CDH1 (E-cadherin), KRT14, and counterstained with DAPI were imaged using a Leica TCD SP8 confocal microscope. Mean intensity was quantified in ROI with the ROI tool in Volocity 6 software. Mean intensity of CDH1 was normalized to the mean intensity of DAPI at each ROI (Supplementary Table 24). For phospho-ERK1/2 (p-ERK) and total ERK1/2 (total ERK), sections were costained for KRT14 and DAPI, and were analyzed as above with the following modifications: within ROI defined on KRT14⁺ cells in adjacent normal or inside the tumors, p-ERK staining was automatically detected using the 'find objects' function of Volocity 6 software, using 12-255 intensity threshold and a minimum object size of 0.5 μm² and a restrictive radius of 2 μm. p-ERK staining was performed in all tumor sections simultaneously. For CDH1, p-ERK and Total ERK quantifications, we verified that staining was not affected in normal tissue of DEN/SOR treated *Notch1*⁻/⁻ mice compared to wild-type mice on tissues stained together on the same slide (*n* = 3 mice, Supplementary Table 24).

### DNA, RNA and scRNA-seq

Methods for sample processing and analysis of sequencing data (DNA sequencing, RNA bulk sequencing and scRNA-seq) are detailed in the Supplementary Note.

### Modeling

Stochastic simulations of clonal dynamics are explained in the Supplementary Note.

### Statistical analysis

Data are expressed as mean values ± s.e.m. unless otherwise stated. *P* values <0.05 were considered significant. Each experiment was performed using several biological replicates, with the exception of technical replicates only for primer validation using standard curves. The numbers of replicates are stated in the legends and in the Supplementary tables. Statistical tests are indicated in figure legends and were performed using GraphPad Prism software 8.3.1 and Python package Scipy 1.7.3 (https://scipy.org/citing-scipy/). No statistical method was used to predetermine sample size. Animals of the correct genotype were randomly assigned to experimental groups.

### Reporting summary

Further information on research design is available in the Nature Portfolio Reporting Summary linked to this article.

## Data availability

Accession numbers for the datasets are as follows: targeted sequencing of Human esophageal epithelium microbiopsies data is deposited in the European Genome-Phenome Archive under the accession code EGAD00001006969. All other sequences are deposited in the European Nucleotide Archive under the following accession codes: targeted sequencing of aged *Notch1*^+/−^ mouse esophageal epithelium

microbiopsies; ERP126992, targeted sequencing of mouse normal esophageal epithelium 28 weeks after DEN SOR treatment; ERP126993, targeted sequencing of mouse esophageal tumors 28 weeks after DEN SOR treatment; ERP126994, transcriptomic analysis of *Notch1* mutant esophageal epithelium; ERP126995, single-cell transcriptional analysis of *Notch1* mutant esophageal epithelium; ERP126996, transcriptomic analysis of *Notch1* mutant esophageal tumors and adjacent normal tissue 28 weeks after DEN SOR treatment; ERP137375.

All numerical data displayed in the figures are shown in Supplementary Tables 2–30.

Mouse strains are available from the Jax repository (https://www.jax.org), except the *Ahcre^ERT* line, which may be obtained by contacting the corresponding author. Source data are provided with this paper.

## Code availability

The codes developed in this study has been made publicly available and can be found at https://github.com/PHJonesGroup/Abby_etal_SI_code.

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

## Acknowledgements

This work is supported by grants from the Wellcome Trust to the Wellcome Sanger Institute (098051 and 296194) and Cancer Research UK Program Grants to P.H.J. (C609/A17257 and C609/A27326). E.A. benefited from the award of a Postdoctoral fellowship, 2016–23, from the Wellcome Sanger Institute. B.A.H. and M.W.J.H. are supported by the Medical Research Council (Grant-in-Aid to the MRC Cancer unit grant MC_UU_12022/9 and NIRG to B.A.H. grant MR/S000216/1). M.W.J.H. acknowledges support from the Harrison Watson Fund at Clare College, Cambridge. B.A.H. acknowledges support from the Royal Society (grant UF130039). S.C.D. benefited from the award of an ESPOD fellowship, 2018–21, from the Wellcome Sanger Institute and the European Bioinformatics Institute EMBL-EBI. G.P. is supported by the Agencia Estatal de Investigación of Spain (grant PID2020-116163GA-I00). A.H. benefited from the award of an EMBO long-term fellowship (EMBO ALTF885-2015) and a Maria Zambrano Grant to attract international talent from Universitat de Barcelona and Ministerio de Universidades and cofunded with Next Generation EU funds. We thank the following staff from the Wellcome Sanger Institute. E. Choolun, T. Metcalf and staff from the RSF facility for technical support with animal research; Y. Hooks, C. Hardy, C. Latimer and staff from the Cancer, Ageing, and Somatic Mutations program support laboratory and staff from the bespoke team for technical support with histology and sequencing.

## Author contributions

E.A., J.C.F, A.H. and P.H.J designed experiments. E.A. and J.C.F performed experiments. E.A., S.C.D, G.P., M.W.J.H, S.H.O and R.S. analyzed experimental and sequencing data. S.C.D. and M.G. created pipelines for copy number and scRNA-seq analyses. M.W.J.H. and B.A.H. performed clone simulations. I.A. and M.W.J.H. supervised statistical analyses. C.W.S. provided neutralizing antibodies. E.A, S.C.D, M.W.J.H. and P.H.J wrote the paper. M.G, B.A.H. and P.H.J supervised the research. All authors discussed the results and commented on the manuscript.

## Competing interests

The authors declare no competing interests.

## Additional information

**Extended data** is available for this paper at https://doi.org/10.1038/s41588-022-01280-z.

**Correspondence and requests for materials** should be addressed to Philip H. Jones.

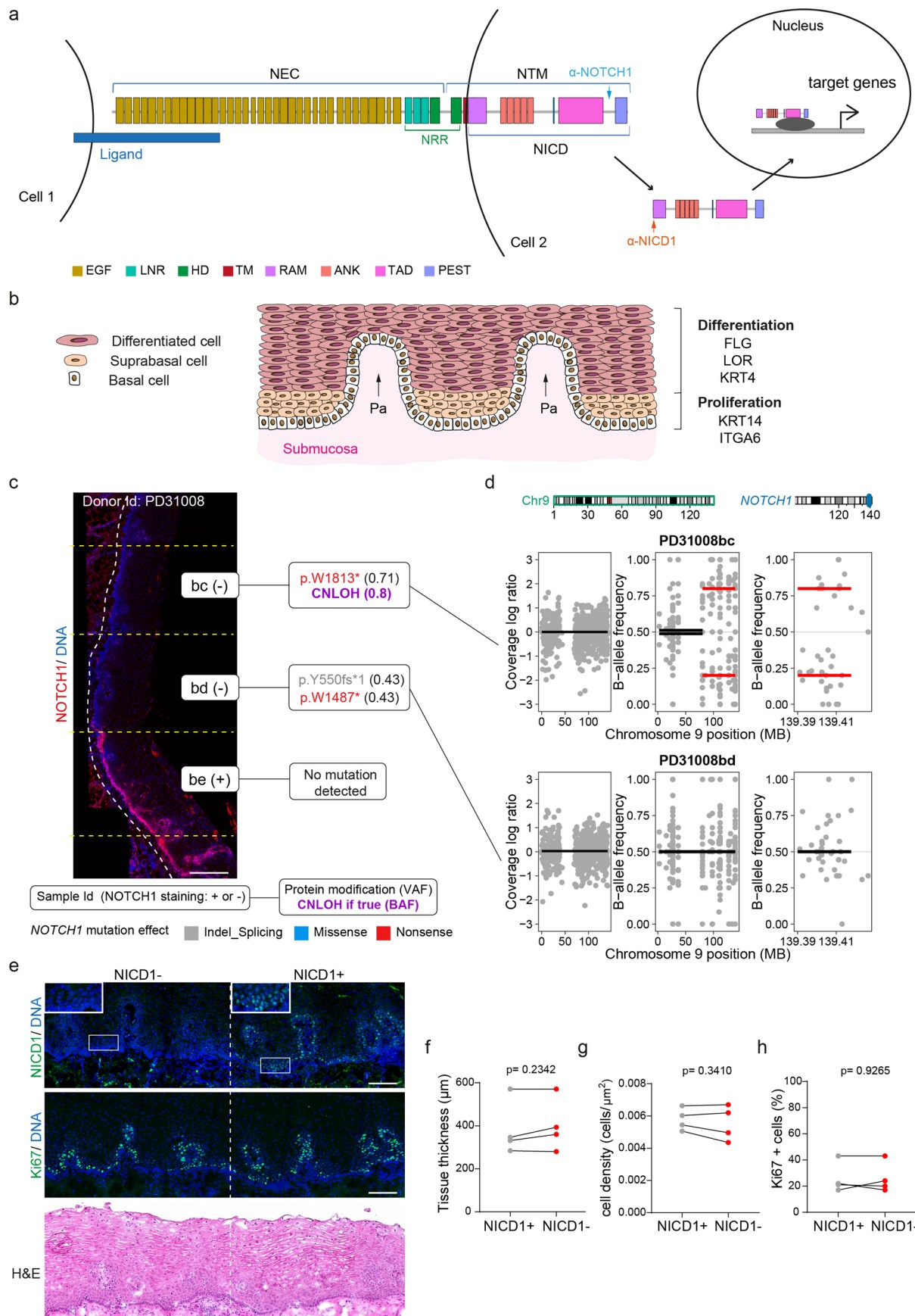

**Extended Data Fig. 1 | See next page for caption.**

# Article

**Extended Data Fig. 1 | Aging human esophageal epithelium is colonized by *NOTCH1* mutant clones. a**. NOTCH1 is composed of an extracellular domain (NEC) and a transmembrane and cytoplasmic unit (NTM). Domains of NOTCH1 are indicated, arrows show epitopes recognized by anti-NOTCH1 (blue) and anti-NICD1 (orange) antibodies. Ligand binding results in proteolytic cleavages, after which the intracellular domain (NICD) migrates to the nucleus and activates transcription. Domains: EGF, epidermal growth factor like repeats, LNR, Lin12/Notch repeats, HD, heterodimerization, TM, transmembrane, RAM, RBP-J associated module, ANK, ankyrin repeats, TAD, *trans*-activation domain, PEST, *rich in* proline, glutamate, serine, and threonine, NRR, negative regulatory region. **b**. Human esophageal epithelium. Proliferation is confined to the lower layers. Differentiating cells migrate to tissue surface. Pa, papillae. Protein expression shown on right. **c**. Representative section stained for NOTCH1 (red) and DNA (blue) showing subset of results in Fig. 1b for donor PD31008. Left: sample identification (Id), NOTCH1 staining status (+ or −). Right: non-synonymous *NOTCH1* mutations, variant allele frequency (VAF) and copy neutral loss of heterozygosity (CNLOH) if detected, with B allele frequency (BAF) value. Mutation effects: Indel_Splicing, gray; Missense, blue; Nonsense, red. Dashed lines delineate epithelium and submucosa (white) and borders of sequenced samples (yellow). Scale bar, 250 μm. **d**. Copy number calls for samples PD31008bc and bd, shown in **c**, Fig. 1d–f and Supplementary table 4. Left plot, analysis of total copy number along chromosome 9 for samples bc and bd. Middle, right plots, BAF along chromosome 9 and *NOTCH1* locus, respectively. Red lines denote significant difference from control, black lines indicate no significant difference. **e**. Successive sections of esophagus from older donors stained for NICD1, Ki67 and Hematoxylin and eosin (H&E). Images representative of 4 donors. Scale bars, 100 μm. **f, g, h**. Tissue thickness (**f**), cell density (**g**) and proportion of proliferative cells (**h**) in NICD1 positive and negative areas. Each dot represents a donor. For **f**, NICD1+, n = 14 areas from 4 donors, NICD1−, n = 15 areas from 4 donors. For **g**, NICD1+: 10795 cells from 4 donors, NICD1−: 11593 cells form 4 donors. For **h**, NICD1+: 5402 cells from 4 donors, NICD1−: 6204 cells from 4 donors. Two-tailed paired t-test. See Supplementary Tables 1–5.

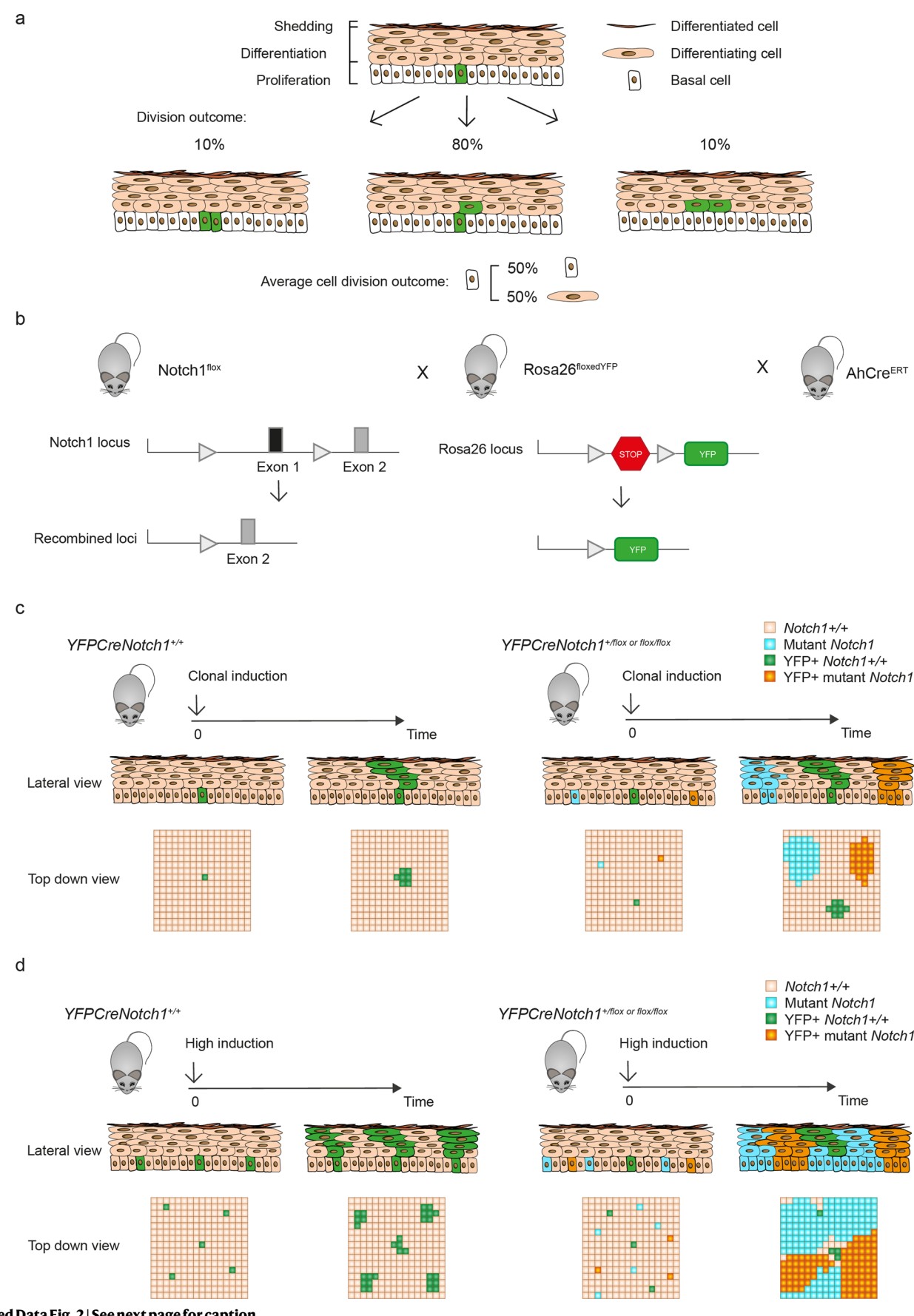

**Extended Data Fig. 2 | See next page for caption.**

**Extended Data Fig. 2 | Lineage tracing of *Notch1* mutant cells in mouse esophageal epithelium. a**. Structure and cellular homeostasis in mouse esophageal epithelium. The basal layer contains progenitor cells that divide to generate progenitor and differentiating daughter cells. Differentiating basal layer cells exit the cell cycle and migrate into the suprabasal layers, moving towards the surface of the epithelium from which they are shed. The division of a progenitor cell (green) produces two progenitors, two differentiating cells or one cell of each type. In homeostatic tissue, the likelihood of each division outcome is balanced and gives on average 50% of progenitors and 50% of differentiating cells across the progenitor population. **b**. *YFPCreNotch1* conditional knock-out mouse strain. *LoxP* sites (gray arrows) flank exon1 of the *Notch1* gene. *Notch1^flox^* animals were crossed with *Rosa26^floxedYFP^* mice carrying a conditional yellow fluorescent protein (YFP) reporter targeted to the *Rosa26* locus and with *AhCre^ERT^* mice carrying an inducible *Cre* recombinase. **c**. For lineage tracing, triple mutant mice were treated with inducing drugs at a dose that resulted in recombination of *Notch1 (blue)*, expression of YFP (green) or both (orange) in scattered individual esophageal basal cells (clonal induction). The recombined cells may expand into clones detected by the reduced intensity (+/−) or absence of NOTCH1 (−/−) and expression of YFP detected by immunostaining. Samples were collected at different time points after induction and the number and location of cells in each clone determined by 3D confocal imaging of sheets of epithelium. **d**. Triple mutant mice were induced with a high dose of drugs, allowing recombination of cells at high density in the tissue. In the case of mutant clones with a competitive advantage over wild type cells, this protocol allowed the coverage of the tissue by mutant clones relatively shortly after induction.

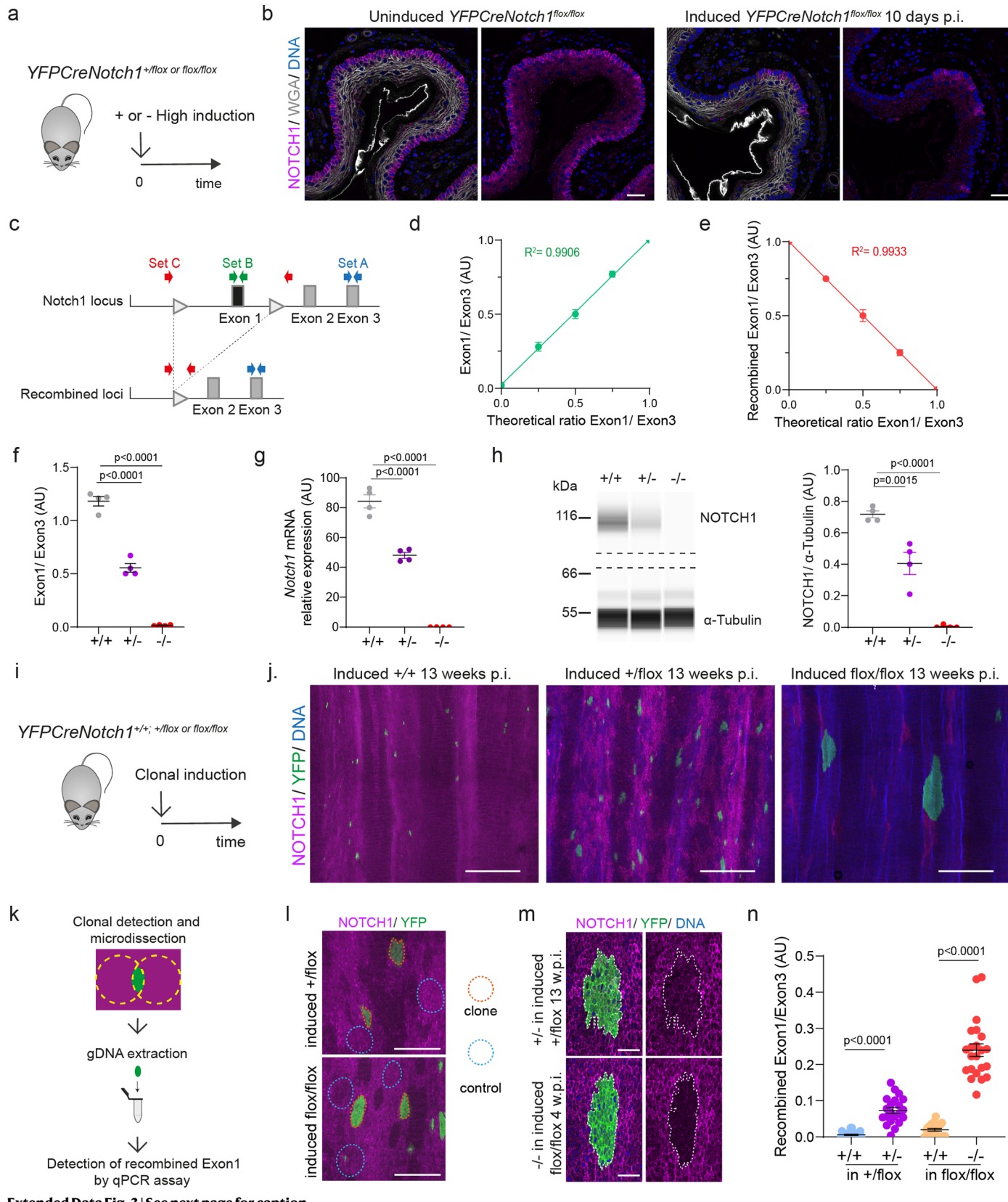

**Extended Data Fig. 3 | See next page for caption.**

**Extended Data Fig. 3 | Monoallelic and biallelic recombination at *Notch1* locus results in reduction of *Notch1* mRNA and protein. a**. Protocol for **b, f-h**. Highly induced *YFPCreNotch1*$^{flox/flox}$ or $^{+/flox}$ mice aged to allow *Notch1* mutant cells to colonize epithelium. Controls, non-induced mice (+/+). **b**. Esophageal sections 10 days post induction (p.i.) stained for NOTCH1 (magenta), Wheat germ agglutinin (WGA) (gray) and DNA (blue). Scale bars, 30 μm. **c**. Quantitative PCR assay for *Notch1* recombination. Primer set B measures floxed exon1, C amplifies recombined locus, A allows normalization. **d, e**. Validation using set B (**d**) and set C (**e**) against standard curve (Mean ± SEM, n = 3 technical replicates). **f-h**. *YFPCreNotch1*$^{flox/flox}$ or $^{+/flox}$ mice and controls aged for 8 weeks, mean ± SEM, each dot represents a mouse, n = 4 mice. **f**. Exon1/Exon3 ratio assay using set B. One-way ANOVA; adjusted p values from Tukey's multiple comparisons test against wild type. **g**. *Notch1:Gapdh* mRNA by RT-qPCR. One-way ANOVA; adjusted p values from Tukey's multiple comparisons test against wild type. **h**. Immune Capillary Electrophoresis of NOTCH1 transmembrane/intracellular domain (NTM1 + NICD1) and α-Tubulin protein. Dashed lines indicate image cropping and arrangement. One-way ANOVA; adjusted p values from Tukey's multiple comparisons test against wild type. **i**. Protocol. *YFPCreNotch1*$^{flox/flox}$ or $^{+/flox}$ or $^{+/+}$ mice were clonally induced, whole mounts stained for NOTCH1, YFP and DAPI. Clones were identified by NOTCH1 staining (Extended data Fig. 2a, Supplementary Note). **j**. Epithelium stained for NOTCH1 (magenta), YFP (green) and DNA (blue) 13 weeks p.i. of *YFPCreNotch1*$^{flox/flox}$ or $^{+/flox}$ or $^{+/+}$ mice. Scale bars, 500 μm. **k-n**. Validation of clonal genotype. **k**. Protocol. Tissues were stained for NOTCH1, YFP and DNA at 4 weeks p.i. for *YFPCreNotch1*$^{flox/flox}$ and 13 weeks p.i for *YFPCreNotch1*$^{+/flox}$. Potential clones were micro-dissected (yellow dotted lines) and qPCR performed using set C. **l, m**. Representative examples of 4 weeks post-induction (w.p.i.) *Notch1*$^{-/-}$ clones and control areas (upper panels) and 13 w.p.i. *Notch1*$^{+/-}$ clones and control areas (lower panels) validated as in **k**. NOTCH1, magenta, YFP, green, DNA, blue. **l**. Projected view. Dotted lines: orange, dissected clones, blue, control areas. Scale bars, 125 μm. **m**. (x, y) basal view. White dotted lines: clone edges. Scale bars, 30 μm. **n**. qPCR assay using primer set C (**c, e**). Mean ± SEM, each dot represents a sample, n = 22 controls and n = 21 +/− clones from 3 *YFPCreNotch1*$^{+/flox}$ mice; n = 21 controls and n = 22 −/− clones from 3 *YFPCreNotch1*$^{flox/flox}$ mice. Two-tailed unpaired Student t-test. AU, arbitrary units. SEM, standard error of mean. See Supplementary Table 6.

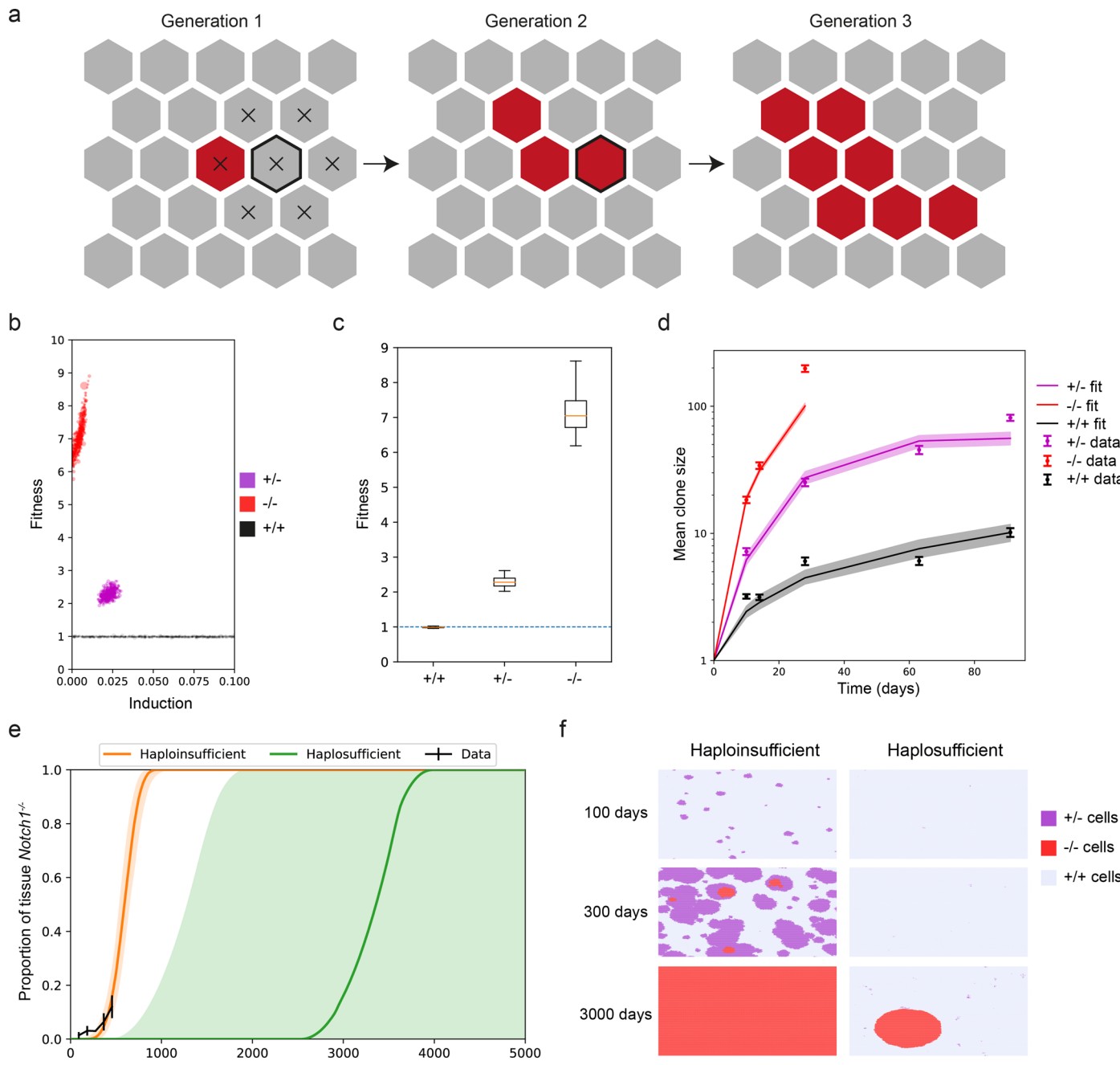

**Extended Data Fig. 4 | Modeling *Notch1* mutant clone expansion. a.**
2-dimensional Wright-Fisher style model of clone dynamics. The basal layer
consists of a hexagonal grid of cells. At time zero, a small proportion of cells is
mutant (red) and the rest wild type (gray). Cells in the next generation are picked
from neighboring cells – for example the cells which can be placed in the outlined
position in generation 2 are those marked with an X in generation 1. Mutant cells
with higher fitness have a higher probability of generating daughters in the next
generation and expand into large clones (Supplementary Note). **b.** Inferred
induction proportion and inferred fitness values from ABC fitting to the lineage
tracing data (Supplementary Note) for *Notch1⁻/⁻* (red), *Notch1⁺/⁻* (purple) and
*Notch1⁺/⁺* (black) clones in the respective animals. Each dot shows an 'accepted'
parameter set. A fitness of 1 (dotted line) is neutral. **c.** Distributions of acceptable
values of the fitness parameter. Whiskers show the upper and lower bounds of
the 95% credible interval, boxes show quartiles, center lines indicate medians of
credible intervals. A fitness of 1 (blue dotted line) is neutral. **d.** Mean clone sizes

from simulations of the parameters at the peak of acceptable distributions (see
Supplementary note). Median and 95% confidence intervals of 100 simulations
shown for the simulation curves. Mean ± standard error of mean are shown for
the experimental data. **e.** Proportion of tissue covered by *Notch1⁻/⁻* clones over
time in simulations using the best-fit for *Notch1⁻/⁻* fitness. *Notch1⁺/⁻* is either
assumed to be neutral (haplosufficient, green) or to have the best fitting fitness
parameter to the experimental analysis of *Notch1⁺/⁻* clones (haploinsufficient,
orange). Curves show median and shaded areas show 95% confidence intervals
of 100 simulations. **f.** Representative snapshot images at 100 days, 300 days
and 3000 days from the simulations shown in e. On the left, *Notch1⁺/⁻* cells are
haploinsufficient (fitting to experimental data), on the right the *Notch1⁺/⁻* cells
are assumed to be haplosufficient (neutral fitness). Cells from each genotype are
color coded. All images show the same number of cells/area of simulated tissue.
See Supplementary Note.

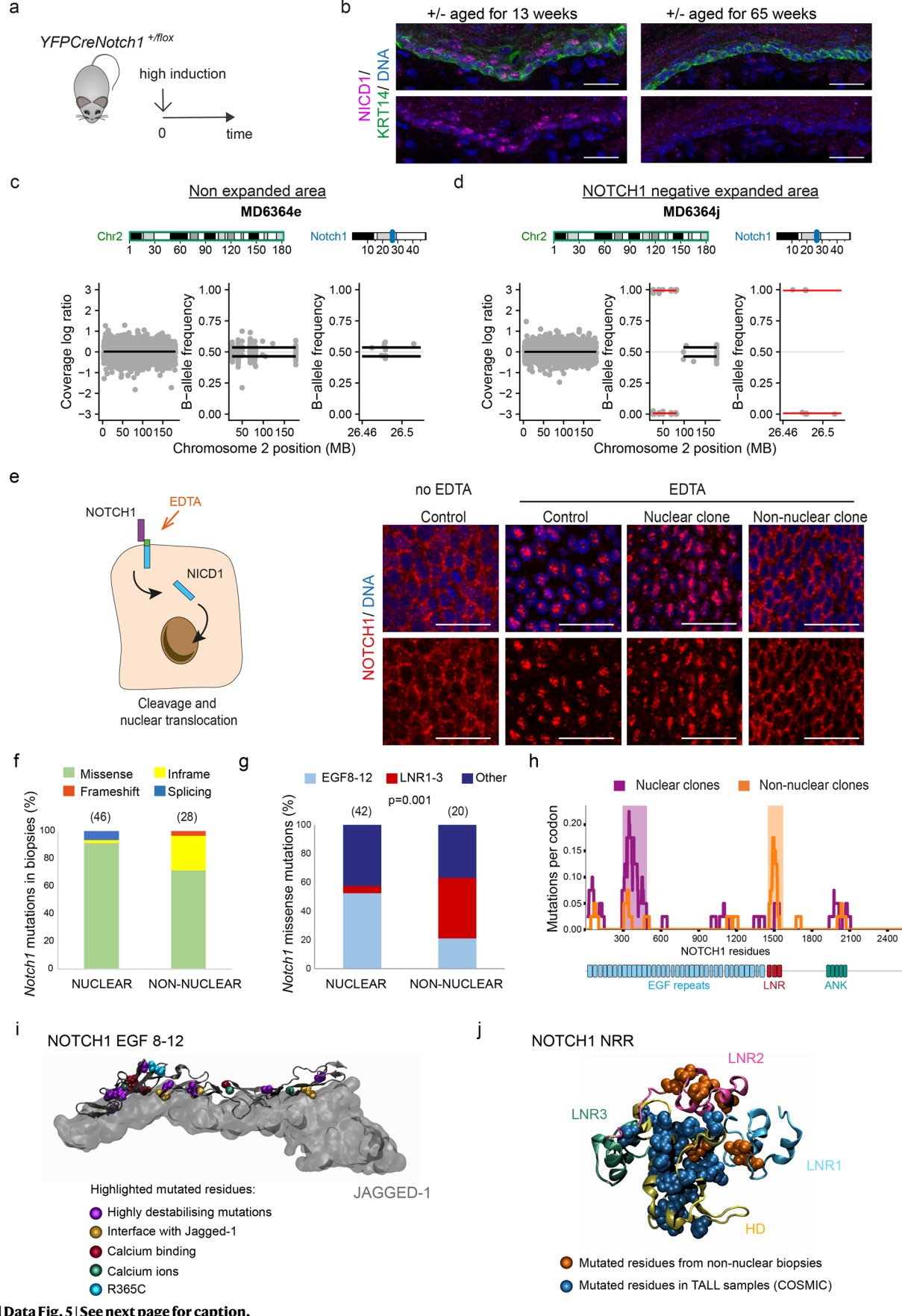

**Extended Data Fig. 5 | See next page for caption.**

**Extended Data Fig. 5 | Analysis of spontaneous mutant clones in *Notch1*+/−
aged esophageal epithelium. a**. Protocol. *YFPCreNotch1 +/flox* mice were induced
at high density and aged. **b**. Representative sections stained for NICD1 (magenta),
KRT14 (green), and DNA (blue) from *Notch1*+/− mice 13 weeks (n = 3) and 65 weeks
(n = 5) after induction. Scale bars, 25 μm. **c, d**. Copy neutral loss of heterozygosity
(CNLOH) analysis from Fig. 3e sequencing showing a representative non-clonal
area (**c**, sample MD6364e) and a NOTCH1 negative clone (**d**, sample MD6364j).
Coverage of off-target reads (left), and B allele fraction (BAF, middle and right)
along chromosome 2 (middle) and at *Notch1* locus (right). Red lines indicate
significant, black lines, no significant difference. **e**. Left: EDTA treatment
activates NOTCH1 cleavage without ligand. Right: representative NOTCH1
positive areas of aged *Notch1*+/− esophagi stained for NOTCH1 (red), YFP and
DNA (blue). Images show non-EDTA control and EDTA treated sequenced tissue
as in Fig. 3e–g (control area, nuclear or non-nuclear clones). Scale bars, 25 μm.
**f**. *Notch1* mutations in clones with or without NOTCH1 nuclear staining, n
mutations in brackets. Mutation effects are color coded. **g**. Location of missense
*Notch1* mutations in nuclear and non-nuclear staining clones, n samples in

brackets. P = 0.001, Chi-square test. **h**. Distribution of *Notch1* missense mutations
in nuclear (purple, n = 42) and non-nuclear staining (orange, n = 20) clones.
EGF, Epidermal Growth Factor like repeats, LNR, Lin12/Notch repeats and ANK,
ankyrin repeats shown. Purple shadow: EGF repeat 8-12 mutations, Orange
shadow: LNR repeat mutations. **i**. Missense mutations (n = 21) in NOTCH1 EGF8-12
in nuclear staining samples. Mutations highlighted on structure of rat NOTCH1
EGF8-12 bound to JAGGED-1 (PDB 5UK5, https://www.rcsb.org/). Mutated
residues: dark red, calcium binding; yellow, residues on interface with JAGGED-1;
purple, highly destabilizing mutations (FoldX ΔΔG > 2 kcal/mol). R365C, is blue,
calcium ions in green. **j**. Mutations shown on human negative regulatory region
(NRR) (PDB 3ETO, https://www.rcsb.org/). Orange, n = 9 missense mutations
from non-nuclear staining clones. Blue, missense mutations from human T-cell
acute lymphoblastic leukemia (https://cancer.sanger.ac.uk/cosmic ref. [27]).
Proportion of missense mutations between LNR1-2 and LNR3-HD in T-ALL and
non-nuclear staining clones is significantly different (Two-sided Fisher exact
tests, p = 1.48e−10 n = 153 mutations). LNR1, blue; LNR2, pink; LNR3, green; HD
domains, yellow. See Supplementary Tables 10 and 11.

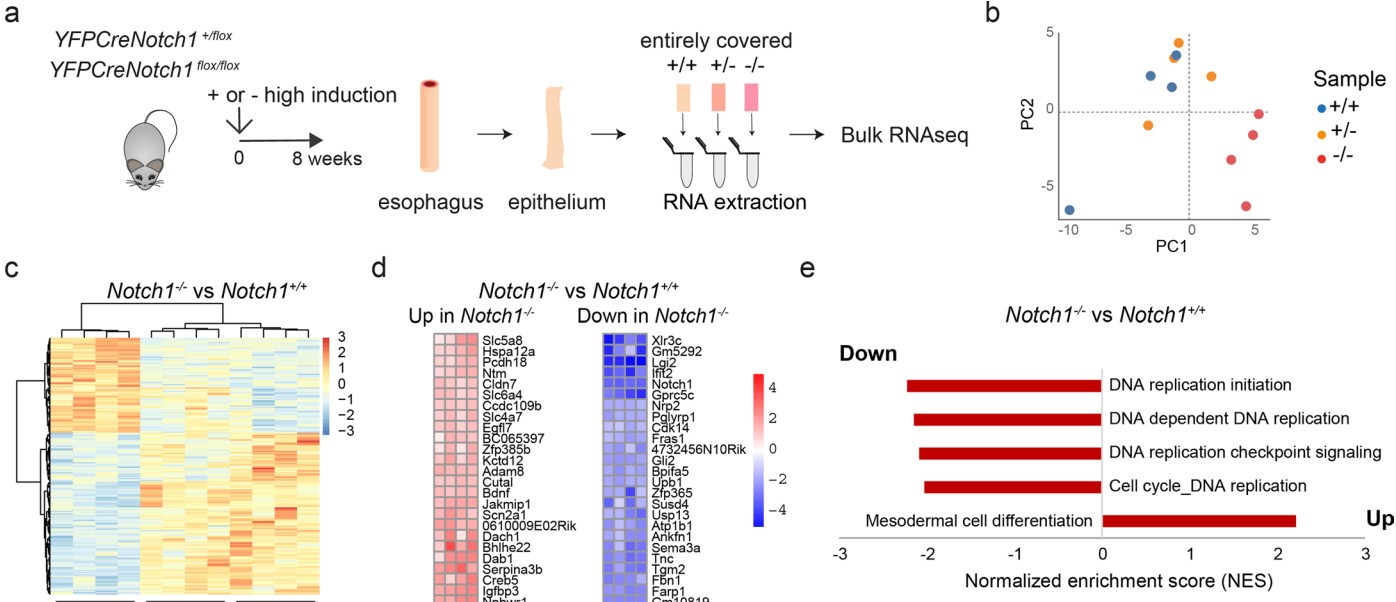

**Extended Data Fig. 6 | Notch1 loss alters transcription. a**. RNA-seq (n = 4 mice per group) was performed on epithelium from highly induced *YFPCreNotch1*^flox/flox^ and *YFPCreNotch1*^+/flox^ mice, aged for 8 weeks to allow mutant colonization. Uninduced mice were used as controls (+/+). **b**. Principal component analysis (PCA) plot showing *Notch1*^+/+^, *Notch1*^+/-^ and *Notch1*^-/-^ samples in two dimensions. Dotted lines indicate the origin of the axes. Sample genotypes are color-coded. **c**. Hierarchical clustering and heat map showing differentially expressed genes between *Notch1*^-/-^ and control tissues, in all three genotypes. **d**. Heat maps showing Log2 fold changes of 25 top differentially expressed genes in *Notch1*^-/-^ compared to *Notch1*^+/+^ tissues, adjusted p-value < 0.05. **e**. Gene Set Enrichment Analysis of *Notch1*^-/-^ tissue vs *Notch1*^+/+^ tissue. Bar chart shows normalized enrichment scores (NES) for the four most significantly downregulated gene sets in *Notch1*^-/-^ tissue vs *Notch1*^+/+^ tissue and the most significantly upregulated gene set. False discovery rate (FDR) q-value <0.05. See Supplementary Tables 12–15.

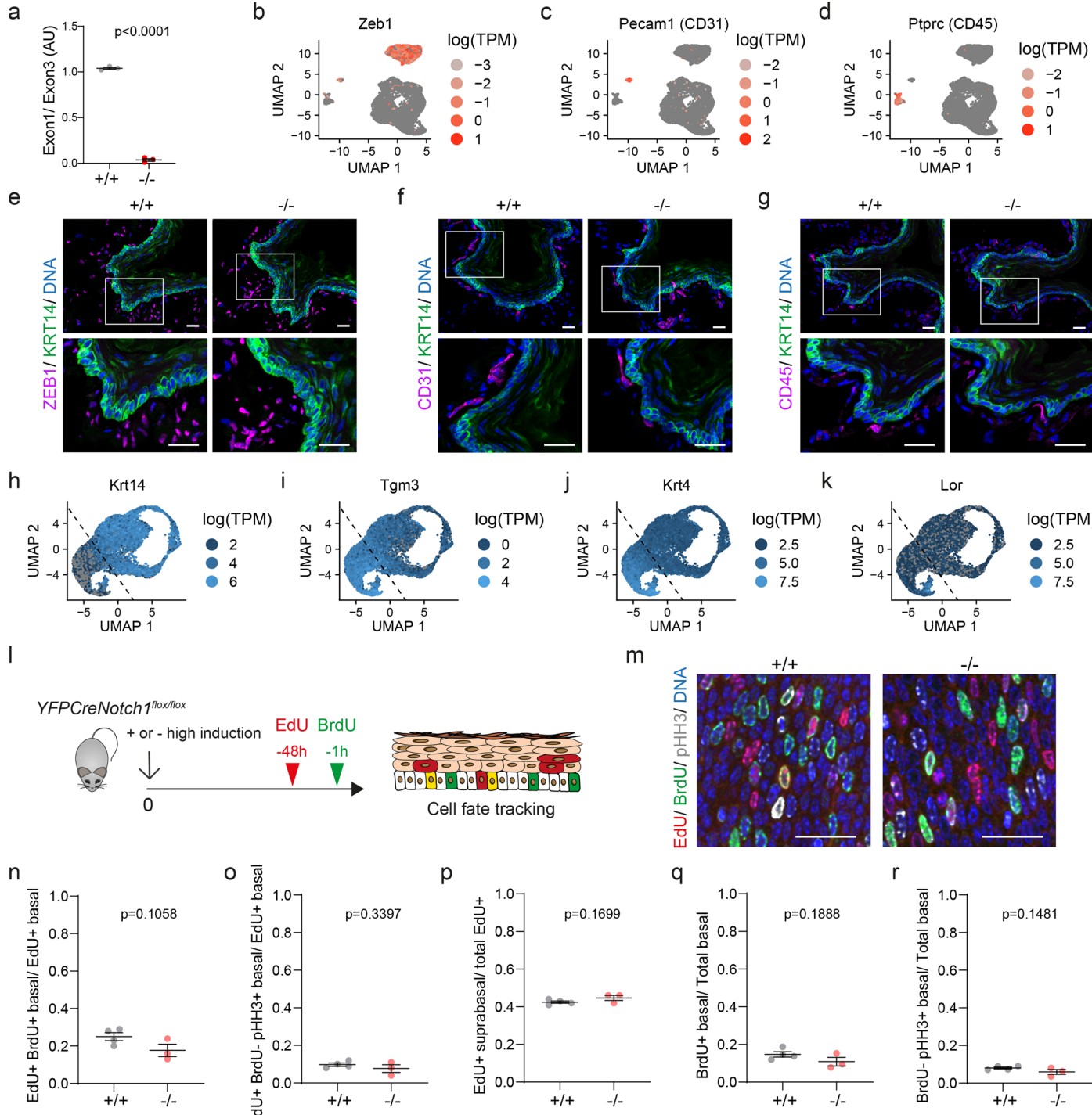

**Extended Data Fig. 7 | See next page for caption.**

**Extended Data Fig. 7 | *Notch1* loss does not alter tissue composition or cell dynamics. a.** qPCR recombination assay of *Notch1* exon1 in epithelium from *YFPCreNotch1^{flox/flox}* mice induced as for single cell RNA-seq (scRNA-seq)and collected 4 weeks later, compared with wild type tissue. Mean ± SEM, each dot represents a mouse, n = 3 mice. Two-tailed unpaired Student's t-test. **b-d.** scRNA-seq as in Fig. 4a (n = 2 mice per genotype; +/+1, n = 2454; +/+2, n = 3194; −/−1, n = 1929; −/−2, n = 5534). Uniform Manifold Approximation and Projection (UMAP) plots show markers of fibroblasts (*Zeb1*, **b**), endothelial cells (*Pecam1/* CD31, **c**) and immune cells (*Ptprc* /CD45, **d**). See Supplementary Note. **e-g.** Sections from *Notch1^{+/+}* and *Notch1^{−/−}* esophagi show KRT14 basal keratinocyte marker, green, DNA, blue and fibroblasts (ZEB1, **e**), endothelial cells (CD31, **f**) and immune cells (CD45, **g**), magenta. Images representative of 3 mice per genotype. Scale bars, 25 μm. **h-k.** UMAP plots show markers of keratinocyte differentiation highlighting basal cells (Krt14, **h**), differentiating cells (*Tgm3,* **i**, *Krt4,***j**) and cornified cells (*Lor*, **k**). Dashed black line separates basal cells and suprabasal cells (Supplementary Note). (+/+1, n = 1555; +/+2, n = 1932; −/−1,

n = 1403; −/−2, n = 3919). **l.** *YFPCreNotch1^{flox/flox}* mice were highly induced or not induced (+/+ controls) and aged for 8 weeks, then EdU was injected 48 h and BrdU 1 h before collection. EdU+ cells are shown in red; BrdU+ cells are green; EdU+; BrdU+ cells are yellow. **m.** *Notch1^{−/−}* and *Notch1^{+/+}* epithelia stained for EdU (red), BrdU (S phase, green), pHH3 (G2/M, gray) and DNA (blue). Scale bars, 25 μm. **n, o.** Ratio of EdU + ; BrdU+ basal cells (S phase, n) or EdU+; BrdU-; pHH3 + (G2/M, o)/total EdU+ basal cells in *Notch1^{+/+}* and *Notch1^{−/−}* epithelia (n = 3856 *Notch1^{+/+}* EdU+ basal cells from 4 mice, n = 2328 *Notch1^{−/−}* basal cells from 3 mice). **p.** EdU+ suprabasal: total EdU+ cells ratio (n = 6696 EdU+ *Notch1^{+/+}* cells from 4 mice; n = 4203 EdU+ *Notch1^{−/−}* cells from 3 mice). **q, r.** BrdU+ basal cells (S phase, **q**) or BrdU−; pHH3+ basal cells (G2/M, **r**) /total basal cells in *Notch1^{+/+}* and *Notch1^{−/−}* epithelia (n = 22669 *Notch1^{+/+}* basal cells from 4 mice, n = 16111 *Notch1^{−/−}* basal cells from 3 mice). For **n-o**, Mean ± SEM, each dot represents a mouse, two-tailed unpaired Student's t-tests. AU, arbitrary unit. SEM, standard error of mean. See Supplementary Tables 16 and 17.

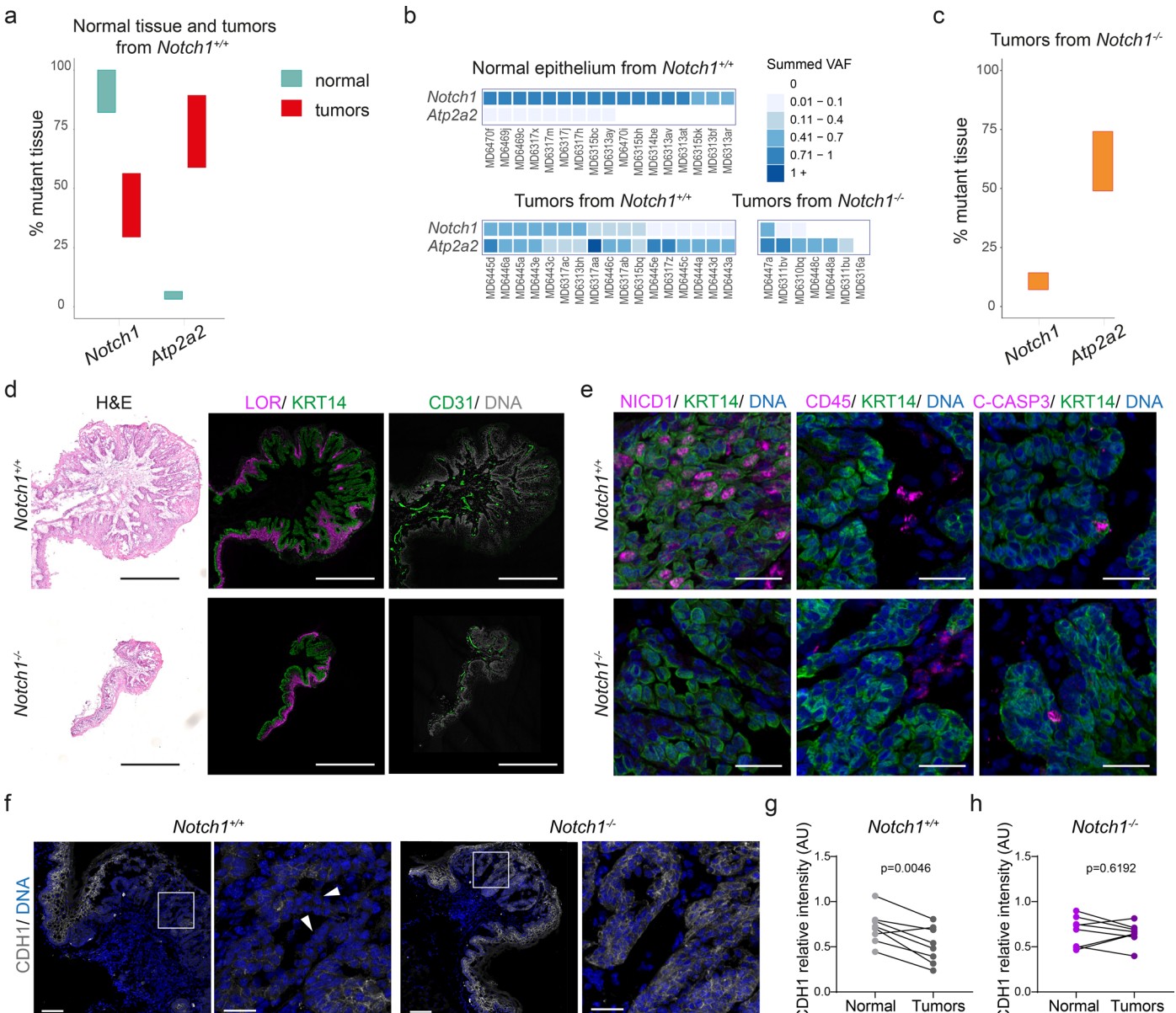

**Extended Data Fig. 8 | Characterization of mouse esophageal tumors.**
**a**–**c**. Mouse esophageal epithelium and tumors from *Notch1*[+/+] esophagus and tumors from *Notch1*[−/−] esophagus (from protocol in Fig.7a) were processed for targeted sequencing. **a**. Proportion of tissue mutant for *Notch1* and *Atp2a2* in tumors from *Notch1*[+/+] esophagus and in adjacent epithelium, estimated from sum of variant allele frequencies (VAFs) of non-synonymous mutations for *Notch1* and *Atp2a2* (n = 115 normal epithelial biopsies from 6 mice, n = 17 tumors from 7 mice; Supplementary Note). **b**. Summed VAF of protein altering mutations in the indicated genes is shown for random *Notch1* +/+ epithelial samples (upper, n = 17/115), all 17 sequenced tumors from *Notch1*[+/+] (lower left) and all 7 tumors from *Notch1*[−/−] esophagus (lower right). **c**. Proportion of tissue mutant for *Notch1* and *Atp2a2* estimated from sum of VAFs of non-synonymous mutations for each gene in tumors from *Notch1*[−/−] esophagus. (n = 7 tumors from 5 mice; Supplementary table 26; Supplementary Note). **d-h**, Protocol. Tumors from *Notch1*[+/+] and *Notch1*[−/−] esophageal epithelia (Fig. 7a) were sectioned

and characterized by immunostaining. **d**. Tumors were stained, from left to right, for Hematoxylin and eosin (H&E), differentiation markers (KRT14 and LOR), endothelial marker CD31 and DNA, gray. **e**. Tumors were stained, from left to right, for active NOTCH1 (NICD1), immune cell marker CD45, apoptosis marker cleaved Caspase 3 in magenta, KRT14, green and DNA, blue. For **d**, **e**, images representative of n = 10 wild type tumors, n = 11 *Notch1*[−/−] tumors for all immunostainings, n = 4 for cleaved Caspase 3. Scale bars in **d**, 500 μm, in **e**, 30 μm. **f**. Tumors were stained for E-cadherin (CDH1, gray) and DNA (blue). Right panels, magnified views of white squares. Arrows indicate keratinocytes with reduced CDH1 staining. Images are representative of n = 8 tumors from 6 mice of each genotype. Scale bars, 100 μm (left panel) and 30 μm (right). **g,h**. Mean CDH1 intensity relative to DNA in tumor compared to adjacent epithelium in *Notch1*[+/+] (**g**) and *Notch1*[−/−] (**h**) (n = 8 tumors from 6 mice for each genotype). Two tailed paired Student's t-test. AU, arbitrary unit. **See** Supplementary Tables 19, 20, 24 and 26.

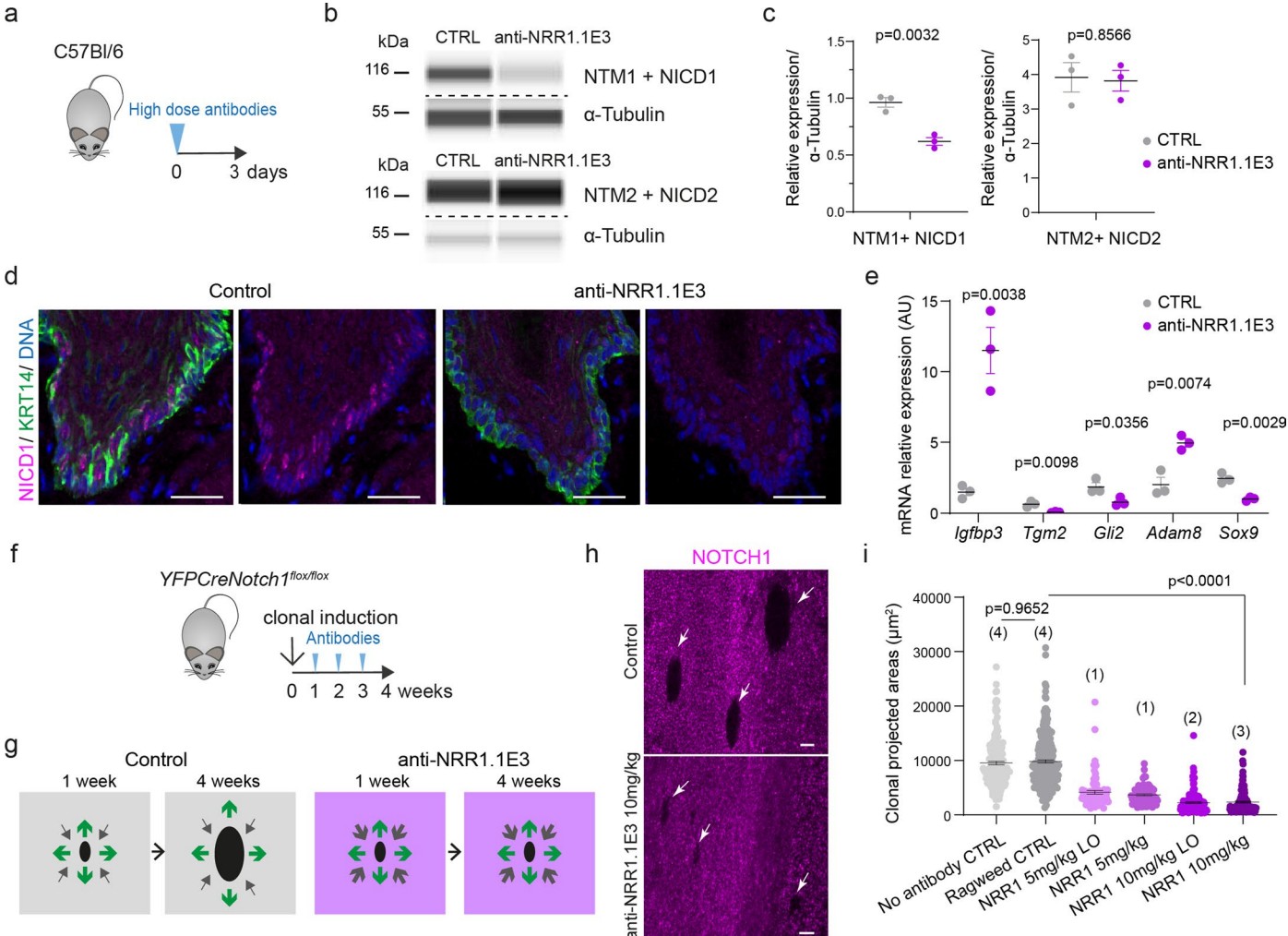

**Extended Data Fig. 9 | Use of anti-NOTCH1 antibody to inhibit NOTCH1 signaling *in vivo*. a.** Protocol for b-e. C57Bl/6 wild type mice were treated with anti-NRR1.1E3 or control antibody (CTRL) for three days before tissue collection. **b, c.** Immune Capillary Electrophoresis was performed on peeled esophageal epithelia of mice in a. Visual representation of cleaved transmembrane and intracellular regions of NOTCH1 (NTM1 + NICD1, top panel, Extended data Fig. 1a) and of NOTCH2 (NTM2 + NICD2, bottom panel), and α-Tubulin proteins. Dashed lines indicate image cropping (**b**). Proteins expression relative α-Tubulin. Mean ± SEM, each dot represents a mouse, n = 3 mice. Two-tailed unpaired Student's t-test (**c**). **d.** Representative images of staining for NICD1 (magenta), KRT14 (green) and nuclei (blue) in sectioned epithelium of 3 mice treated with control or anti-NRR1.1E3 antibodies. Scale bars, 25 μm. **e.** RT-qPCR for markers of *Notch1* loss of function identified by bulk RNA-seq analysis (Extended data

Fig. 6d, Supplementary Table 12) relative to *Gapdh* transcript in control and anti-NRR1.1E3 treated samples. Mean ± SEM, each dot represents a mouse, n = 3. Two-tailed unpaired Student's t-test. **f.** Protocol. *YFPCreNotch1^{flox/flox}* mice were induced at clonal density. One week later, mice were treated with anti-NRR1.1E3 or control antibodies for 3 weeks. **g.** Principle of assay shown in **f**, If anti-NRR1.1E3 treatment blocks NOTCH1 signaling, all cells have equal fitness and expansion of *Notch1^{−/−}* clones is halted[21]. **h.** Representative images of NOTCH1 negative clones in EDTA peeled esophageal epithelia treated with control or anti-NRR1.1E3 antibody from 3 mice. Scale bars, 50 μm. **i.** Projected area of clones negative for NOTCH1 staining. Mean ± SEM, each dot represents a clone. Number of mice analyzed is in brackets. One-way ANOVA; Tukey's multiple comparisons test, adjusted p-values versus control antibody. AU, arbitrary unit. SEM, standard error of mean. LO, loading. **See**, Supplementary table 25.

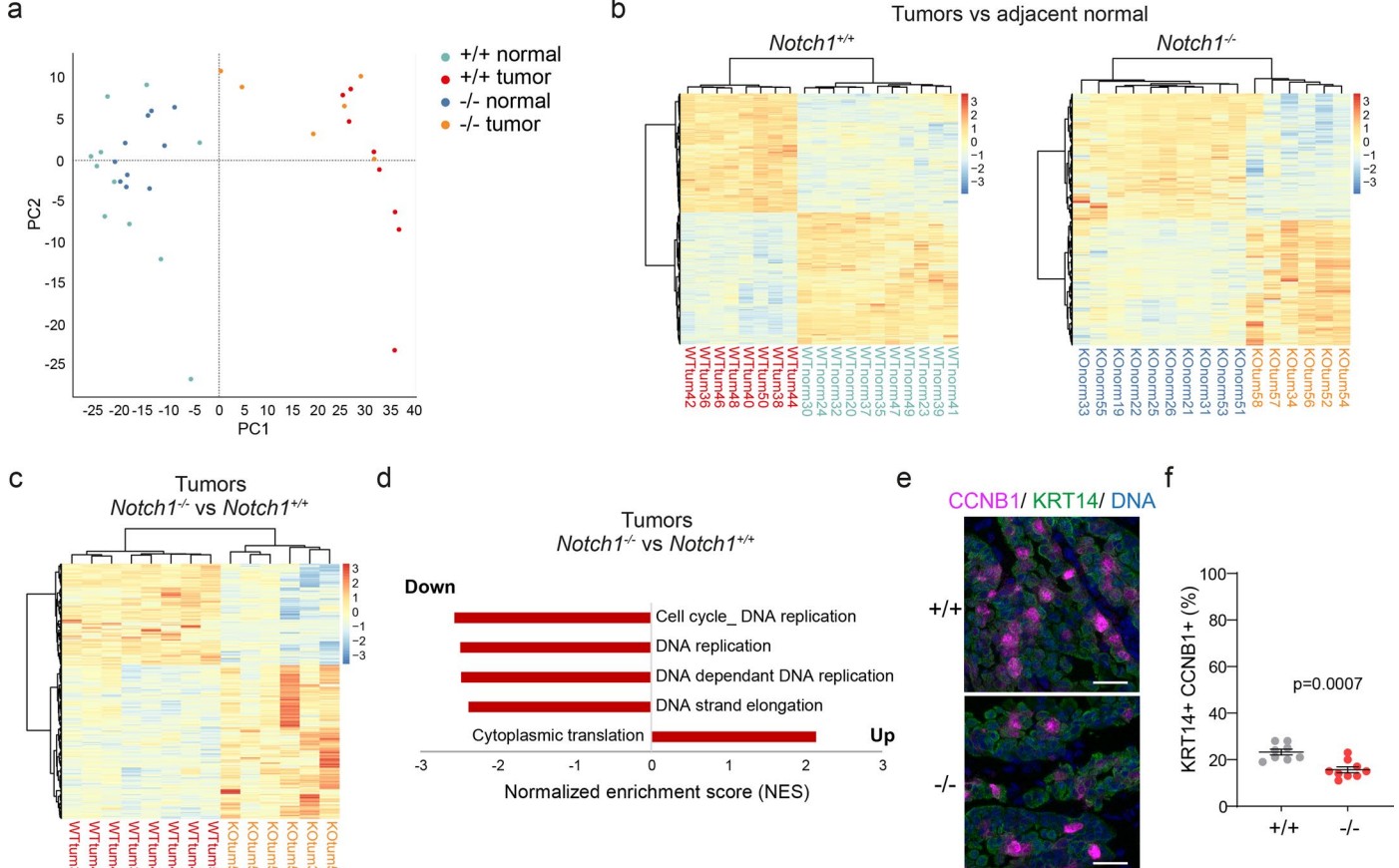

**Extended Data Fig. 10 | Transcriptomic characterization and cellular phenotype of tumors from *Notch1*+/+ and *Notch1*–/– esophagus. a-d.** RNA-seq analysis of *Notch1*+/+ and *Notch1*–/– esophageal tissue and tumors, see Fig. 8a. **a**. Principal component analysis (PCA) plot showing in two dimensions all biological replicates from *Notch1*+/+ normal epithelium (n = 11 biopsies from 7 mice, green), tumors from *Notch1*+/+ epithelium (n = 8 tumors from 4 mice, red), *Notch1*–/– normal epithelium (n = 10 biopsies from 7 mice, blue) and tumors from *Notch1*–/– epithelium (n = 6 tumors from 5 mice, orange). Dotted lines indicate the origin of the axes. **b**. Hierarchical clustering and heat map showing differentially expressed genes between tumors and adjacent normal tissue, in *Notch1*+/+ mice (left) and *Notch1*–/– mice (right). **c**. Hierarchical clustering and heat map showing differentially expressed genes between tumors from *Notch1*+/+ and *Notch1*–/– esophagus. **d**. Gene Set Enrichment Analysis (GSEA) of tumors from *Notch1*–/– vs *Notch1*+/+ esophagus. Normalized enrichment score (NES) of altered Biological Process gene sets in tumors. False discovery rate, FDR q-value<0.05. **e**. Representative images of n = 8 tumors from *Notch1*+/+ and n = 9 tumors from *Notch1*–/– esophagus. KRT14 (green), CCNB1 (magenta). DNA, blue. Scale bars, 30 μm. **f**. Percentage of CCNB1 positive; KRT14 expressing keratinocytes within tumors from *Notch1*+/+ and *Notch1*–/– esophagus. Mean ± SEM, each dot represents a tumor, +/+: n = 8 tumors from 4 mice; –/–: n = 9 tumors from 7 mice. Two-tailed unpaired Student's t-test. See Supplementary Tables 24 and 27–29.

# nature research

| | |
|---|---|

# Reporting Summary

Nature Research wishes to improve the reproducibility of the work that we publish. This form provides structure for consistency and transparency in reporting. For further information on Nature Research policies, see our Editorial Policies and the Editorial Policy Checklist.

## Statistics

For all statistical analyses, confirm that the following items are present in the figure legend, table legend, main text, or Methods section.

| n/a | Confirmed | |
|---|---|---|
| ☐ | ☒ | The exact sample size ($n$) for each experimental group/condition, given as a discrete number and unit of measurement |
| ☐ | ☒ | A statement on whether measurements were taken from distinct samples or whether the same sample was measured repeatedly |
| ☐ | ☒ | The statistical test(s) used AND whether they are one- or two-sided<br>*Only common tests should be described solely by name; describe more complex techniques in the Methods section.* |
| ☐ | ☒ | A description of all covariates tested |
| ☐ | ☒ | A description of any assumptions or corrections, such as tests of normality and adjustment for multiple comparisons |
| ☐ | ☒ | A full description of the statistical parameters including central tendency (e.g. means) or other basic estimates (e.g. regression coefficient) AND variation (e.g. standard deviation) or associated estimates of uncertainty (e.g. confidence intervals) |
| ☐ | ☒ | For null hypothesis testing, the test statistic (e.g. $F$, $t$, $r$) with confidence intervals, effect sizes, degrees of freedom and $P$ value noted<br>*Give P values as exact values whenever suitable.* |
| ☒ | ☐ | For Bayesian analysis, information on the choice of priors and Markov chain Monte Carlo settings |
| ☒ | ☐ | For hierarchical and complex designs, identification of the appropriate level for tests and full reporting of outcomes |
| ☒ | ☐ | Estimates of effect sizes (e.g. Cohen's $d$, Pearson's $r$), indicating how they were calculated |

*Our web collection on statistics for biologists contains articles on many of the points above.*

## Software and code

Policy information about availability of computer code

| | |
|---|---|
| Data collection | Confocal images were obtained using acquisition software Leica Application Suite X (LAS X). Confocal Z stack images were rendered and analyzed with Volocity 6 Software (Perkin Elmer). Hematoxylin and eosin (H&E) stained tissues were rendered and analyzed with NanoZoomer Digital Pathology software (NDP.view2, Hamamatsu). Immune capillary electrophoresis was performed and analyzed using Compass for SW version 4.1.0. qPCR data were obtained using StepOne Software v2.3. |
| Data analysis | Confocal images were analyzed using Volocity 6 Software (Perkin Elmer). Hematoxylin and eosin (H&E) stained tissues were analyzed with NanoZoomer Digital Pathology software (NDP.view2, Hamamatsu). Immune capillary electrophoresis was analyzed using Compass for SW version 4.1.0. qPCR data were analyzed using StepOne Software v2.3. GraphPad Prism 8.3.1. was used for data plotting and statistical analysis. Python package Scipy 1.7.3 (https://scipy.org/citing-scipy/) was also used for statistics. For DNA sequencing, paired-end reads were aligned with BWA-MEM (v.0.7.17, https://github.com/lh3/bwa) with optical and PCR duplicates marked using Biobambam2 (v.2.0.86, https://gitlab.com/german.tischler/biobambam2). For clonal datasets, substitution mutations were called using the CaVEMan (Cancer Variants through Expectation Maximization, version 1.13.14) variant caller (http://cancerit.github.io/CaVEMan). Insertions and deletions were called using cgpPindel (http://cancerit.github.io/cgpPindel, version 3.3.0). Mutations were annotated using VAGrENT (https://github.com/cancerit/VAGrENT, version 3.3.3). For subclonal calling in highly mutagenized tissue, we used ShearwaterML algorithm from the deepSNV package (v1.21.3, https://github.com/gerstung-lab/deepSNV). We used the maximum-likelihood implementation of the dNdScv algorithm (v0.0.1.0, https://github.com/im3sanger/dndscv) to identify genes under positive selection.<br>For RNAseq, the alignment files were sorted and duplicate-marked using Biobambam2 2.0.54, and the read summarization performed by the htseq-count script from version 0.6.1p1 of the HTSeq framework (Anders et al., 2015; Dobin et al., 2013 ). Differential gene expression was analyzed using the DESeq2 R package version 1.20.0 (Love et al., 2014 ), and the downstream pathway analysis and visualization using R version 3.5.3 (https://www.R-project.org/) and the packages Pheatmap version 1.0.12 (https://cran.r-project.org/package=pheatmap), RColorBrewer version 1.1.2 (https://cran.r-project.org/package=RColorBrewer), clusterProfiler version 3.8.1 (Yu et al., 2012) and org.Mm.eg.db version 3.6.0 (https://bioconductor.org/packages/org.Mm.eg.db/). For Gene Set Enrichment Analysis (GSEA), we used GO BP or KEGG gene sets v7.5.1 from the Molecular Signature Database (MSigDB) in GSEA software v4.2.3. (Subramanian et al. 2005). The functions of the DEGs from DESeq2 analysis were annotated using the database for annotation, visualization, and integrated discovery (DAVID v6.8) |

(Huang da et al., 2009).
For sc-RNAseq, alignment of the sequencing reads and expression quantification was performed for each library individually using the CellRanger pipeline version 3.0.2 (10xGenomics). We subsequently used EmptyDrops version 1.2.2 to detect empty droplets in the raw feature count matrix output from CellRanger and discarded any barcode identified as an empty droplet. All the subsequent analysis described was performed in R version 4.1.3 (https://www.R-project.org/ ) using the Seurat software package version 4.0.3.
Custom codes for clone simulations, for copy number analysis and sc-RNAseq analyses are available at https://github.com/PHJonesGroup/ Abby_etal_SI_code

For manuscripts utilizing custom algorithms or software that are central to the research but not yet described in published literature, software must be made available to editors and reviewers. We strongly encourage code deposition in a community repository (e.g. GitHub). See the Nature Research guidelines for submitting code & software for further information.

## Data

Policy information about availability of data

All manuscripts must include a data availability statement. This statement should provide the following information, where applicable:
- Accession codes, unique identifiers, or web links for publicly available datasets
- A list of figures that have associated raw data
- A description of any restrictions on data availability

Accession numbers for the datasets are as follows:-Targeted sequencing of Human esophageal epithelium microbiopsies (EGA): EGAD00001006969. -Targeted sequencing of aged Notch1 +/- mouse esophageal epithelium microbiopsies (ENA): ERP126992-Targeted sequencing of mouse normal esophageal epithelium 28 weeks after DEN SOR treatment (ENA): ERP126993 -Targeted sequencing of mouse esophageal tumours 28 weeks after DEN SOR treatment (ENA): ERP126994-Transcriptomic analysis of Notch1 mutant esophageal epithelium (ENA): ERP126995 -Single cell transcriptional analysis of Notch1 mutant esophageal epithelium (ENA): ERP126996.- Transcriptomic analysis of Notch1 mutant esophageal tumors and adjacent normal tissue 28 weeks after DEN SOR treatment (ENA): ERP137375.
All data displayed in the figures are available in Supplementary tables 2-30.
The codes developed in this study has been made publicly available and can be found at https://github.com/PHJonesGroup/Abby_etal_SI_code

# Field-specific reporting

Please select the one below that is the best fit for your research. If you are not sure, read the appropriate sections before making your selection.

☒ Life sciences    ☐ Behavioural & social sciences    ☐ Ecological, evolutionary & environmental sciences

For a reference copy of the document with all sections, see nature.com/documents/nr-reporting-summary-flat.pdf

# Life sciences study design

All studies must disclose on these points even when the disclosure is negative.

| | |
|---|---|
| Sample size | Sample size was not predetermined by statistical methods. Sample size was determined by pilot studies for lineage tracing and clonal sequencing studies and by previously published studies for highly mutagenized sequencing and carcinogenesis studies. |
| Data exclusions | Data exclusion was only made in RNA sequencing studies to improve the quality of the datasets. Quality control of RNA-seq study: 'Transcriptomic analysis of Notch1 mutant esophageal tumors and adjacent normal tissue 28 weeks after DEN SOR treatment ' revealed one outlier sample on PCA plot out of the initial 36 samples dataset. The outlier sample was removed from the analysis. Quality control of RNA-seq study: 'Transcriptomic analysis of Notch1 mutant esophageal epithelium' revealed an outlier control sample on PCA plot but deep analysis of the noise identified 151 genes with aberrant signal within the control group. Thorough checks with complete analysis were performed with and without these genes and/ or the sample, leading to the conclusion that none of these actions modified the conclusions of the analysis but excluding these genes and not the affected control sample for final analysis preserved the most data and resulted in removing 7 false positive hits. For 'Single cell transcriptional analysis of Notch1 mutant esophageal epithelium', poor quality cells were filtered out as is considered best practice. Details are provided in respective Methods and Supplementary Note. |
| Replication | For Human histological study, three donors were analyzed per age group (young, middle-aged and elderly). For Human sequencing, we analyzed multiple biopsies from 6 distinct donors. For mouse studies, experiments were performed with at least 3 mice per time point constituting 3 independent biological replicates except on two occasions.  scRNA-seq was performed on two biological replicates per genotype as each tissue yielded sequencing data from thousands of cells. Findings were further verified in separate experiments (Immunostaining and EdU/BrdU lineage tracing in young mice; Immunostaining and EdU in aged mice)  involving 3 to 4 biological replicates. Pilot neutralizing antibody titration involved some dosing that were not repeated but the final dosage was confirmed with 3 biological replicates. All attempts at replicating the findings from the study were successful. |
| Randomization | For mouse experiments, mice of relevant genotype were randomly assigned to each experimental protocol. For Human study, donors were randomly assigned for sequencing/ histological analysis based on their age at tissue collection. |
| Blinding | Blinding was performed but its feasibility was sometimes limited. Technicians and investigators were blinded to group allocation during mice treatments, except when performing treatments that required such information (high inductions, antibody treatments). Samples were systematically given identification numbers so that investigators were blinded to genotype or treatment during processing and analysis when this was applicable. Blinding was not applicable or effective when the information of material were required for analysis or when the experiments required sampling using immunostaining that reflected the genotype. |

# Reporting for specific materials, systems and methods

We require information from authors about some types of materials, experimental systems and methods used in many studies. Here, indicate whether each material, system or method listed is relevant to your study. If you are not sure if a list item applies to your research, read the appropriate section before selecting a response.

## Materials & experimental systems

| n/a | Involved in the study |
|-----|----------------------|
| ☐ | ☒ Antibodies |
| ☒ | ☐ Eukaryotic cell lines |
| ☒ | ☐ Palaeontology and archaeology |
| ☐ | ☒ Animals and other organisms |
| ☐ | ☒ Human research participants |
| ☒ | ☐ Clinical data |
| ☒ | ☐ Dual use research of concern |

## Methods

| n/a | Involved in the study |
|-----|----------------------|
| ☒ | ☐ ChIP-seq |
| ☒ | ☐ Flow cytometry |
| ☒ | ☐ MRI-based neuroimaging |

## Antibodies

**Antibodies used**

Unconjugated primary antibodies
Protein (clone) species and clonality Company Reference Dilution
KRT14 chicken polyclonal Biolegend 906001 1/1000
KRT14 rabbit polyclonal Covance PRB-155P 1/1000
Ki67 (SP6) rabbit monoclonal Abcam ab16667 1/500
KRT6a ( Poly19057) rabbit polyclonal Biolegend PRB-169P 1/1000
Ki67 (MIB1) mouse monoclonal Agilent M724029-2 1/500
NOTCH1 (D1E11) rabbit monoclonal Cell Signaling Technology 3608 1/200-1/1000
GFP chicken polyclonal Invitrogen A10262 1/500
Loricrin (AF 62) rabbit polyclonal Covance PRB-145P 1/2000
Keratin 4 (6B10) mouse monoclonal Vector Labs VP-C399 1/1000
NICD1 (Val1744; D3B8) rabbit monoclonal Cell Signaling Technology4147 1/100
E-Cadherin (24E10) rabbit monoclonal Cell Signaling Technology 3195 1/500
CD45 (30-F11 ) rat monoclonal Biolegend 103102 1/200
CD31 ( (MEC7.4) rat monoclonal Abcam ab7388 1/200
cleaved caspase 3 rabbit polyclonal Abcam ab2302 1/200
BrdU [BU1/75 (ICR1)] rat monoclonal Abcam ab6326 1/250
CCNB1 rabbit polyclonal Cell Signaling Technology4138 1/200
ZEB1 (E2G6Y) rabbit monoclonal Cell Signaling Technology70512 1/500
Phospho-Erk1/2 -Thr202/Tyr204 (D13.14.4E) rabbit monoclonal Cell Signaling Technology 4370 1/200
Erk1/2 (137F5) rabbit monoclonal Cell Signaling Technology 4695 1/200

Conjugated primary antibodies
Protein/fluorophore Conjugation method Company Reference Dilution
ITGA6-Alexa 647 Company Biolegend 313610 1/200
Phalloidine-Alexa 488 Company Invitrogen A12379 1/200
WGA-Alexa 647 Company Invitrogen W32466 1/500
K6a-Alexa 555 Thermo Fisher labeling kit (A20187) Biolegend PRB-169P 1/500
Histone H3 -phospho S10- Alexa 647 Alexa Fluor® 647 (Company) Abcam ab196698 1/10000

Seconday antibodies
Host/Fluorophore Target Company Reference Dilution
Donkey Alexa-555 anti-Rabbit Invitrogen A31572 1/500
Donkey Alexa-488 anti-rabbit Invitrogen A21206 1/500
Donkey Alexa-647 anti-rabbit Invitrogen A31573 1/500
Donkey Alexa-488 anti-chicken Jackson 703-545-155 1/250
Donkey Alexa-488 anti-mouse Invitrogen A21202 1/500
Goat Alexa-555 anti-rat Invitrogen A21434 1/500
Donkey Alexa-647 anti-mouse Invitrogen A31571 1/500
Donkey Alexa-488 anti-rat Invitrogen A21208 1/500

**Validation**

NOTCH1 (D1E11) rabbit monoclonal was validated for immunofluorescence, immunohistochemistry and Immune Capillary Electrophoresis in our study using a knock-out NOTCH1 mouse model. Furthermore, the protein conservation between Human and Mouse is very high and immunofluorescence and immunohistochemistry assays in Human tissues revealed areas showing both expression and absence of expression within the same tissues, further confirmed being mutant areas by DNA sequencing. NICD1 (Val1744; D3B8) rabbit monoclonal antibody is a highly cited antibody (https://www.cellsignal.co.uk/products/primary-antibodies/cleaved-notch1-val1744-d3b8-rabbit-mab/4147) and we also validated it both in immunofluorescence and Immune Capillary Electrophoresis in our study using a knock-out mouse model. BrdU [BU1/75 (ICR1)] is a highly cited rat monoclonal antibody, suitable for immunofluorescence (https://www.abcam.com/brdu-antibody-bu175-icr1-proliferation-marker-ab6326.html). Alexa

Fluor® 647 Anti-Histone H3 (phospho S10) antibody [mAbcam 14955] (ab196698) is the conjugated version of highly cited mouse monoclonal ab14955 (https://www.abcam.com/histone-h3-phospho-s10-antibody-mabcam-14955-ab14955.html). ZEB1 (E2G6Y) #70512 rabbit monoclonal antibody  is validated in immunofluorescence by Cell Signaling (https://www.cellsignal.com/products/primary-antibodies/zeb1-e2g6y-xp-rabbit-mab/70512). Cyclin B1 Antibody #4138 is a highly cited and validated antibody from Cell Signaling (https://www.cellsignal.com/products/primary-antibodies/cyclin-b1-antibody/4138). Phospho-Erk1/2 #4370 is  a highly cited and validated antibody from Cell signaling (https://www.cellsignal.com/products/primary-antibodies/phospho-p44-42-mapk-erk1-2-thr202-tyr204-d13-14-4e-xp-rabbit-mab/4370).   Total Erk1/2 #4695 is a highly cited and validated antibody from Cell Signaling (https://www.cellsignal.com/products/primary-antibodies/p44-42-mapk-erk1-2-137f5-rabbit-mab/4695). All other primary antibodies were used for immunofluorescence, validation is described in PMID: 17330052, PMID: 22821983, PMID: 24814514 and PMID: 27548914.

## Animals and other organisms

Policy information about studies involving animals; ARRIVE guidelines recommended for reporting animal research

| | |
|---|---|
| Laboratory animals | Laboratory animals were mice from C57BL/6J background or from a transgenic mixed C57BL/6J and 129X1/SvJ background (YFPCreNotch1). Strains used were as indicated for each experiments:  Rosa26floxedYFPAhCreERTNotch1flox (YFPCreNotch1 with Notch1 genotype status precised for each experiment) and C57Bl/6. Mouse housing was carried out in individually ventilated cages (19-23°C, RH55%±10%, 12/12 light dark cycle, 15-20 air changes per hour). Mice were fed on standard chow. Mice were maintained on a specific and opportunistic pathogen free health status and were immune competent. Animals were not involved in any previous experiments. Both male and female adult mice at 10-16 weeks of age at the start of the experiments were used. |
| Wild animals | The study did not involve wild animals |
| Field-collected samples | The study did not involve samples collected from the field |
| Ethics oversight | UK government Home Office project licences, which include stringent local and government ethical review. UK Home Office Project Licenses 70/7543, P14FED054 or PF4639B40. |

Note that full information on the approval of the study protocol must also be provided in the manuscript.

## Human research participants

Policy information about studies involving human research participants

| | |
|---|---|
| Population characteristics | 4 female and 6 male Human donors, aged 20 to 78 years old. |
| Recruitment | Esophageal tissue was obtained from deceased organ donors from whom organs were being retrieved for transplantation. Informed consent was obtained from next of kin. Consecutive cases were recruited. There was no self selection bias. The sample is likely to be representative of organ transplant donor population in Eastern England. |
| Ethics oversight | Informed consent for the use of tissue was obtained from the donor's relatives (REC reference: 15/EE/0152 NRES Committee East of England - Cambridge South). |

Note that full information on the approval of the study protocol must also be provided in the manuscript.

