## [Peer Review File · Nature Genetics]

Peer Review Information

Manuscript Title: Notch1 mutation drives clonal expansion in normal esophageal epithelium but impairs tumor growth

Corresponding author name(s): Professor Philip Jones

Reviewer Comments & Decisions:

Decision Letter, initial version:

12th May 2021

Dear Professor Jones,

Your Article, "Notch1 mutation drives clonal expansion in normal esophageal epithelium but impairs tumor growth" has now been seen by 4 referees. Please note that Reviewers #2 and #3 reviewed the work together and have uploaded the same report.

You will see from their comments copied below that while they find your work of considerable potential interest, they have raised quite substantial concerns that must be addressed. In light of these comments, we cannot accept the manuscript for publication as is, but would be very interested in considering a revised version that addresses these concerns.

We hope you will find the referees' comments useful as you decide how to proceed. If you wish to submit a substantially revised manuscript, please bear in mind that we will be reluctant to approach the referees again in the absence of major revisions.

To guide the scope of the revisions, the editors discuss the referee reports in detail within the team, including with the chief editor, with a view to identifying key priorities that should be addressed in revision and sometimes overruling referee requests that are deemed beyond the scope of the current study. We hope that you will find the prioritised set of referee points to be useful when revising your study. Please do not hesitate to get in touch if you would like to discuss these issues further.

Overall, the reviewers agree that you're tackling an intriguing question using an elegant experimental approach and, fortunately, no serious technical issues are raised. From our reading of the reports, we think that the main critique is around the lack of a satisfying mechanism to explain how Notch1 mutations increase fitness yet restrain tumour growth. Moreover, Reviewer #1 would also like you to reconcile these data with the fact that Notch1 mutations have been identified in oesophageal cancers. We agree that this mechanistic insight is important, and will be required for our further consideration. How you choose to address this experimentally is up to you, you'll see that your reviewers have made some suggestions, but you are welcome to consider other approaches.

The other reviewer comments should also be addressed in full, either experimentally (where appropriate/possible) or textually.

If you choose to revise your manuscript taking into account all reviewer and editor comments, please highlight all changes in the manuscript text file. At this stage we will need you to upload a copy of the manuscript in MS Word .docx or similar editable format.

*2) If you have not done so already please begin to revise your manuscript so that it conforms to our Article format instructions, available [here](http://www.nature.com/ng/authors/article_types/index.html). Refer also to any guidelines provided in this letter.

[redacted]

If you wish to submit a suitably revised manuscript we would hope to receive it within 6 months. If you cannot send it within this time, please let us know. We will be happy to consider your revision so long as nothing similar has been accepted for publication at Nature Genetics or published elsewhere. Should your manuscript be substantially delayed without notifying us in advance and your article is eventually published, the received date would be that of the revised, not the original, version.

Thank you for the opportunity to review your work.

Sincerely,

Safia Danovi
Editor
Nature Genetics

Referee expertise:

Referee #1: oesophageal cancer, somatic mutations

Referee #2 (reviewed with Reviewer #3): somatic mutations, lineage tracing

Referee #3 (reviewed with Reviewer #2): somatic mutations, lineage tracing

Reviewer #4: Notch signalling

Reviewers' Comments:

Reviewer #1:

Remarks to the Author:

The paper by Emilie Abby, Phil Jones and co-authors aims to understand the role of somatic Notch1 LoF mutations in normal oesophagus and in the development of oesophageal cancer.

Notch1 is a known cancer gene in several cancer types, including oesophageal cancer. However recently it has been found mutated at higher frequency in adult oesophagus. Understanding the role of Notch1 mutations in normal oesophagus and in tumourigenesis is therefore a hot topic in the field.

In the first part of the study the authors very elegantly show how heterozygous and then homozygous mutations in Notch1 give the cell higher fitness in both mouse and human. Notch1 mutant cells are progressively able to colonise the whole oesophagus, to then revert back to wild type behaviour, so that the oesophagus remains histologically normal. What remains to be understood is why do Notch1 mutant cells have higher fitness? No significant differences emerge from the transcriptional profile thus suggesting that these cells are phenotypically similar. Strangely, the authors do not think of any possible role played by non-epithelial cells and/or cell non-autonomous mechanisms that could impact differently on the fitness of Notch1 mut and wt cells. In such a detailed study, this is a missing opportunity.

In the second part of the study, the authors aim to understand the role of Notch mutations in tumourigenesis and their conclusions are that Notch1 loss impairs tumour growth. This part of the study is in my opinion less robust and does not explain a big paradox: Notch inactivating mutations are observed and frequent in ESCC (~10% of TCGA ESCC bear clonal truncating mutations). How can mutations that impair tumour growth be selected for in cancer cells? This evidence is completely neglected by the authors although it in principle invalidates their hypothesis.

General comments

p.2: the roles of Notch1 in cancer are not 'controversial': as other cancer genes (TP53, XXX) can act as a tumour suppressor or as an oncogene, depending on the context. The first 2 references cited here (1,2) do not support either role: they report higher frequency of mutations of Notch1 in normal oesophagus. These references should be removed from this paragraph.

p.3: why has Kendall's tau-b test been preferred to Spearman correlation to assess the association bw Notch1 mutations and age?

Then they investigate how NOTCH1 mutant clones colonise normal epithelium in mouse and notice progressively larger sizes of Notch1^{+/-} and Notch1^{-/-} clones. However they do not observe different rates of cell division of basal cells but a lower rate of cell division in the suprabasal cells in the mutant clones

Figure 2m,n: shouldn't a lower proliferation of suprabasal cells be expected in ^{-/-} vs ^{+/-} since ^{-/-} cells have higher fitness and their clones reach larger size than ^{+/-} cells? This was a surprising result that needs further investigation. Probably a direct comparison between the two conditions in the same setting will shed some light.

Figure 3b,e,f: I am not entirely clear why areas with missense mutations whose VAF is >0.2 are negative for Notch1 staining? Shouldn't the mutated allele still be able to produce some protein, even though not functional? Are these mutations all able to alter the binding of the antibody used for staining (although this seems unlikely). This requires further investigation. I would tend to disagree with the conclusion of the paragraph (Overall the effect of these mutations on NOTCH1 expression is consistent with our observations in human tissue) because in human the absence of Notch staining was associated with complete loss of both alleles.

p.17: "This observation associated with the absence of detectable nuclear NOTCH1 after EDTA treatment in this group of clones suggests that the mutations they carry at the NRR domain may lock it into its auto-inhibited conformation, preventing the cleavage of NOTCH". This needs to be proven. There may be other effect that the mutations are inducing, other than locking the protein into an auto-inhibited conformation

p.20: where are the simulation of neutral mutations? How do the authors assess neutrality?

p.3: "negative staining areas in the two oldest donors carried nonsense or essential splice mutations or indels in NOTCH1 with copy neutral loss of heterozygosity (CNLOH) of the NOTCH1 locus (human GRCh37- chr9:139,388,896-139,440,238) or a further mutation, likely to disrupt the second NOTCH1 allele 1." This is a result obtained in this study, reference to Martincorena et al is inappropriate.

“NOTCH1 protein and NICD1, which is detectable in the nucleus during active signalling” this needs to be referenced as it is based on previous knowledge

p.25: “NOTCH1 mutations are under strong positive selection in normal human esophagus and occupy 30% to 80% of the epithelium as early as middle-age but are only found in about 10% of esophageal squamous cell carcinomas (ESCC) 1,3.” Ref.2 should be cited here

Reviewer #2:

Remarks to the Author:

Comments to the Authors:

This manuscript elegantly describes the conflicting action of Notch1 mutation in normal and tumorigenic esophageal epithelium. In aging human esophagus, sequencing reveals that Notch1 mutant clones which occupy a significant proportion of the epithelium carry biallelic alterations. To test the fitness impacts of Notch mutant clones, the authors developed a conditional transgenic mouse where one or both alleles of Notch could be deleted and induce a fluorescent lineage reporter. The authors used lineage tracing and Notch staining in this transgenic mouse to quantify the fitness advantage that the loss of one allele confers over wild type epithelium. This fitness advantage was further magnified by the loss of the second allele. Sequencing of mutant clones revealed slight differences in transcription. However, the authors reported no functional difference in the morphological analysis of the esophageal epithelium. Finally, the authors induced tumors in this mouse model and found that Notch mutants were less numerous in tumors compared to the normal adjacent epithelium. Complete loss of Notch1 and antibody blockade of Notch1 resulted in significantly smaller tumors, but no change in tumor number. The authors are to be commended for an exhaustive analysis of Notch1 mutations in human patient specimens and their application of an informative mouse model for studying Notch1 clonal competition and its effects on tumorigenesis. The summary is clear and overall the manuscript is well written. An appropriate review of the literature is included. However, the major question in the study is to determine how Notch1^{-/-} clones spread throughout the esophagus yet are rarely found in tumors. Interestingly, the group's 2020 Nature Genetics paper describes a very elegant model in which cell fitness is relative to neighboring clones (as cartooned in Figure 6f, 2020). From this perspective Notch1 mutations could inhibit tumorigenesis through two likely mechanisms, Notch1^{-/-} could intrinsically inhibit tumorigenic potential or Notch1^{-/-} could extrinsically inhibit tumorigenic potential. The current manuscript seems to suggest an extrinsic mechanism whereby increased cell fitness of Notch1^{-/-} cells prevents the growth of tumors. This is a very attractive hypothesis and extremely exciting. If proven, it would be a significant contribution to the field and warrant reconsideration. This concern is the most important to address and listed as priority 1 below. Several additional concerns are also provided – that are likely more easily rectified through textual revision and/or additional analysis/reanalysis of available data.

Major Experimental Concerns

1. The current data/experiments do not address the paradoxical abundance of human Notch1^{+/+} esophageal cancers despite the field/clonal expansion of Notch1^{-/-} in aged epithelium. The tumor modeling using the high induction model seems to test the intrinsic hypothesis because most cells will be Notch1^{-/-} and receive secondary hits in the same cell type. However, the extrinsic hypothesis is left untested. Extrinsic forces of Notch1^{-/-} super-fit cells could inhibit neighboring mutant cells and reduce deterministic tumor outcomes and similar again to previous competition experiments in the esophagus (<https://www.nature.com/articles/s41588-020-0624-3>). This is a very interesting concept, particularly given the Adaptive Oncogenesis model proposed by DeGregori or Winton's competition measurements in the colon.

A very important experiment will be to induce infrequent clonal Notch1^{-/-} mutations, then chemically incite tumors. This would determine whether tumor incidence decreases in a mosaic epithelium with Notch^{+/+}, ^{+/-}, and ^{-/-} clones already present. In this manner, Notch^{-/-} clones would be placed in direct competition with tumorigenic clones – a more naturalistic model of the human condition. This would also enable the direct demonstration that somatic mutations capable of driving clonal expansions could act as tumor suppressors.

2. The overall enthusiasm for the human data set (figure 1) is somewhat tempered by the limited number of patients in the study – albeit the concern is somewhat mitigated by the inclusion of several technical replicates. The authors are to be commended for the attempt at coregistering the imaging data with genomics mutation data. However, the presentation in 1c is very difficult to follow and poorly labelled (i.e. x axis). It is also difficult to draw conclusions without knowledge of intrapatient and interpatient variability. Can conclusive images for high or low notch be magnified and then “coregistered” in panel c with appropriate callouts/highlights etc. Most interesting would be an intrapatient analysis of the same section, with clear annotations of high/low notch correlating with VAF/CNLOH.

3. In figure 2, three recombination events are shown: Mutant Notch1, YFP+ Mutant Notch1, and YFP Notch^{+/+}. What is the frequency of these events where recombination of only YFP or only Notch occur? Previously these single recombination events have been reported to be highly differential depending on the transgene (PMID: 24355609). Interpreting the majority of the lineage tracing data is predicated on being able to identify NOTCH^{+/+}, Notch^{+/-}, and Notch^{-/-}. It is unclear from the current data if IF alone is a reasonable solution. Additional in situ proof is likely necessary (mRNA FISH etc.).

4. Similar to 2 above, overall basal patch size is provided in figure 2 (basal cell number per clone) and similar patches provided in figure 3c. For the clonal experiments it would very important to show

whether or not overall clone number changes and how competition proceeds over time (i.e. +/+, +/-, -/- clone numbers over time and clone sizes over time).

5. The EdU experiments (instantaneous labelling – 1hr and pulse-chasing – 48 hrs) are important for understanding how clones grow larger in the Notch1^{-/-} epithelium. Given that scRNASeq did not reveal changes in the cell number (see below if more analysis is warranted). If there are no differences in proliferation then there must be differences in the rate of differentiation, thus the importance of the 48 hour chase. The 48 hours timepoint is poorly described. I think the data support the conclusion that the reduction in EdU positive suprabasal ^{-/-} is indicative of less differentiation and/or the increase in suprabasal ^{+/+} is indicative of forced differentiation to provide “space” for ^{-/-} cells. If this is the case, then do the scRNAseq data support this conclusion? (Fig 2 “l, j, m, n” could all use appropriate figure headings). Please consider clarifying the description of the data and the presentation of the data.

6. Figure 4 genomics data is not overly helpful and could be moved to the supplement. The scRNAseq is really underutilized, particularly understated is the potential for difference in cell states between the genotypes. Was a pseudotime trajectory analysis performed / could it be useful to understand how homeostasis is maintained in Notch1 ^{-/-} ? Does this data support the overall lineage training / EdU analysis conclusions?

Minor Comments

VAF1 and VAF2 are not well defined, how do these relate to one another? Which mutation is being analyzed?

In Extended Data 3g, the text remarks that Notch^{+/-} could be distinguished from Notch^{-/-} based on intensity of the signal. An inset demonstrating the clear difference in intensity would be helpful to boost the confidence in these results.

The imaging is rather difficult to interpret due to the size of the presented images, consider revising accordingly.

In the Tamoxifen experiments for Figure 5 Notch^{flox/flox} mice were injected with a quarter of the dose of the Notch^{+/+} and Notch^{+/-} mice. This lower induction rate is then compared to a higher induction rate to demonstrate no differences in tissue morphology and function.

The genetic nomenclature for the mice is very difficult to follow and sometimes not uniform throughout the legends.

Figure 7i denotes 16 weeks of antibody treatment, but the text reports 6 weeks of antibody treatment.

The manuscript is well-written but has a few Grammatical errors:

- ED Figure 3a legend: allowiNotch1
- ED Figure 3f: non uniform italics used
- No callout for Figure 4b in text
- Page 31: "short-term analysis the revealed"
- "recombination of in..." poor grammar p7
- Figure 2b, add figure label for Notch mRNA
- "excision of p 9"

Reviewer #3:

Remarks to the Author:

Comments to the Authors:

This manuscript elegantly describes the conflicting action of Notch1 mutation in normal and tumorigenic esophageal epithelium. In aging human esophagus, sequencing reveals that Notch1 mutant clones which occupy a significant proportion of the epithelium carry biallelic alterations. To test the fitness impacts of Notch mutant clones, the authors developed a conditional transgenic mouse where one or both alleles of Notch could be deleted and induce a fluorescent lineage reporter. The authors used lineage tracing and Notch staining in this transgenic mouse to quantify the fitness advantage that the loss of one allele confers over wild type epithelium. This fitness advantage was further magnified by the loss of the second allele. Sequencing of mutant clones revealed slight differences in transcription. However, the authors reported no functional difference in the morphological analysis of the esophageal epithelium. Finally, the authors induced tumors in this mouse model and found that Notch mutants were less numerous in tumors compared to the normal adjacent epithelium. Complete loss of Notch1 and antibody blockade of Notch1 resulted in significantly smaller tumors, but no change in tumor number. The authors are to be commended for an exhaustive analysis of Notch1 mutations in human patient specimens and their application of an informative mouse model for studying Notch1 clonal competition and its effects on tumorigenesis. The summary is clear and overall the manuscript is well written. An appropriate review of the literature is included. However, the major question in the study is to determine how Notch1^{-/-} clones spread throughout the esophagus yet are rarely found in tumors. Interestingly, the group's 2020 Nature Genetics paper describes a very elegant model in which cell fitness is relative to neighboring clones (as cartooned in Figure 6f, 2020). From this perspective Notch1 mutations could inhibit tumorigenesis through two likely mechanisms, Notch1^{-/-} could intrinsically inhibit tumorigenic potential or Notch1^{-/-} could extrinsically inhibit tumorigenic potential. The current manuscript seems to suggest an extrinsic mechanism whereby increased cell fitness of Notch1^{-/-} cells prevents the growth of tumors. This is a very attractive hypothesis and extremely exciting. If proven, it would be a significant contribution to the field and warrant reconsideration. This concern is the most

important to address and listed as priority 1 below. Several additional concerns are also provided – that are likely more easily rectified through textual revision and/or additional analysis/reanalysis of available data.

Major Experimental Concerns

1. The current data/experiments do not address the paradoxical abundance of human Notch1^{+/+} esophageal cancers despite the field/clonal expansion of Notch1^{-/-} in aged epithelium. The tumor modeling using the high induction model seems to test the intrinsic hypothesis because most cells will be Notch1^{-/-} and receive secondary hits in the same cell type. However, the extrinsic hypothesis is left untested. Extrinsic forces of Notch1^{-/-} super-fit cells could inhibit neighboring mutant cells and reduce deterministic tumor outcomes and similar again to previous competition experiments in the esophagus (<https://www.nature.com/articles/s41588-020-0624-3>). This is a very interesting concept, particularly given the Adaptive Oncogenesis model proposed by DeGregori or Winton's competition measurements in the colon.

A very important experiment will be to induce infrequent clonal Notch1^{-/-} mutations, then chemically incite tumors. This would determine whether tumor incidence decreases in a mosaic epithelium with Notch ^{+/+}, ^{+/-}, and ^{-/-} clones already present. In this manner, Notch^{-/-} clones would be placed in direct competition with tumorigenic clones – a more naturalistic model of the human condition. This would also enable the direct demonstration that somatic mutations capable of driving clonal expansions could act as tumor suppressors.

2. The overall enthusiasm for the human data set (figure 1) is somewhat tempered by the limited number of patients in the study – albeit the concern is somewhat mitigated by the inclusion of several technical replicates. The authors are to be commended for the attempt at coregistering the imaging data with genomics mutation data. However, the presentation in 1c is very difficult to follow and poorly labelled (i.e. x axis). It is also difficult to draw conclusions without knowledge of inpatient and interpatient variability. Can conclusive images for high or low notch be magnified and then “coregistered” in panel c with appropriate callouts/highlights etc. Most interesting would be an inpatient analysis of the same section, with clear annotations of high/low notch correlating with VAF/CNLOH.

3. In figure 2, three recombination events are shown: Mutant Notch1, YFP+ Mutant Notch1, and YFP Notch^{+/+}. What is the frequency of these events where recombination of only YFP or only Notch occur? Previously these single recombination events have been reported to be highly differential depending on the transgene (PMID: 24355609). Interpreting the majority of the lineage tracing data is predicated on being able to identify NOTCH^{+/+}, Notch^{+/-}, and Notch^{-/-}. It is unclear from the current data if IF alone is a reasonable solution. Additional in situ proof is likely necessary (mRNA FISH etc.).

4. Similar to 2 above, overall basal patch size is provided in figure 2 (basal cell number per clone) and similar patches provided in figure 3c. For the clonal experiments it would very important to show whether or not overall clone number changes and how competition proceeds over time (i.e. +/+, +/-, -/- clone numbers over time and clone sizes over time).

5. The EdU experiments (instantaneous labelling – 1hr and pulse-chasing – 48 hrs) are important for understanding how clones grow larger in the Notch1^{-/-} epithelium. Given that scRNASeq did not reveal changes in the cell number (see below if more analysis is warranted). If there are no differences in proliferation then there must be differences in the rate of differentiation, thus the importance of the 48 hour chase. The 48 hours timepoint is poorly described. I think the data support the conclusion that the reduction in EdU positive suprabasal -/- is indicative of less differentiation and/or the increase in suprabasal +/+ is indicative of forced differentiation to provide “space” for -/- cells. If this is the case, then do the scRNAseq data support this conclusion? (Fig 2 “l, j, m, n” could all use appropriate figure headings). Please consider clarifying the description of the data and the presentation of the data.

6. Figure 4 genomics data is not overly helpful and could be moved to the supplement. The scRNAseq is really underutilized, particularly understated is the potential for difference in cell states between the genotypes. Was a pseudotime trajectory analysis performed / could it be useful to understand how homeostasis is maintained in Notch1^{-/-} ? Does this data support the overall lineage tracing / EdU analysis conclusions?

Minor Comments

VAF1 and VAF2 are not well defined, how do these relate to one another? Which mutation is being analyzed?

In Extended Data 3g, the text remarks that Notch^{+/-} could be distinguished from Notch^{-/-} based on intensity of the signal. An inset demonstrating the clear difference in intensity would be helpful to boost the confidence in these results.

The imaging is rather difficult to interpret due to the size of the presented images, consider revising accordingly.

In the Tamoxifen experiments for Figure 5 Notch^{flox/flox} mice were injected with a quarter of the dose of the Notch^{+/+} and Notch^{+/flox} mice. This lower induction rate is then compared to a higher induction rate to demonstrate no differences in tissue morphology and function.

The genetic nomenclature for the mice is very difficult to follow and sometimes not uniform throughout the legends.

Figure 7i denotes 16 weeks of antibody treatment, but the text reports 6 weeks of antibody treatment.

The manuscript is well-written but has a few Grammatical errors:

- ED Figure 3a legend: allowiNotch1
- ED Figure 3f: non uniform italics used
- No callout for Figure 4b in text
- Page 31: "short-term analysis the revealed"
- "recombination of in..." poor grammar p7
- Figure 2b, add figure label for Notch mRNA
- "excision of p 9"

Reviewer #4:

Remarks to the Author:

The manuscript by Abby et al describes a comprehensive work focuses on understanding how Notch1 mutations drive clonal expansion in normal esophageal epithelium but (somewhat surprisingly) impairs tumor growth. The work encompasses remarkably wide set assays ranging from sequencing of tissues from human donors, clonal analysis via imaging and sequencing of mouse models, and comparative analysis of tumors vs healthy tissues. These provide a clear understanding of a highly intriguing phenomenon: That large fraction of the normal esophageal epithelium accumulates mutations in Notch1 as we age. These mutations in Notch1 promote clonal expansion allowing mutant clones to take over neighboring wildtype tissue. The growth advantage is conferred by Notch1 heterozygous mutants (N1+/-) and even more so by homozygous mutants (N1-/-). A computational model based on analysis of clone size distribution largely captures the observed dynamics. Surprisingly, these Notch1 mutants do not affect the normal tissue morphology or characteristics (but do show specific changes in gene expression). Moreover, experiments inducing tumors in the tissue showed that tumors are less likely to accumulate Notch1 mutations compared to normal tissue and that deletion of Notch1 or inhibiting it using anti-Notch1 blocking antibody reduces tumor growth. This suggests a potential treatment path for esophagus cancer. Additional analysis such as identifying the specific types of Notch1 mutations (either in the ligand binding domain or in the NRR domain) and their effect on Notch activation, and quantitative analysis of growth rates in clones provide further support for the conclusions and details on the mechanisms the behind enhanced clonal expansion.

The manuscript is written in an exceptionally clear manner, and the data is well presented and overall convincing. While there are a few questions that require clarifications, I do not have any major issues

with the scientific claims and the interpretation of the experimental results. I therefore recommend the manuscript for publication in Nature genetics after addressing the small concerns below.

Minor issues:

1. I am a little confused by the results of the EDU experiments shown in Figure 2g-n. I understand why the short EDU experiments show that proliferation in the basal levels is not affected by Notch1 mutation. This means that the differences observed in the longer EDU experiment are due to differences in relative turnover rates, namely, when competing Notch1 mutant and WT cells, the latter turn over faster. How does this fit the clonal expansion model? Also, how do the results showing higher EDU positive cells at the WT boundary support the clonal expansion model? Moreover, since it is later claimed that Notch1^{-/-} is more efficient in clonal expansion than Notch1^{+/-}, shouldn't you expect to see even lower EDU levels in the former?
2. In the computational model you estimate the fitness advantage of the Notch1^{+/-} and Notch1^{-/-} cells over WT. First, it is unclear why in ED Figure 4b, the best fit for Fitness are obtained only in specific ranges of induction levels. Since the induction ranges for N1^{+/-} and N1^{-/-} do not overlap how can a single preferred parameter set be deduced? (I'm probably just missing some explanation).
3. It would be useful to relate the values of fitness advantages you get in the model to the experimentally measured changes in EDU positive cells. Do these match the observed values.
4. In page 3, the sentence '94% (15/16) of negative staining areas in the two oldest donors carried nonsense or essential splice...'. Shouldn't it be 'missense' instead of 'nonsense'?
5. In the second to last paragraph in the discussion you should replace 'pleiotrophic effects' with 'pleiotropic effects'

Author Rebuttal to Initial comments

Response to reviewers

We are most grateful to the reviewers for their careful reading of the manuscript and insightful comments which have significantly improved the revised paper. In the below our responses are in blue.

As we have responded to the reviewers we have sought to clarify that the manuscript addresses the function of mutant *Notch1* in different stages of the colonization of normal tissue and in cancer development. Having shown the extent of *NOTCH1* mutation and LOH in human esophagus, we use a mouse model to show that the rapid clonal expansion of *Notch1* mutant cells in normal epithelium is due to an imbalance in mutant cell fate, with more progenitors than differentiating cells are generated per cell division. This enables clone growth in a normal tissue where division rates are constrained by cellular homeostasis (a cell can only divide when a nearby cell differentiates out of the proliferative layer). When the second *Notch1* allele is lost, the mutant cells promote the differentiation of adjacent wild type cells, further accelerating mutant clonal expansion. Following this phase of colonization, the now *Notch1* null epithelium reverts to a near normal state, as shown by scRNA seq and other analyses. Finally, we show

that loss of *Notch1* impairs the growth of tumors carrying *Atp2a2* driver mutations. As tumor cell division is not subject to constraints of a normal homeostatic tissue, the faster cells divide the faster the tumor will grow. Loss of *Notch1* impairs *Atp2a2* mutant signaling and decreases the rate of tumor cell division compared with *Notch1* wild type lesions. We conclude that *Notch1* illustrates how inactivating mutations in the same gene can drive clonal expansion in a normal tissue but inhibit tumor growth, by virtue of the distinct changes in cellular dynamics that result from *Notch1* loss in the context of a wild type normal tissue and a mutated tumor.

Reviewer#1: Remarks to the Author:

The paper by Emilie Abby, Phil Jones and co-authors aims to understand the role of somatic *Notch1* LoF mutations in normal oesophagus and in the development of oesophageal cancer.

Notch1 is a known cancer gene in several cancer types, including oesophageal cancer. However recently it has been found mutated at higher frequency in adult oesophagus. Understanding the role of *Notch1* mutations in normal oesophagus and in tumourigenesis is therefore a hot topic in the field.

In the first part of the study the authors very elegantly show how heterozygous and then homozygous mutations in *Notch1* give the cell higher fitness in both mouse and human. *Notch1* mutant cells are progressively able to colonise the whole oesophagus, to then revert back to wild type behaviour, so that the oesophagus remains histologically normal. What remains to be understood is why do *Notch1* mutant cells have higher fitness?

We thank the reviewer for their comments on the part of the paper which addresses the expansion of *Notch1* mutant clones within normal epithelium and agree that the basis of the higher fitness of *Notch1* mutant cells is indeed an important issue that we address in the first part of the text.

As the reviewer comments, lineage tracing indicates that *Notch1* mutant clones grow significantly larger than wild type clones (**previous Fig.2d-f and EDFig3f-k; now gathered into Fig.2 for clarity**).

We apologize for the confusion caused by our description of the EdU experiments. To understand the cellular mechanism of underpinning *Notch1* mutant clone growth, we performed EdU labelling assays at 1 and 48 hours post labelling. While the former is widely used, the latter is not and we now seek to explain it more clearly.

The proportion of basal cells positive for EdU at 1 hour is a measure of the fraction of cells in S phase. This is similar for in cells within *Notch1* +/- and -/- clones and wild type cells distant from mutant clones (**Fig.2e-h**), arguing *Notch1* mutant basal cells do not have an increased proliferation rate.

The 48 hour EdU experiment is performed track the fate of in S phase cells over the following 48 hour period. After taking up EdU, cells proceed through mitosis and both daughters inherit the EdU label. The pair of labelled cells may remain in the basal layer, or one or both may differentiate and exit the basal layer (**Fig. 2i**). The ratio of the number of EdU labelled suprabasal cells to the total of EdU labelled cells reflects the generation of differentiating cells in the basal layer and their stratification into the suprabasal cell layers.

In *Notch1* +/- and *Notch1* -/- mutants, this ratio is decreased within mutant clones, consistent with a tilt in mutant progenitor cell fate so that a higher proportion of progenitors and fewer differentiating cells are produced per average cell division (**Fig.2k**). Moreover, with *Notch1*-/- clones we also see an increase in the suprabasal EdU+: total EdU + ratio in the *wild type* cells at the clone margin compared with wild type cells further from the mutant clones. This, indicates that wild type cells next to clones differentiate and stratify at an increased rate (**Fig.2j,l**).

Significantly, basal cell density (cells/unit area) is the same in mutant clones and wild type cells (**Fig.2m,n**). To maintain a constant cell density, cell division is linked with the exit of a nearby cell from the basal layer (Mesa et al 2018). Hence, driving differentiation and stratification of wild type cells at the clone edge permits *Notch1*-/- cell division, further increasing the proportion of mutant progenitors (**Fig. 2o**).

Such 'supercompetition' by inducing differentiation of adjacent wild type cells contributes to the enhanced fitness of *Notch1* null over *Notch1* heterozygous cells (**Fig. 3h, ED Fig. 4**). Previous studies have seen a similar induction of keratinocyte differentiation adjacent to Notch pathway mutants (Alcolea et al 2014, Lowell et al 2000).

We have now clarified the account of the EdU assays and the mechanism of *Notch1* mutant clonal expansion in Fig2 and in the text as follows:

Page 11:

We first counted the proportion of basal cells positive for EdU at 1 hour after labelling, which measures the fraction of basal cells in S phase, to assess whether an increase in the rate of cell division contributed to mutant clone expansion (Fig.2e, f). This value was similar for cells within *Notch1*^{+/+} clones and wild type cells distant from clones (Fig.2g). Within *Notch1*^{-/-} mutant clones the proportion of EdU positive basal cells was marginally lower than in wild type cells (Fig. 2h). We conclude neither *Notch1*^{+/+} nor *Notch1*^{-/-} clonal expansion results from an increase in mutant cell division rate compared with wild type cells.

The 48 hour EdU experiment labels S phase cells and tracks the fate of the two cells generated by the subsequent mitosis over the following 48 hours. The pair of labelled cells may remain in the basal layer, or one or both may differentiate and exit the basal layer (**Fig. 2i**). The ratio of EdU labelled suprabasal cells to the total EdU labelled cells reflects the rate of production of differentiating cells in the basal layer and their stratification into the suprabasal cell layers. In *Notch1^{+/-}* and *Notch1^{-/-}* clones, this ratio is decreased, consistent with a tilt in mutant progenitor cell fate so that more progenitor and fewer differentiating daughters are produced per average cell division (**Fig.2k**). Strikingly, adjacent to *Notch1^{-/-}* clones there was an increase in the suprabasal EdU+: total EdU+ cell ratio in the *wild type* cells at the clone margin compared with wild type cells further from the mutant clone (**Fig.2j,l**).

This, along with a small decrease in the proportion of wild type S phase cells at the clone edge, indicates that wild type cells adjacent to the clone exit the cell cycle, differentiate and stratify out of the basal layer at an increased rate, a phenomenon also reported in previous studies of Notch inhibited keratinocytes interacting with wild type cells (**Fig. 2h**) (Alcolea et al 2014, Lowell et al 2000).

These observations explain the increased fitness of *Notch1^{-/-}* over *Notch1^{+/-}* clones. Cell density was similar in both mutant genotypes and wild type areas suggesting that the linkage between cell division and the exit of a nearby differentiating cell from the basal layer is maintained (Fig.2m,n). Within this constraint, the driving of wild type cell differentiation and stratification permits *Notch1^{-/-}* cell division at the clone edge, accelerating clonal expansion (Fig. 2o).'

No significant differences emerge from the transcriptional profile thus suggesting that these cells are phenotypically similar.

We performed transcriptomic analysis to phenotype epithelium that had been fully colonized by *Notch1^{-/-}* cells, reflecting the situation in aging humans (**Fig4, EDFig.6**). We have now clarified this point in figures **Fig.4a, EDFig.6** and in the text:

Page 24:

The replacement of large areas of the esophageal epithelium by clones lacking functional NOTCH1 in aging humans might be expected to result in cellular phenotypes. To explore the effects of *Notch1* loss in the murine esophagus we first performed bulk RNA sequencing on peeled epithelium from wild type, and highly induced, fully colonized *Notch1^{+/-}* and *Notch1^{-/-}* esophagus (Extended data Fig.3f-h, Extended data Fig. 6a-e). '

We agree with Reviewer 1 that the conclusion that these analyses is that, once occupied by *Notch1* mutant cells, the mutant epithelium reverts to a phenotype similar to wild type tissue.

Strangely, the authors do not think of any possible role played by non-epithelial cells and/or cell nonautonomous mechanisms that could impact differently on the fitness of *Notch1* mut and wt cells. In such a detailed study, this is a missing opportunity.

As discussed above, we do uncover a non-cell autonomous mechanism of differentiation induction in wild type keratinocytes at the margin of *Notch1* null clones. In terms of analyzing other cell types, we only induce *Notch1* mutation in the epithelium, as we set out to model ageing human esophagus. For our scRNA seq analysis we therefore used peeled and dissociated *Notch1*^{-/-} and wild type epithelium. There were sufficient fibroblasts to allow us to investigate this cell type however. **Reviewer Figure 1a** shows the overlay of the fibroblasts from the mutant and wild type libraries in UMAP space. The plots show no shape or separation related to the type of library from which they came. This is highly suggestive that fibroblasts in *Notch1*^{-/-} and *Notch1*^{+/+} epithelium have a similar transcriptional phenotype. Moreover, **Reviewer Figure 1 b-c** shows the cell cycle phases of fibroblasts in UMAP space, again there is no substantive difference comparing *Notch1*^{-/-} and *Notch1*^{+/+} epithelium.

a.

b.

c.

Reviewer Figure 1: Single-cell analysis of the fibroblasts population from *Notch1*^{+/+} and *Notch1*^{-/-} peeled and dissociated mouse epithelia

a. UMAP plot shows an overlay of the cells annotated as fibroblasts from each library (+/+1, n=641; +/+2, n=717; -/-1, n=341; -/-2, n=437). b. UMAP plot of fibroblasts and the likely cell cycle phase they belong to (G1, G2/M or S). c. Stacked bar chart shows the proportion of fibroblasts per library assigned to each cell cycle phase.

To further address this issue, we used an *in vitro* competition assay with 3D cultures of keratinocytes (**Reviewer Fig.2a**) (Fernandez-Antoran et al 2019, Herms et al 2021). A 50/50 mixture of YFP labelled *Notch1*^{+/+} or *Notch1*^{-/-} cells with YFP negative wild type cells was co-cultured and the proportion of cells of each genotype determined by flow cytometry at several time points (**Reviewer Fig.2b**). The assay shows a neutral competition between wild type cells while the *Notch1* mutant cells rapidly outcompete the wild type cells (**Reviewer Fig. 2c-e**). These findings confirm that the competitive advantage of *Notch1* mutant over wild type cells is independent of other cell types. In view of the limited space we have not included these results in the manuscript at present, but can do so should the Reviewers deem this desirable.

Reviewer Fig.2: In vitro competition assay shows that Notch1 clonal expansion is 'cell-type' autonomous

a. Primary 3D cultures of mouse esophageal keratinocytes were generated by culturing esophageal explants on inserts (Herms et al 2021). Keratinocytes proliferated and cover the insert and the esophageal explant is removed. Keratinocytes reach confluence and stratify. b. Cultures were generated from YFP^{Cre}Notch1^{+/+} or YFP^{Cre}Notch1^{fllox/fllox} mouse esophagus. Keratinocytes were dissociated and infected with an adenovirus encoding CRE recombinase (AdCRE) or not (AdNull). CRE expression lead to the nearly total recombination of YFP expression at Rosa26 locus in YFP⁺; +/+ cells and of both Rosa26^{YFP} and Notch1 loci in YFP⁺; -/- cells. YFP⁺; +/+ or YFP⁺; -/- cells were mixed 50/50

with the respective YFP-; +/+ (AdNull treated) cells. Cells were cultured for 4 days then collected at day 0, 15 and 30 for FACS analysis. c, d. SSC-A/GFP FACS plots show the keratinocytes populations positive and negative for YFP staining (GFP) at time (T) 0, 15 and 30 days. Gates were used to determine the proportion of YFP positive and positive cells. Results from competition between YFP+ and YFP- Notch1 wild type cells are shown in c. Results from competition between YFP+ Notch1^{-/-} cells and YFP- Notch1^{+/+} cells are shown in d. e. Plot showing the evolution of the proportion of YFP+ population between 0 and 30 days in each competition assay. Competition between Notch1^{+/+} cells shows no enrichment of the YFP+ population indicating neutral competition (grey). The proportion of YFP+ Notch1^{-/-} cells increased over time indicating that they outcompete Notch1^{+/+} cells (red). Mean ± SEM, n=4 biological replicates for YFP+; +/+ vs YFP-; +/+ competition assay; n=6 biological replicates for YFP+; -/- vs YFP-; +/+ competition assay.

In the second part of the study, the authors aim to understand the role of Notch mutations in tumourigenesis and their conclusions are that Notch1 loss impairs tumour growth. This part of the study is in my opinion less robust and does not explain a big paradox: Notch inactivating mutations are observed and frequent in ESCC (~10% of TCGA ESCC bear clonal truncating mutations). How can mutations that impair tumour growth be selected for in cancer cells? This evidence is completely neglected by the authors although it in principle invalidates their hypothesis.

The reviewer raises an important point. The occurrence of *NOTCH1* mutants in a minority of ESCC may appear paradoxical. Motivated by this we first performed an in depth copy number analysis of the *NOTCH1* locus in ESCC from the TCGA data set (Reviewer Fig.3)(Consortium 2017). Out of 90 ESCC for which information was available, 12 have a *NOTCH1* mutation, of which only 7 bear mutations affecting all copies of the gene, with 4 having truncating *NOTCH1* mutations (Reviewer Fig.3a). By comparison, 86/90 of ESCC carry *TP53* mutations (Reviewer Fig. 3b). It is also noticeable that unlike *TP53*, *NOTCH1* mutations are always found associated with at least one other mutation among the genes considered as a potential ESCC drivers, such as *TP53* or *KMT2D* (Consortium 2017). We would submit that it is possible that detrimental effects of *NOTCH1* mutation on tumor growth in 8% of ESCC, may be bypassed by mutation and copy number alterations in this subset of tumors.

Next, one must consider that *NOTCH1* mutants are selected in ESCC (Martincorena 2017). Selection reflects the full history of a mutant clone. In the case of a tumor this includes selection of mutations in normal tissue prior to tumor formation. Selection may be estimated from the ratio of protein altering to synonymous mutations (dN:dS), positive selection being indicated by the non-synonymous mutations exceeding the expected the number of synonymous mutations for a given gene allowing for parameters such as the mutational spectrum (Martincorena et al 2018, Martincorena et al 2017, Martincorena et al 2015). dN:dS ratios were compared for *NOTCH1*, *TP53*, and *KMT2D* in the TCGA dataset and in normal human esophagus (both of which have a median donor age of between 48-58 years) (Consortium 2017, Martincorena et al 2018). *NOTCH1* is very strongly selected in normal tissue and less so in ESCC while the converse is the case for *TP53* and *KMT2D* (Reviewer Fig.3c). Indeed, this reflects the proportion of

NOTCH1 mutant epithelium being ca. 8-10 fold higher than the proportion of mutant ESCC. We hypothesize that *NOTCH1* positive selection in the tumors is the result of mutational selection in the normal tissue prior tumor formation. The prevalence of *NOTCH1* mutants is strikingly depleted in ESCC in comparison with normal middle aged and older tissue consistent with the hypothesis that wild type *NOTCH1* favors tumor growth.

a.

b.

c.

Reviewer Fig.3: Analysis of genetic alterations at NOTCH1 locus in ESCC from TCGA dataset

a. Summary of the copy number and mutation status of chromosome 9 and NOTCH1 across 90 oesophageal squamous cell carcinoma tumours from TCGA. Each row represents a tumour and shows the copy number status along the whole of chromosome 9, colours represent different copy number states (loss of heterozygosity (LOH) with different major allele copy number in green, orange and pink, altered copy number without LOH in purple, unaltered copy number in white and the centromere in grey). No homozygous deletions were detected. The location of the NOTCH1 gene locus is highlighted via a vertical black line and at the end of each row the NOTCH1 mutation status is annotated (pink star when all available NOTCH1 copies are mutated, blue triangle if some NOTCH1 copies are mutated, white when no NOTCH1 mutation has been detected). b. Heatmap showing mutation calling and clonality analysis at loci from genes significantly mutated in ESCC (Consortium 2017). Symbols show in each ESCC and for each gene whether mutations are clonal and whether mutations effect all of the gene's copies. c. dN:dS ratios for *KMT2D*, *TP53* and *NOTCH1* in ESCC and normal epithelium, data from (Consortium 2017, Martincorena 2018).

In addition to the carcinogenesis experiments in the previous version of the manuscript, we have now studied further the mechanism by which *Notch1* loss impairs tumor growth. We performed RNA sequencing on both tumors and adjacent normal tissue from *Notch1*^{+/+} and *Notch1*^{-/-} esophageal epithelium (**Fig.8, ED Fig.8**). Analysis shows reduced cell division associated transcripts in the *Notch1*^{-/-} tumors compared to tumors expressing *Notch1* (**Fig.8d-f, ED Fig.8 f-h**). These findings were further validated by finding a significant reduction in the proportion of proliferating keratinocytes expressing the G2/M phase markers CyclinB1 and pHH3 in Notch null tumors (**Fig.8 g-j**). Moreover, the *Notch1*^{-/-} tumors show reduced p-ERK staining, consistent with decreased signaling downstream of the *Atp2a2* mutation that drives mouse tumor formation, which may contribute to this reduced cell division, (**Fig.8 k,l**).

We conclude that *Notch1* mutation drives clonal expansion in normal esophageal epithelium but impairs tumor growth. Within normal epithelium, *Notch1* mutant clones expand within the spatial constraint of the basal layer, where cell division coupled to the exit of differentiating cells through stratification (Piedrafita et al 2020). The rate of division of *Notch1* mutant cells was not increased compared to that of wild type cells, but the clones grow by as mutant progenitors produce more progenitors than differentiating daughters on average. In the tumor context, the disorganized structure of the lesion means there is no spatial constraints on the ability of cells to divide. The rate of cell division thus limits the rate of tumor growth. *Notch1* mutant tumor cells have a lower cell division rate compared with *Notch1* wild type cells and so *Notch1* mutant tumors grow more slowly than *Notch1* wild type lesions (**Fig.8m**).

We now comment on the presence of *NOTCH1* mutations in a minority of human ESCC and on our findings in this study in the discussion as follows

Page 41:

'Might these findings be of relevance in humans? That over 90% of human ESCC retain one or more wild type copies of *NOTCH1* while arising from an epithelium where the majority of cells have biallelic *NOTCH1* disruption argues wild type *NOTCH1* does promote cancer development. The complexity of genomic alterations in human ESCC may explain the ability of the minority of tumors to bypass the benefit of retaining wild type *NOTCH1*(Consortium 2017). '

General comments

p.2: the roles of Notch1 in cancer are not 'controversial': as other cancer genes (TP53, XXX) can act as a tumour suppressor or as an oncogene, depending on the context. The first 2 references cited here (1,2) do not support either role: they report higher frequency of mutations of Notch1 in normal oesophagus. These references should be removed from this paragraph.

We agree that 'controversial' is an overstatement, have removed the citation of the 2 references at this point and have revised the text as follows:

Page 2:

Different studies have suggested that *NOTCH1* is a tumor suppressor or conversely may promote esophageal carcinogenesis (Lubin et al 2018, Natsuizaka et al 2017, Sawangarun et al 2018). '

p.3: why has Kendall's tau-b test been preferred to Spearman correlation to assess the association bw Notch1 mutations and age?

We used Kendall's tau because of some recommendations that it can be more accurate with low sample sizes:

From the SciPy documentation for spearmanr: "The p-values are not entirely reliable but are probably reasonable for datasets larger than 500 or so"

(<https://docs.scipy.org/doc/scipy/reference/generated/scipy.stats.spearmanr.html>)

<https://www.statisticssolutions.com/kendalls-tau-and-spearman-rank-correlation-coefficient/>

"[Kendall's Tau] P values are more accurate with smaller sample sizes"

However, since Spearman is more commonly used than Kendall's tau, and is a highly similar statistic (Gilpin 1993), we would be happy to show these results instead if that is preferred. The Spearman correlation results for the same data are correlation=-0.85, p=0.004.

Then they investigate how NOTCH1 mutant clones colonise normal epithelium in mouse and notice progressively larger sizes of Notch1+/- and Notch1-/- clones. However, they do not observe different

rates of cell division of basal cells but a lower rate of cell division in the suprabasal cells in the mutant clones

We believe this comment arises from a lack of clarity in our description of the organization of the mouse esophageal epithelium and of the 48 hour EdU assay in the first version of the manuscript. Unlike humans, in mouse esophagus, proliferation is confined to the basal layer of the tissue. We now state this explicitly and clarify the dynamics of mouse esophageal epithelium as follows:

Page 8:

'Mouse esophageal epithelium consists of layers of keratinocytes in which proliferation is restricted to progenitor cells in the basal layer (**Extended data Fig. 2a**). Differentiating cells cease dividing, leave the basal layer and migrate towards the epithelial surface where they are shed. Progenitor cell division is linked to exit of a nearby differentiating cell from the basal layer so a constant cell density is maintained (Mesa et al 2018). Dividing progenitors generate either two progenitor daughters that remain in the basal layer, two differentiating daughters or one cell of each type. In wild type tissue, the probabilities of each progenitor outcome are balanced, generating equal proportions of progenitor and differentiated cells, maintaining cellular homeostasis across the progenitor population (**Extended data Fig. 2a**) (Doupe et al 2012, Piedrafita et al 2020). Mutations can alter the balance of progenitor fate leading to excessive production of progenitors driving the growth of mutant clones (Alcolea et al 2014, Murai et al 2018).'

As discussed above, the 48 hour EdU experiment labels S phase cells and tracks the fate of the 2 daughter cells generated at the following mitosis over a 48 hour period. The pair of labelled cells may remain in the basal layer, or one or both may differentiate and exit the basal layer (**Fig. 2i**). The ratio of the EdU labelled suprabasal cells to the total EdU labelled cells reflects the production of differentiating cells in the basal layer and their stratification into the suprabasal cell layers.

Figure 2m,n: shouldn't a lower proliferation of suprabasal cells be expected in $-/-$ vs $+/-$ since $-/-$ cells have higher fitness and their clones reach larger size than $+/-$ cells? This was a surprising result that needs further investigation. Probably a direct comparison between the two conditions in the same setting will shed some light.

Fig. 2m,n. is now Fig 2. k,l. As noted previously, the 48 hour EdU assay reports non-dividing, differentiated suprabasal cells, which is reduced in both $+/-$ and $-/-$ mutant cells. This reflects the bias in mutant progenitor cell fate with more progenitor and fewer differentiating daughter cells are produced per average cell division. As discussed above, the increased fitness of $-/-$ cells over $+/-$ may reflect non-cell autonomous effects on wild type cells adjacent to the clone (**Fig 2l, j**). Increased stratification of wild type cells at the clone margin permits mutant cell division facilitating the expansion of the $-/-$ mutant clone with its increased proportion of progenitor cells.

Figure 3b,e,f: I am not entirely clear why areas with missense mutations whose VAF is >0.2 are negative for Notch1 staining? Shouldn't the mutated allele still be able to produce some protein, even though not functional? Are these mutations all able to alter the binding of the antibody used for staining (although this seems unlikely). This requires further investigation. I would tend to disagree with the conclusion of the paragraph (Overall the effect of these mutations on NOTCH1 expression is consistent with our observations in human tissue) because in human the absence of Notch staining was associated with complete loss of both alleles.

We regret the lack of clarity in explaining this complex figure that has caused confusion.

Fig. 3b shows how NOTCH1 protein is lost over time in some areas of the *Notch1*^{+/-} esophageal epithelium as mice age due to spontaneous mutations in the remaining *Notch1* allele. Mutations that disrupt protein expression result in regions of negative staining, these accumulate over time **Fig. 3c**.

Fig. 3e shows ageing *Notch1*^{+/-} mice in which we induced YFP as a lineage marker to assist in clonal detection alongside NOTCH1 staining. The colored dots illustrate the findings, for example the area with a yellow circle is likely to be clonal as shown by YFP staining, and negative for NOTCH1. In **Fig. 3f** we show the results of sequencing micro dissected epithelium from the experiment illustrated in **Fig. 3e**. Mutations and copy neutral loss of heterozygosity (CNLOH) are found at single *Notch1* locus in the expanded areas in comparison to control. This plot indicates that negative NOTCH1 staining areas in +/- mice are associated either with a *Notch1* mutation or CNLOH. The types of mutations detected in the positive and negative areas are displayed in **Fig. 3g**. It shows that expanded negative areas are associated with Nonsense or Indel/Splicing mutations. We would submit these results are consistent with the data in the human samples. Missense mutations are however are found in positive NOTCH1 staining areas, as they still allow the production of the protein, in agreement with the Reviewer's comment.

We have revised our explanation of this figure in the new version of the text as follows

Page 18:

Among clones carrying a *Notch1* mutation, NOTCH1 positive staining clones harbored mainly missense mutations (85%) while NOTCH1 negative staining clones harbored mainly Indel/splicing (51%) or nonsense mutations (46%) (Fig. 3g). Overall the effect of these mutations on NOTCH1 expression is consistent with our observations in human tissue (Fig.1).

p.17: “This observation associated with the absence of detectable nuclear NOTCH1 after EDTA treatment in this group of clones suggests that the mutations they carry at the NRR domain may lock it into its auto-inhibited conformation, preventing the cleavage of NOTCH”. This needs to be proven. There may be other effect that the mutations are inducing, other than locking the protein into an auto-inhibited conformation

-We agree with Reviewer 1 that there could be other explanations for this observation so we have now changed this sentence as follows:

Page 19:

“This observation associated with the absence of detectable nuclear NOTCH1 after EDTA treatment in this group of clones suggests that the mutations they carry at the NRR domain may prevent the cleavage of NOTCH1.”

p.20: where are the simulation of neutral mutations? How do the authors assess neutrality?

The Reviewer is referring to the sentence: “In simulations, neutral mutations do not spread across the tissue and are likely to disappear from the tissue over time.” We regret the lack of clarity on this point in the first version of the manuscript.

The simulations are not shown here, but this is a key feature of the neutral stochastic single progenitor model that describes the behavior of progenitor cells in squamous epithelia including those of the murine esophagus (Clayton et al 2007, Doupe et al 2012, Piedrafita et al 2020). Neutrality is defined as not changing the selective advantage of cells, which, depending on how the cell dynamics are described, could mean not altering the cell fitness or cell division outcomes. Neutral mutations can be seen in this study as the YFP reporter in the wild type, *Notch1* +/- lineage tracing, but more detail on neutral cell dynamics is available in (Clayton et al 2007, Doupe et al 2012, Piedrafita et al 2020).

Neutral behavior is substantially smaller in extent than non-neutral growth, and would therefore not be visible on the same plot as the haplosufficient and haploinsufficient simulations. To illustrate this, the blue line at the bottom of **Reviewer Figure 4** shows the mean of 10 neutral simulations run compared to the data shown in **ED Fig4e**.

Reviewer Figure 4: Proportion of tissue covered by Notch1^{-/-} clones over time. The mean of 10 simulations in which both Notch1^{+/-} and Notch1^{-/-} mutations are neutral (blue), compared to the experimental data (black). The Notch1 mutation rate in the simulations was set so that the haploinsufficient simulations (Extended Data Figure 4e) approximated the experimental data.

We have changed the text to remove “In simulations” and added the references to be clearer that we are not showing these simulations in this study, but are referring to previous studies. The new sentence reads:

Page 24 in main text:

‘Neutral mutations do not spread across the tissue (Clayton et al 2007, Doupe et al 2012).’

p.3: “negative staining areas in the two oldest donors carried nonsense or essential splice mutations or indels in NOTCH1 with copy neutral loss of heterozygosity (CNLOH) of the NOTCH1 locus (human GRCh37- chr9:139,388,896-139,440,238) or a further mutation, likely to disrupt the second NOTCH1 allele 1.” This is a result obtained in this study, reference to Martincorena et al is inappropriate.

-We agree and have now removed this reference.

“NOTCH1 protein and NICD1, which is detectable in the nucleus during active signalling” this needs to be referenced as it is based on previous knowledge

-We have now added a review on NICD and its activity (Bray & Gomez-Lamarca 2018).

p.25: "NOTCH1 mutations are under strong positive selection in normal human esophagus and occupy 30% to 80% of the epithelium as early as middle-age but are only found in about 10% of esophageal squamous cell carcinomas (ESCC) 1,3." Ref.2 should be cited here

-We have added reference 2.

Reviewer #2: Remarks to the Author:

This manuscript elegantly describes the conflicting action of Notch1 mutation in normal and tumorigenic esophageal epithelium. In aging human esophagus, sequencing reveals that Notch1 mutant clones which occupy a significant proportion of the epithelium carry biallelic alterations. To test the fitness impacts of Notch mutant clones, the authors developed a conditional transgenic mouse where one or both alleles of Notch could be deleted and induce a fluorescent lineage reporter. The authors used lineage tracing and Notch staining in this transgenic mouse to quantify the fitness advantage that the loss of one allele confers over wild type epithelium. This fitness advantage was further magnified by the loss of the second allele. Sequencing of mutant clones revealed slight differences in transcription. However, the authors reported no functional difference in the morphological analysis of the esophageal epithelium. Finally, the authors induced tumors in this mouse model and found that Notch mutants were less numerous in tumors compared to the normal adjacent epithelium. Complete loss of Notch1 and antibody blockade of Notch1 resulted in significantly smaller tumors, but no change in tumor number. The authors are to be commended for an exhaustive analysis of Notch1 mutations in human patient specimens and their application of an informative mouse model for studying Notch1 clonal competition and its effects on tumorigenesis. The summary is clear and overall the manuscript is well written. An appropriate review of the literature is included.

However, the major question in the study is to determine how Notch1^{-/-} clones spread throughout the esophagus yet are rarely found in tumors. Interestingly, the group's 2020 Nature Genetics paper describes a very elegant model in which cell fitness is relative to neighboring clones (as cartooned in Figure 6f, 2020). From this perspective Notch1 mutations could inhibit tumorigenesis through two likely mechanisms, Notch1^{-/-} could intrinsically inhibit tumorigenic potential or Notch1^{-/-} could extrinsically inhibit tumorigenic potential. The current manuscript seems to suggest an extrinsic mechanism whereby increased cell fitness of Notch1^{-/-} cells prevents the growth of tumors. This is a very attractive hypothesis and extremely exciting. If proven, it would be a significant contribution to the field and warrant reconsideration. This concern is the most important to address and listed as priority 1 below. Several additional concerns are also provided – that are likely more easily rectified through textual revision and/or additional analysis/reanalysis of available data.

Major Experimental Concerns

1. The current data/experiments do not address the paradoxical abundance of human Notch1^{+/+} esophageal cancers despite the field/clonal expansion of Notch1^{-/-} in aged epithelium. The tumor modeling using the high induction model seems to test the intrinsic hypothesis because most cells will be Notch1^{-/-} and receive secondary hits in the same cell type. However, the extrinsic hypothesis is left untested. Extrinsic forces of Notch1^{-/-} super-fit cells could inhibit neighboring mutant cells and reduce deterministic tumor outcomes and similar again to previous competition experiments in the esophagus (<https://www.nature.com/articles/s41588-020-0624-3> [nature.com]). This is a very interesting concept, particularly given the Adaptive Oncogenesis model proposed by DeGregori or Winton's competition measurements in the colon.

A very important experiment will be to induce infrequent clonal Notch1^{-/-} mutations, then chemically incite tumors. This would determine whether tumor incidence decreases in a mosaic epithelium with Notch ^{+/+}, ^{+/-}, and ^{-/-} clones already present. In this manner, Notch^{-/-} clones would be placed in direct competition with tumorigenic clones – a more naturalistic model of the human condition. This would also enable the direct demonstration that somatic mutations capable of driving clonal expansions could act as tumor suppressors.

We thank the reviewer for raising the very interesting and important question about a potential extrinsic effect of *Notch1* mutant on carcinogenesis.

In this work we study the effect of *Notch1* mutation on high grade dysplastic tumors generated by a combination of exposure to a nitrosamine (DEN) and a promoter (Sorafenib) (Frede et al 2016). We find that tumors generated in esophagus null for *Notch1* are smaller than those produced from wild type mice and that treatment of established lesions with a function blocking NOTCH1 antibody decreases lesion size. These findings argue *Notch1* promotes the growth of established tumors. There is no difference in the density of lesions across genotypes arguing *Notch1* does not play a role in the initiation of macroscopic tumors generated with this two-stage protocol (Fig. 7). Interestingly, in tumors generated in wild type mice, we find *Notch1* mutants occupy 30-50% of the lesion and NOTCH1 signaling remains active whereas in the surrounding epithelium the prevalence of *Notch1* mutation approaches 100% resulting in nearly complete inactivation of signaling as assessed by cleavage of the Notch1 cytoplasmic domain (Fig. 6b-g).

The reviewer comments 'Extrinsic forces of Notch1^{-/-} super-fit cells could inhibit neighboring mutant cells and reduce deterministic tumor outcomes' and suggests that we investigate if 'tumor incidence decreases in a mosaic epithelium with Notch ^{+/+}, ^{+/-}, and ^{-/-} clones already present'. We agree that the concept of 'super-fit' clones having a tumor suppressive role is of great interest and indeed have demonstrated this using a dominant negative *Mam1* mutant fused to GFP with microscopic intraepithelial lesions which are

displaced from esophageal epithelium by clonal expansion (Colom et al 2021). The use of the transgenic, GFP linked construct allowed direct visualization of a superfit clone in an epithelium that was carrying a high density of mutant clones, most frequently mutant *Notch1*, due to treatment with DEN and subsequent clonal expansion. Our present work studies the 5% or so of lesions that survive this process of selection and reach macroscopic size (Colom et al 2021).

We explored the feasibility of the experiments suggested by the reviewers by evaluating the prevalence of DEN induced *Notch1* mutant clones in the esophageal epithelium at the time point we collect macroscopic tumors. Epithelium from wild type mice treated with DEN and Sorafenib was subjected to deep targeted sequencing (**Fig. 6a-c**) and immunostaining for NOTCH1 protein (**Reviewer Fig. 5**). The high background rate of *Notch1* mutation and the evidence of large clonal expansions negative for NOTCH1 protein expression argues that the effect of inducing infrequent clonal *Notch1*⁻ mutations may be challenging to detect. More conclusively, prior deletion of *Notch1* from the entire esophagus resulted in no detectable change in tumor density compared with wild type tissue with this protocol (**Fig. 7c**).

Reviewer Figure 5: NOTCH1 staining shows the mosaic of Notch1 mutations in DEN and Sorafenib treated *Notch1*^{+/+} esophageal epithelium

a. *Notch1* wild type mice were treated with DEN and Sorafenib. Tissue was collected 28 weeks after treatment. b. Peeled esophageal epithelium was stained for NOTCH1 (grey) and DNA (blue). Staining shows a mosaic of *Notch1* mutant clones (staining positive and negative for NOTCH1) covering the epithelium (Fig.6a-c). Scale bar, 500 μ m.

In DEN treated wild type tissue, the ‘extrinsic’ selection of microtumors might be expected to result in selection of lesions carrying *Notch1* mutations, as these would be expected to resist competition from super fit clones in the normal epithelium (Colom et al 2021). However, DNA sequencing shows that the developed lesions that are focus of this study are actually depleted in *Notch1* mutations compared with the surrounding epithelium, paralleling the relative prevalence of *NOTCH1* mutations in human epithelium and ESCC (ED Fig 8 a,b). Despite developing while surrounded by rapidly expanding *Notch1* mutant clones in the epithelium, these tumors retained functional NOTCH1 and grew larger than *Notch1*^{-/-} tumors. This argues that wild type *Notch1* has a direct role in promoting tumor growth. Tumors carried mutations in *Atp2a2*, which is a driver of ESCC in mice, acting by increasing the levels of wild type KRAS protein, and found to be mutated only rarely in normal epithelium (Liu et al 2001, Prasad et al 2005) (ED Fig 8 a,b). We conclude that *Atp2a2* mutation and/or other genetic events initiate the development of macroscopic tumors in the DEN/Sorafenib model and that *Notch1* mutation does not positively contribute to this process. Overall these results replicate the ‘the paradoxical abundance of human *Notch1*^{+/+} esophageal cancers despite the field/clonal expansion of *Notch1*^{-/-} in aged epithelium’ and that the model used in our study is suitable to address this paradox.

We discuss these issues in the text as follows:

Pages 30/31:

-‘In tumors, the most selected gene was the known mouse esophageal tumor driver *Atp2a2*, which is rarely mutated and not selected in normal epithelium (Extended data Fig. 8a,b, Supplementary table 13) (Liu et al 2001, Prasad et al 2005). *Notch1* mutations were similar to the protein disrupting mutants in normal tissue with no evidence of activating mutations. However, *Notch1* mutants were both less selected and less prevalent in tumors than in the adjacent tissue (Fig. 6b, c, Extended data Fig. 8a,b, Supplementary table 13). Immunostaining confirmed a higher proportion of cells staining positive for NOTCH1 and NICD1 in tumors than in the normal epithelium (Fig. 6d-g, Supplementary table 14). These findings indicate *Notch1* wild type cells are more likely to contribute to tumors than those carrying *Notch1* mutants. Overall the mouse model parallels the findings in humans with substantially higher prevalence of *NOTCH1* mutations in normal human esophageal epithelium than human esophageal squamous cell carcinoma. This led us to further investigate the role of *Notch1* in tumorigenesis in the mouse.’

We have now investigated the mechanism by which *Notch1* loss alters tumor growth (Fig.8 and ED Fig8). As discussed above we have now established that the tumors share a common oncogenic driver mutation, *Atp2a2* (ED Fig.8a-c). We next micro-dissected sections of tumors and adjacent normal epithelium and performed RNA sequencing (Fig.8a-f, ED Fig.8 d-h). Comparison of the transcriptomes profiles revealed that, in both genotypes, the key processes altered in growing tumors were an upregulation of DNA replication and cell cycle transcripts and a downregulation of mRNAs associated with lipid metabolic processes. These changes are consistent the reported effects of *Atp2a2* mutation on keratinocytes, i.e. activation of oncogenic signaling pathways, decreased differentiation and tumorigenesis (Celli et al 2012,

Hong et al 2010, Liu et al 2001, Prasad et al 2005). Comparison of tumors generated from *Notch1*^{+/+} and *Notch1*^{-/-} epithelium revealed that DNA replication and cell cycle associated transcripts are significantly downregulated in *Notch1*^{-/-} tumors (Fig. 8d-f, ED Fig. 8f-

h). We validated these findings by staining for the proliferation markers phospho-Histone H3 and CCNB1, finding both were significantly reduced in the *Notch1*^{-/-} tumors (Fig.8g-j). Finally, we measured phospho-ERK1/ERK2 staining in the tumors, as *Atp2a2* mutation is known to activate the RAS to ERK signaling axis (Celli et al 2012, Prasad et al 2005). We found a significant decrease in phospho-ERK staining in *Notch1*^{-/-} tumors compared to *Notch1*^{+/+} lesions (Fig.8k,l). These findings are consistent with the pro-proliferative signaling resulting *Atp2a2* mutation that sustains tumor growth being attenuated in the absence of *Notch1*.

We discuss this in the text as follows:

Page 36:

To investigate the mechanism by which *Notch1* loss alters tumor growth, we further characterized tumors collected 28 weeks after DEN and Sorafenib treatment (Fig.7a). DNA sequencing of *Notch1*^{-/-} tumors confirmed that they share the same oncogenic driver *Atp2a2* (6/7 tumors with nonsynonymous mutations) as the tumors from *Notch1* wild type genotype (17/17 tumors) (Extended data Fig.8a-c, Supplementary table 13) (Liu et al 2001, Prasad et al 2005). RNA sequencing and comparison of the transcriptomes of tumors and adjacent normal tissue showed that in both genotypes, the key processes altered in tumors were an upregulation of transcripts encoding genes linked with DNA replication, cell cycle and RNA processing and a downregulation of mRNAs associated with lipid metabolism (Fig.8a-c, Extended Data Fig.8 d-e, Supplementary table 17). These changes are consistent the reported effects of *Atp2a2* mutation on keratinocytes, with activation of oncogenic signaling pathways, decreased differentiation and tumorigenesis (Celli et al 2012, Hong et al 2010, Liu et al 2001, Prasad et al 2005). Comparison of tumors generated from *Notch1*^{+/+} and *Notch1*^{-/-} epithelium revealed that DNA replication and cell cycle associated transcripts were significantly downregulated in *Notch1*^{-/-} tumors (Fig. 8d-f, ED Fig. 8f-h, Supplementary table 17). We therefore quantified the proportion of cycling cells in the tumors using proliferation markers phospho-Histone H3 and CCNB1 within KRT14+ cells. Both markers revealed that cycling keratinocytes were significantly reduced in tumors from *Notch1*^{-/-} esophagus (Fig. 8g-j, Supplementary table 15). Finally, as RAS/MEK/ERK signaling may contribute to tumor growth and is activated in *Atp2a2* mutant cells, we measured phospho-ERK1/ERK2 and total ERK1/ERK2 staining in the tumors and found a significant decrease of the former in tumors from *Notch1*^{-/-} esophagus compared with lesions from *Notch1*^{+/+} epithelium (Fig.8k,l, Supplementary table 15) (Hong et al 2010, Prasad et al 2005). These findings are consistent with signaling downstream of *Atp2a2* mutation, which promotes tumor cell proliferation, being attenuated in tumor cells lacking *Notch1* (Fig.8m).[†]

Finally, in the discussion we address the question as to why *Notch1* loss-of-function mutations have a strong growth advantage in the normal tissue but lead to smaller tumors as follows: Pages 40/41:

‘These results shed light on the disparity in the prevalence of *NOTCH1* mutations in normal esophageal epithelium and tumors. A consequence of normal progenitor behavior is that most clones carrying mutations that do not alter progenitor cell behavior are lost by differentiation, making biallelic loss unlikely. Mutations reducing the function of one *Notch1* allele confer a competitive advantage on mutant progenitors, giving them a greatly enhanced probability of founding persistent, expanding clones. As the heterozygous mutant population grows, the likelihood that the remaining allele will be lost increases. When this happens, it confers a further increase in fitness (**Fig.3h**). By driving wild type cell differentiation, *Notch1* null cells at the clone margins can divide within the homeostatic constraint of cell division being linked to differentiation, resulting in extensive colonization of the epithelium as wild type and heterozygous cells are displaced (**Fig. 2o**). Such a mechanism would explain the selection of the clones with biallelic *NOTCH1* disruption that come to dominate normal human esophagus.

Once an area has been colonized by biallelic *Notch1* mutants, the behavior of mutant progenitors reverts towards that of wild type progenitors so that the epithelium remains histologically normal in both humans and mice. Loss of *Notch1* does impact the transcription of some genes involved in DNA replication, but scRNAseq, cell tracking and histological analyses indicate that cell proliferation and differentiation and epithelial structure and histology are not significantly perturbed. Such reversion towards a normal cell state explains the normal appearance of aged human esophageal epithelium despite *NOTCH1* signaling being disrupted in the majority of the tissue.

In the DEN/Sorafenib generated, *Atp2a2* mutant tumors, proliferating cells are found in multiple layers of a disorganized epithelium, in marked contrast to normal tissue in which proliferating cells are confined to a single basal cell layer. Therefore, the constraint which links cell division to the exit of differentiating cells from the basal cell layer to maintain cellular homeostasis does not operate in the tumors (Frede et al 2016). In the tumor context, the faster cells divide, the faster the lesion will expand. As loss of *Notch1* slows the cell division rate, consistent with disruption of signaling downstream of the *Atp2a2* driver mutation, *Notch1*^{-/-} lesions are smaller than those than wild type tumors (**Fig. 8m**).

Might these findings be of relevance in humans? That over 90% of human ESCC retain one or more wild type copies of *NOTCH1* while arising from an epithelium where the majority of cells have biallelic *NOTCH1* disruption argues wild type *NOTCH1* does promote cancer development. The complexity of genomic alterations in human ESCC may explain the ability of the minority of tumors to bypass the benefit of retaining wild type *NOTCH1* (Consortium 2017).

We conclude that *Notch1* illustrates how inactivating mutations in the same gene can drive clonal expansion in a normal tissue but inhibit tumor growth, by virtue of the distinct changes in cellular dynamics that result from *Notch1* loss in the context of an expanding clone in a wild type normal tissue and in a mutated tumor. Our results further raise the possibility that blockade of NOTCH1 may have therapeutic potential in early esophageal neoplasia by reducing the growth of premalignant tumors. Multiple NOTCH1 inhibitors are in preclinical and clinical development and further investigation of their role in esophageal neoplasia seems warranted.'

2. The overall enthusiasm for the human data set (figure 1) is somewhat tempered by the limited number of patients in the study – albeit the concern is somewhat mitigated by the inclusion of several technical replicates. The authors are to be commended for the attempt at coregistering the imaging data with genomics mutation data. However, the presentation in 1c is very difficult to follow and poorly labelled (i.e. x axis). It is also difficult to draw conclusions without knowledge of intrapatient and interpatient variability. Can conclusive images for high or low notch be magnified and then “coregistered” in panel c with appropriate callouts/highlights etc. Most interesting would be an intrapatient analysis of the same section, with clear annotations of high/low notch correlating with VAF/CNLOH.

In the revised version of the manuscript we have doubled the number of donors from 3 to 6 and the number of micro-dissected and sequenced samples is now increased to 86. We have also revised our presentation of the data following Reviewers 2&3 recommendations (**Fig.1c-f and ED Fig. 1c,d**). We have included a new heatmap showing the NOTCH1 staining, Donor ID, *NOTCH1* mutations types, copy neutral LOH (CNLOH) and the total number of Notch1 mutations in each biopsy (**Fig.1d**). For clarity, *NOTCH1* mutations VAF were summed per effect (indel_splicing, missense or nonsense) are shown in distinct colors and in a stacked bar for each biopsy to visualize the variability of the genomic features for each sample within and between donors.

Our main interest with this nearly clonal sampling was to characterize *NOTCH1* mutant clones in middle aged and elderly patient, and particularly understand the frequency of inactivating biallelic mutations and copy neutral LOH. Consequently, we have now plotted for each donor the proportion of mono and biallelic *NOTCH1* mutations (**Fig.1f**), showing the consistently high proportion of samples with *NOTCH1* biallelic mutations in middle aged and elderly donors. A new chart shows that biopsies with negative *NOTCH1* staining are mainly associated with nonsense and Indel/splicing mutations while missense mutations are enriched in biopsies with *NOTCH1* protein expression (**Fig.1e**).

Finally, as recommended, in order to illustrate the density of mutations within a small piece of esophageal epithelium in elderly donors, we now show the image of one section used for the sampling and co-register the staining and genetic information (mutation, VAF and CNLOH if detected) associated with each biopsies (Fig. 1c).

The mutational calling dataset and dNdS analysis has now been updated in **Supplementary tables 2**.

The following changes have been made in the main text Page 3:

To determine whether this staining pattern reflected *NOTCH1* mutations, we immunostained histological sections from 6 donors, aged 45 to 77 years for *NOTCH1* micro-dissected positive or negative areas and performed targeted sequencing for 322 genes associated with cancer (Fig. 1b). 257 protein altering somatic mutations were identified across 86 samples. Consistent with previous studies, the predominant mutations were in *NOTCH1*, *TP53*, *NOTCH2*. Protein altering mutations in *NOTCH1* represented more than a third of all mutation calls (Martincorena et al 2018, Yokoyama et al 2019) (Supplementary tables 1, 2, Supplementary Note). *NOTCH1* mutations were near clonal with an average variant allele frequency (VAF) of 0.36 and were detected in 80% (69/86) of all biopsies (Fig. 1c,d). 93% (25/27) of negative staining areas detected in five donors carried nonsense, essential splice mutations or indels in *NOTCH1* with copy neutral loss of heterozygosity (CNLOH) of the *NOTCH1* locus (human GRCh37- chr9:139,388,896-139,440,238) or a further mutation, likely to disrupt the second *NOTCH1* allele (Fig. 1d,e). Of the positive staining samples, 59% (35/59) carried a missense *NOTCH1* mutation and most of these had either CNLOH or a second mutation (Fig. 1d,e,f, Extended data Fig. 1c-d, Supplementary Table 2). Overall, the majority (77%, 53/69) of biopsies carrying *NOTCH1* mutations had biallelic alterations, an observation that was consistent across donors (Fig. 1f).

3. In figure 2, three recombination events are shown: Mutant Notch1, YFP+ Mutant Notch1, and YFP Notch+/. What is the frequency of these events where recombination of only YFP or only Notch occur? Previously these single recombination events have been reported to be highly differential depending on the transgene (PMID: 24355609). Interpreting the majority of the lineage tracing data is predicated on being able to identify NOTCH+/, Notch+/-, and Notch-/. It is unclear from the current data if IF alone is a reasonable solution. Additional in situ proof is likely necessary (mRNA FISH etc.).

We agree that recombination events are very variable between genetic loci. In *AhCre^{ERT} Rosa26^{flox}YFP Notch1^{flox} triple transgenic (YFPCreNotch1)* animals, recombination at *Notch1* locus appeared to occur more frequently than at *Rosa26^{flox}YFP* locus but recombination of both loci in the same cells were the rarest events. The recombination rates were dependent on the dose of Tamoxifen injected in the mice, in our clonal experiments we consistently used the same dose of inducing drugs (Beta Naptho Flavone at 80 mg.kg-1 and Tamoxifen at 0.125 mg), This dose was set in order to have enough YFP+ mutant clones detectable but yet avoid clonal fusion too early after induction. For example in clonally induced

YFPCreNotch1 mice, at 4 weeks post-induction, we observed 67% of YFP-; *Notch1*^{-/-} clones; 6% YFP+; *Notch1*^{-/-} clones and 27% of YFP+ *Notch1*^{+/+} clones (Reviewer Fig.6).

We detail these proportions in the methods section ‘Lineage tracing using a YFP reporter’ Page 52: For example, in *YFPCreNotch1*^{flox/flox} induced mice at 4 weeks post-induction, we observed 67±2% of YFP-; *Notch1*^{-/-} clones; 6±1% YFP+; *Notch1*^{-/-} clones and 27±3% of YFP+; *Notch1*^{+/+} clones (Data obtained from 3 mice).

Reviewer Figure 6: Recombination events in clonally induced Notch1^{fllox/fllox} esophageal epithelium
a. YFPCreNotch1^{fllox/fllox} mice were induced at clonal level and 4 weeks later, esophageal epithelium was collected, peeled and stained for NOTCH1 (red), YFP (green) and DNA (Blue). Recombination of YFP is independent of the Notch1 locus so YFP clones may be YFP+;Notch1^{+/+} (blue arrow) or YFP+; Notch1^{-/-} (yellow arrow) in YFPCreNotch1^{fllox/fllox} animals depending on whether Notch1 was recombined. Non YFP labelled Notch1 recombined clones were often detected (orange arrow). Scale bar, 500µm. b. Observed recombination events were counted in three clonally induced YFPCreNotch1^{fllox/fllox} mice, 4 weeks post-induction. Proportions are from n=527, 641 and 535 total events counted from 3 mice.

We agree that accurate discrimination between Notch1^{-/-} vs Notch1^{+/+} and Notch1^{+/-} vs Notch1^{+/+} clones in induced mice is essential for our lineage tracing datasets. As previously shown in **Fig.2a-c**, **now shown in EDFig. 3g-h**, both *Notch1* mRNA and NOTCH1 protein expression are halved or abolished in *Notch1^{+/-}* or *Notch1^{-/-}* keratinocytes respectively.

As mRNA FISH is also staining based and challenging to quantify in a thick tissue wholemount, we have further validated the genotyping *Notch1^{-/-}* and *Notch1^{+/-}* clones based on NOTCH1 staining by using a qPCR based assay to measure the recombination of *Notch1* locus in the genomic DNA of microdissected clones (**new ED Fig.3l-o**). Genomic DNA extracted from PFA fixed tissue is fragmented, so we designed an additional set of primers (set C in **ED Fig. 3c**) to specifically detect the recombined *Notch1* locus, which is validated in **ED Fig. 3e**. In **ED Fig. 3l-n** we show how we were able to isolate clones by use of a micro biopsy punch. The assay confirms that immunostaining is a reliable means to detect *Notch1* mutant clones in comparison to the control areas. Despite DNA fragmentation, we observed an intermediate level of recombination in *+/-* clones compared with *-/-* clones, consistent with the genotyping from immunostaining (**ED Fig.3o**).

We have now updated the figures and text accordingly:

Page 10:

We set up an assay to determine the recombination status of exon 1 of *Notch1* and analyzed wild type and entirely recombined *Notch1*^{+/-} and *Notch1*^{-/-} esophageal epithelium finding both *Notch1* mRNA and protein expression were halved in *Notch1*^{+/-} and abolished in *Notch1*^{-/-} cells compared with wild type keratinocytes (Extended data 3a-h, Supplementary table 4).

We then performed genetic lineage tracing by inducing recombination in scattered single progenitor cells in *YFPCreNotch1*^{+/+}, *YFPCreNotch1*^{+/*fl*ox} or *YFPCreNotch1*^{*fl*ox/*fl*ox} mice. The growth of the YFP expressing clones in each mouse was detected by imaging sheets of epithelium stained for both YFP and NOTCH1 at multiple time points (Fig. 2a). We were able to identify YFP+ *Notch1*^{+/-} clones and YFP+ *Notch1*^{-/-} clones in the respective mice from the intensity of NOTCH1 immunostaining, a method validated by measuring recombination of the *Notch1* allele in micro-dissected clones (Fig. 2b, Extended data Fig. 3i-o).

a

4. Similar to 2 above, overall basal patch size is provided in figure 2 (basal cell number per clone) and similar patches provided in figure 3c. For the clonal experiments it would very important to show whether or not overall clone number changes and how competition proceeds over time (i.e. +/+, +/-, -/- clone numbers over time and clone sizes over time).

We have improved the clarity of the manuscript in presenting the clonal data. The evolution of clone size data with time is now plotted in main Fig. 2c-d, where the number of basal and suprabasal cells per clone in *Notch1*^{+/+}, +/- and -/- clones presented at time points from induction until clone fusion prevented further clonal analysis. Clone fusion is seen beyond 4 weeks in induced *Notch1*^{*fl*ox/*fl*ox} esophagus and beyond 13 weeks in induced *Notch1*^{+/*fl*ox} epithelium (explained in ‘Lineage tracing using a YFP reporter’ paragraph of the methods and in the figure legends). The full data set of cell counts in each clone is given in **Supplementary table 5**. Despite these technical limitations related to our *YFPCreNotch1* model, our genetic lineage tracing dataset was complete enough to implement a model that fitted the clonal size data (**ED Fig.4**) and robustly support the conclusion that *Notch1*^{-/-} clones are fitter than *Notch1*^{+/-} clones that are fitter than wild type clones. These findings are consistent with the 48 hour EdU data discussed above.

In terms of clone density, unlike clone size, this is highly variable between mice, a common feature of Tamoxifen induced *Cre* systems, making estimation of the evolution of clone number unreliable.

5. The EdU experiments (instantaneous labelling – 1hr and pulse-chasing – 48 hrs) are important for understanding how clones grow larger in the *Notch1*^{-/-} epithelium. Given that scRNASeq did not reveal changes in the cell number (see below if more analysis is warranted). If there are no differences in proliferation then there must be differences in the rate of differentiation, thus the importance of the 48 hour chase. The 48 hours timepoint is poorly described. I think the data support the conclusion that the reduction in EdU positive suprabasal -/- is indicative of less differentiation and/or the increase in suprabasal +/+ is indicative of forced differentiation to provide “space” for -/- cells. If this is the case, then do the scRNAseq data support this conclusion? (Fig 2 “l, j, m, n” could all use appropriate figure headings). Please consider clarifying the description of the data and the presentation of the data.

We apologize for the confusion caused by our description of the EdU experiments. To understand the cellular mechanism of underpinning *Notch1* mutant clone growth, we performed EdU labelling assays at 1 and 48 hours post labelling. While the former is familiar the latter is not and we now seek to explain it more clearly.

The proportion of basal cells positive for EdU at 1 hour is a measure of the fraction of cells in S phase. This ratio is not increased in cells within *Notch1* +/- and -/- clones compared to wild type cells distant from mutant clones (**Fig.2e-h**), arguing *Notch1* mutant basal cells do not have an increased proliferation rate.

The 48 hour EdU experiment was performed to track the fate of in S phase cells over the following 48 hour period. After taking up EdU, cells proceed through mitosis and both daughters inherit the EdU label. The pair of labelled cells may remain in the basal layer, or one or both may differentiate and exit the basal layer (**Fig. 2i**). The ratio of the number of EdU labelled suprabasal cells to the total of EdU labelled cells reflects the generation of differentiating cells in the basal layer and their stratification into the suprabasal cell layers.

In *Notch1* +/- and *Notch1*-/- mutants, this ratio is decreased within mutant clones, consistent with a tilt in mutant progenitor cell fate so that a higher proportion of progenitors and fewer differentiating cells are produced per average cell division (**Fig.2k**). Moreover, with *Notch1*-/- clones we also see an increase in the suprabasal EdU+: total EdU + ratio in the *wild type* cells at the clone margin compared with wild type cells further from the mutant clones. This, indicates that wild type cells next to clones differentiate and stratify at an increased rate (**Fig.2j,l**).

Significantly, basal cell density (cells/unit area) is the same in mutant clones and wild type cells (**Fig.2m,n**). To maintain a constant cell density, cell division is linked with the exit of a nearby cell from the basal layer (Mesa et al 2018). Hence, driving differentiation and stratification of wild type cells at the clone edge allows *Notch1*-/- cell division, further increasing the proportion of mutant progenitors (**Fig. 2o**).

Such 'supercompetition' by inducing differentiation of adjacent wild type cells contributes to the enhanced fitness of *Notch1* null over *Notch1* heterozygous cells (**Fig. 3h, ED Fig. 4**). Previous studies by ourselves and others have seen a similar induction of keratinocyte differentiation adjacent to Notch pathway mutants (Alcolea et al 2014, Lowell et al 2000).

We have now clarified the account of the 48 hour EdU cell tracing assay and the mechanism of *Notch1* mutant clonal expansion in Fig2 and in the text as follows:

Page 11:

We first counted the proportion of basal cells positive for EdU at 1 hour after labelling, which measures the fraction of basal cells in S phase, to assess whether an increase in the rate of cell division contributed to mutant clone expansion (Fig. 2e,f). This value was similar for cells within *Notch1*^{+/-} clones and wild type cells distant from clones (Fig. 2g). Within *Notch1*^{-/-} mutant clones

the proportion of EdU positive basal cells was marginally lower than in wild type cells (Fig. 2h). We conclude neither *Notch1*^{+/-} nor *Notch1*^{-/-} clonal expansion results from an increase in mutant cell division rate compared with wild type cells.

The 48 hour EdU experiment labels S phase cells and tracks the fate of the two cells generated by the subsequent mitosis over the following 48 hours. The pair of labelled cells may remain in the basal layer, or one or both may differentiate and exit the basal layer (Fig. 2i). The ratio of EdU labelled suprabasal cells to the total EdU labelled cells reflects the rate of production of differentiating cells in the basal layer and their stratification into the suprabasal cell layers. In *Notch1*^{+/-} and *Notch1*^{-/-} clones, this ratio is decreased, consistent with a tilt in mutant progenitor cell fate so that more progenitor and fewer differentiating daughters are produced per average cell division (Fig. 2k). Strikingly, adjacent to *Notch1*^{-/-} clones there was an increase in the suprabasal EdU+: total EdU+ cell ratio in the *wild type* cells at the clone margin compared with wild type cells further from the mutant clone (Fig. 2j,l). This, along with a small decrease in the proportion of wild type S phase cells at the clone edge, indicates that wild type cells adjacent to the clone exit the cell cycle, differentiate and stratify out of the basal layer at an increased rate, a phenomenon also reported in previous studies of Notch inhibited keratinocytes interacting with wild type cells (Fig. 2h)(Alcolea et al 2014, Lowell et al 2000). These observations explain the increased fitness of *Notch1*^{-/-} over *Notch1*^{+/-} clones. Cell density was similar in both mutant genotypes and wild type areas suggesting that the linkage between cell division and the exit of a nearby differentiating cell from the basal layer is maintained (Fig. 2m,n). Within this constraint, the driving of wild type cell differentiation and stratification permits *Notch1*^{-/-} cell division at the clone edge, accelerating clonal expansion (Fig. 2o).'

If this is the case, then do the scRNAseq data support this conclusion?

The sc-RNAseq assay is not informative on the mechanism of *Notch1* clonal expansion because it was performed on epithelium that had been fully colonized by *Notch1*^{-/-} cells vs wild type tissue. The aim of this assay is to understand the long term consequences of losing *Notch1* on tissue homeostasis as detailed on our answer to point 6. We have now clarified the context of this assay in the main text as follows: Page 24:

'To further characterize the phenotype of fully colonized *Notch1*^{-/-} epithelium at cellular level we performed single cell RNA sequencing (scRNA-seq). '

6. Figure 4 genomics data is not overly helpful and could be moved to the supplement. The scRNAseq is really underutilized, particularly understated is the potential for difference in cell states between the

genotypes. Was a pseudotime trajectory analysis performed / could it be useful to understand how homeostasis is maintained in *Notch1*^{-/-} ?

We thank the reviewer for the opportunity to discuss the single cell RNAseq data further.

As stated in response to point 5, the sc-RNAseq was indeed performed to understand how homeostasis is maintained in the epithelium covered by *Notch1*^{-/-} keratinocytes. As shown in the first version of the manuscript (**previous ED Fig.6d, new Fig.4d**), upon constructing the original UMAP space that integrated knockout (KO) and wild type (WT) keratinocytes we noticed there was no systematic 'shift' where KO cells separated from WT cells, resulting in distinct structural differences observable in the UMAP space. We observed that well established basal cell markers systematically dropped in expression in the same area of the UMAP space where differentiation markers increased. To capture the boundary between basal and differentiating cells, we set out a 'line' in this area separating putative basal cells and suprabasal cells and analyzed the proportion of cells per library (**previous Fig.4.i and ED Fig.6e; new Fig.4e, new ED Fig.6m-p**). Based on this method we concluded that there is no significant difference in basal and suprabasal cellular composition between *Notch1*^{-/-} and *Notch1*^{+/+} epithelium.

In order to go further and compare the keratinocytes cells state between genotypes we have now applied a second method based on cell clustering using esophageal cell state markers from a recent study scRNAseq in mouse esophagus (**new Fig.4f-h**) (McGinn et al 2021). Cell clustering was calculated and each established cluster was associated to a cell state by plotting a heatmap showing the expression of cell states markers (**new Fig.4.f-g**). As in (McGinn et al 2021), we could place cells in to the following 3 compartments: cycling basal cells → clusters 8,6,9 and 4; resting basal cells → clusters 10, 0, 1 and 11 and differentiating cells → clusters 3, 5, 2 and 7. We were then able to calculate the proportion of cells per cluster for each library and the proportion of cells in cell state (**new Fig.4h**). There was no substantial difference in the proportion of cells in each compartment. As this approach is dependent upon cell clustering, we applied Milo that looks for differential cell density between experimental conditions without using the cell clusters (**Supplementary Note, Reviewer Figure 7**) (Dann et al 2022). Milo constructs groups of cells by integrating cell-to-cell distances and subsequently tests each of these groups (termed neighborhoods) for a significant difference between knock-out and wild type cells. **Reviewer figure 7** shows the location of these neighborhoods (circles) and that none have been found significant (significant circles would be colored differently from white), leading to the conclusion that the existing UMAP space does not show a significant 'shift' in cells in the knockout samples.

Reviewer Figure 7: UMAP plots of keratinocytes from all libraries show constructed neighborhoods (circles) from Milo which tests for differential cell density between *Notch1*^{+/+} and *Notch1*^{-/-} conditions (Dann et al 2022). The absence of colored circles show that there is no significant difference.

Beyond the analysis presented in the manuscript we also performed RNA velocity analysis with Velocyto (La Manno et al 2018) (**Reviewer Fig.8**) and trajectory analysis using Monocle (**Reviewer Fig.9**) (Trapnell et al 2014).

We use RNA velocity analysis to explore whether cells perhaps `move` differently within the shape, for example that cells commit to differentiation in a different way in the knock-out tissue. If cells do this, then perhaps this would show up as a different RNA velocity trajectory with arrows pointing in a different direction in the two genotypes. We therefore inferred RNA velocity using Velocyto, **Reviewer Fig. 8a** shows the inferred velocity for the +/+ cells, **Reviewer Fig. 8b** for the -/-. Both figures are largely identical, suggesting there is no substantial difference in the trajectory that cells take across the established shape.

The downside of Velocyto is that it does not provide a differential trajectory analysis. We wondered whether perhaps there was a still more subtle difference in the expressed genes between the experimental conditions. We therefore performed trajectory analysis using Monocle, which, after processing the raw counts, applying scaling and batch effect adjustment, infers a trajectory (**Reviewer Fig. 9a**) and subsequently infers groups of differentially expressed genes along it (**Reviewer Fig.9b**).

The results show that Monocle is able to roughly capture the trajectory of the expected two different states: Cycling and differentiating cells. Most striking however is the differential expression of the gene groups shown in **Reviewer Fig. 9b** cluster +/+ 1 (labelled CTRL 1) / -/- 1 and +/+ 2 / -/- 2 together, suggesting that the differential expression along the trajectory is dominated by library-to-library differences, even after batch effect correction, and is not showing any meaningful signal relating to differences in *Notch1* genotype.

We conclude that all of these analyses show essentially the same result as was presented in the manuscript: there is no observable difference between *Notch1*^{-/-} and *Notch1*^{+/+} keratinocytes in these single cell RNAseq data. And this is consistent with orthogonal experimental work throughout the manuscript.

We comment on our deeper analysis of single cell RNAseq data as follows: Pages 24/25:

We used the expression level of these markers to discriminate basal and suprabasal cells in UMAP space (Supplementary Note). The analysis revealed similar proportions of basal and suprabasal keratinocytes in the control and *Notch1*^{-/-} tissue (Fig. 4e). We conducted a more detailed analysis using keratinocytes states defined in a previous study (McGinn et al 2021). Cell clustering was used to assign cells to one of three cell states defined by the expression of cell state markers: cycling basal, resting basal or differentiating cells (Fig4.f-g). The proportion of cells per cluster for each library and the proportion of cells in cell state was calculated. There were no substantial differences between genotypes (Fig.4h, Supplementary Note).

Reviewer Figure 8: Results from RNA velocity analysis showing wild type cells (A) and knock out cells (B).

Reviewer Figure 9: Monocle analysis of scRNAseq data

A: Inferred trajectory from processed raw counts, after applying scaling and batch effect adjustment
 B: Groups of differentially expressed genes along the trajectory shown in A.

- 'Does this data support the overall lineage training / EdU analysis conclusions?'

The scRNAseq data indicates that once epithelium has been completely colonized by *Notch1*^{-/-} cells their phenotype reverts from clonal expansion towards that of wild type homeostatic normal epithelium. The genetic lineage tracing data in **Fig. 2** describes the phase of rapid clonal expansion of *Notch1* mutant clones, which is different from the stage at which scRNAseq was performed.

This conclusion is consistent with data from the experiment shown in **Fig. 5** in which mutant animals were aged. The thickness of the epithelium, density of basal cells, percentage of Ki67 positive basal cells, 1 hour EdU positive basal cells and proportion of EdU positive cells that had stratified after 48 hours are not significantly different in wild type and *Notch1*^{-/-} aged epithelium. These mouse results are also consistent with findings in human epithelium lacking NICD expression (**ED Fig. 1f-h**).

To further validate the sc-RNAseq findings, we have now performed an additional EdU/BrdU cell tracing experiment in epithelium fully colonized by *Notch1*^{-/-} or compared with wild type, in mice of similar age to those used in for the sc-RNAseq experiment (**ED Fig. 6q-w, Supplementary Note**). This experiment consisted of injecting EdU48h and BrdU 1h before tissue collection. Tissues were collected and stained

for EdU (48 hour tracked cells), BrdU (1 hour assay of S-phase cells) and phospho-Histone H3 (pHH3, a G2/M phase marker). The ratio of suprabasal EdU +ve: total EdU+ve cells, reflecting the generation of differentiating cells and their stratification, the ratio of EdU+ BrdU+ basal cells and the ratio of EdU+ pHH3+ basal cells reflecting the generation of proliferating basal cells were not significantly different between wild type and *Notch1*^{-/-} epithelium. Moreover, the overall proportion of BrdU positive basal cells indicating cells in S Phase and the overall percentage of pHH3+ve, BrdU-ve mitotic basal cells were not significantly different in wild type and *Notch1*^{-/-} epithelium, consistent with the scRNA seq findings (ED Fig.6s-w).

We comment on these findings in the main text as follows:

Page 25:

We next performed a cell tracking assay to validate the scRNAseq findings. Mice with *Notch1*^{-/-} esophageal epithelium and littermate controls, were injected EdU and BrdU at 48h and 1h respectively before tissue collection (Extended data Fig.6q). Staining for EdU revealed the fate of S phase cells over the following 48 hours, BrdU positive cells were currently in S-phase and phospho-Histone H3 pHH3), is a G2/M phase marker (Extended data Fig. 6r). The ratio of suprabasal EdU +ve: total EdU+ve cells, reflecting the generation of differentiating cells and their stratification, the proportion of BrdU positive basal cells and the percentage of pHH3 +ve, BrdU-ve basal cells were all similar in wild type and *Notch1*^{-/-} epithelium, consistent with the scRNA-seq findings (Extended data Fig.6s-w,

Supplementary table 11, Supplementary Note).

As *Notch1* biallelic mutations cover large portions of the epithelium in elderly Human, we examined the epithelium in induced *YFP-CreNotch1*^{flox/flox} mice and control littermates that were aged for 52 weeks. Tissue thickness, basal cell density and expression of the differentiation markers KRT14, KRT4 and LOR and the proliferation marker Ki67 were similar in both genotypes, (Fig.5a-d, Supplementary table 12). Pulse labelling and short-term lineage tracing for 48 hours with EdU confirmed no significant difference in the proportion of S phase cells or in the stratification of differentiating cells respectively between *Notch1*^{-/-} and wild type esophagus (Fig.5 e-h).

,

(

- 'Figure 4 genomics data is not overly helpful and could be moved to the supplement.'

As suggested, we have now moved the bulk RNAseq data to **ED Fig.6a-e**. The volcano plots have been removed and replaced by a chart showing the top GO biological processes up and down regulated in *Notch1*^{-/-} epithelium vs wild type and we now state:

Page 24:

- 'Gene set enrichment analysis showed that transcripts of genes involved in DNA replication were downregulated in *Notch1*^{-/-} colonized epithelium (**Extended data Fig. 6e, Supplementary table 9**). '

Minor Comments

VAF1 and VAF2 are not well defined, how do these relate to one another? Which mutation is being analyzed?

In the original version of the paper, VAF1 and 2 were the Variant allele frequency (VAF) of the two largest *NOTCH1* mutations in each biopsy (the vast majority of biopsies only had one or two *NOTCH1* mutations). However, the original plot is now replaced by a clearer heatmap (**new Fig. 1d**), as described in our response to point 2.

In Extended Data 3g, the text remarks that *Notch*^{+/-} could be distinguished from *Notch*^{-/-} based on intensity of the signal. An inset demonstrating the clear difference in intensity would be helpful to boost the confidence in these results.

The imaging is rather difficult to interpret due to the size of the presented images, consider revising accordingly.

We regret that the figures in the first version and their explanation were insufficiently clear. First, we would like to clarify that the *Notch1*^{+/-} and *Notch1*^{-/-} clones were induced in different mice: *Notch1*^{+/-} clones were induced and analyzed in *Notch1*^{+/*fl*ox} mice; *Notch1*^{-/-} clones were induced and analyzed in *Notch1*^{*fl*ox/*fl*ox} mice. In both mouse backgrounds, mutant clones were to be distinguished from wild type cells by the combination of NOTCH1 and YFP staining.

We have clarified this point in the main text:

Page 10:

We then performed genetic lineage tracing by inducing recombination in scattered single progenitor cells in *YFPCreNotch1^{+/+}*, *YFPCreNotch1^{+/-}* or *YFPCreNotch1^{flox/flox}* mice. The growth of the YFP expressing clones in each mouse was detected by imaging sheets of epithelium stained for both YFP and NOTCH1 at multiple time points (Fig. 2a). We were able to identify YFP+ *Notch1^{+/-}* clones and YFP+ *Notch1^{-/-}* clones in the respective mice from the intensity of NOTCH1 immunostaining, a method validated by measuring recombination of the *Notch1* allele in micro-dissected clones (Fig. 2b, Extended data Fig. 3i-o).'

In new ED Fig.3 m-n we show new pictures of clonally induced *Notch1^{-/-}* and *Notch1^{+/-}* clones with magnified images showing no specific signal in the knock-out clone while the heterozygous clone still expresses NOTCH1, at lower intensity than in surrounding the wild type cells. We also have enlarged the images of *+/+*, *+/-* and *-/-* clones in Fig. 2b.

To render imaging clearer, we have also enlarged images or changed for clearer ones throughout the rest of the manuscript when possible.

In the Tamoxifen experiments for Figure 5 *Notch^{flox/flox}* mice were injected with a quarter of the dose of the *Notch^{+/+}* and *Notch^{+/-}* mice. This lower induction rate is then compared to a higher induction rate to demonstrate no differences in tissue morphology and function.

Fig.5 is an aging experiment where mice were induced and aged for 52 weeks. A lower dose of Tamoxifen was used for the *YFPCreNotch1^{flox/flox}* mice in order to minimize the recombination of the *Notch1* allele in the corneal epithelium, which may lead to corneal opacification and keratinization. This was seen in using high dose Tamoxifen in older *YFPCreNotch1^{flox/flox}* mice so the low dose was used in this experiment to avoid morbidity in the animals.

This is precised in 'Aging experiments' methods section as follows: Page 53:

'A lower dose of Tamoxifen was used for the *YFPCreNotch1^{flox/flox}* mice in order to minimize the recombination of the *Notch1* allele in the corneal epithelium, possibly leading to corneal opacification and keratinization (Movahedan et al 2013).'

At this lower dose of Tamoxifen (0.25mg), induced *Notch1*^{-/-} cells were able to cover the full epithelium within 2 to 3 months, while *Notch1*^{+/-} cells needed a higher dose (1mg) to reach similar coverage within this period.

We have now added an additional experiment that shows no significant difference in cell proliferation or stratification between *Notch1*^{-/-} and wildtype epithelium irrespective of whether a high dose induction protocol is used and animals aged for 2 or 3 months weeks or (**Extended data Fig. 6q-w**) or a low level induction is used and the tissue is aged for a year (**Fig 5**).

The genetic nomenclature for the mice is very difficult to follow and sometimes not uniform throughout the legends.

We apologize for the lack of clarity on this point.

We now uniformly use the short name *YFPCreNotch1* for the triple transgenic mice as presented in page 8 of the main text and in **Extended data Figure 2**. When naming mice, we precise the genotype as follows: *YFPCreNotch1*^{+/+}; *YFPCreNotch1*^{+/*flox*} or *YFPCreNotch1*^{*flox*/*flox*} mice. When indicating the genotype of cells, clones or tissue, we only give the status of the *Notch1* allele using *Notch1*^{+/+}, *Notch1*^{+/-}, *Notch1*^{-/-} (or ^{+/+}, ^{+/-} and ^{-/-} in figures). Throughout the manuscript, non-induced tissue or non-induced part of tissues express wild type *Notch1* transcript and therefore are named ^{+/+} by extension. In clonally induced tissue, we use ^{+/*flox*} or ^{*flox*/*flox*} when explaining the distinction between unrecombined areas (called ^{+/+} by extension) and induced clones (^{+/-} or ^{-/-}) within the same esophagus (**Fig. 2, Extended data Fig.3**).

Figure 7i denotes 16 weeks of antibody treatment, but the text reports 6 weeks of antibody treatment.

The start of the antibody injections was at 9 weeks post-carcinogen exposure. This is now detailed on the schematic so it is now clear that the treatment was performed between 9 and 16 weeks after carcinogen exposure and therefore lasted for 6 weeks as indicated in the legends and in the methods.

The manuscript is well-written but has a few Grammatical errors:

- ED Figure 3a legend: allowiNotch1
- ED Figure 3f: non uniform italics used
- No callout for Figure 4b in text
- Page 31: "short-term analysis the revealed"
- "recombination of in..." poor grammar p7
- Figure 2b, add figure label for Notch mRNA
- "excision of p 9"

We are most grateful to the reviewer for highlighting these errors which have been corrected.

Reviewer #3: See responses to Reviewer #2.

Remarks to the Author:

Comments to the Authors:

This manuscript elegantly describes the conflicting action of Notch1 mutation in normal and tumorigenic esophageal epithelium. In aging human esophagus, sequencing reveals that Notch1 mutant clones which occupy a significant proportion of the epithelium carry biallelic alterations. To test the fitness impacts of Notch mutant clones, the authors developed a conditional transgenic mouse where one or both alleles of Notch could be deleted and induce a fluorescent lineage reporter. The authors used lineage tracing and Notch staining in this transgenic mouse to quantify the fitness advantage that the loss of one allele confers over wild type epithelium. This fitness advantage was further magnified by the loss of the second allele. Sequencing of mutant clones revealed slight differences in transcription. However, the authors reported no functional difference in the morphological analysis of the esophageal epithelium. Finally, the authors induced tumors in this mouse model and found that Notch mutants were less numerous in tumors compared to the normal adjacent epithelium. Complete loss of Notch1 and antibody blockade of Notch1 resulted in significantly smaller tumors, but no change in tumor number. The authors are to be commended for an exhaustive analysis of Notch1 mutations in human patient specimens and their application of an informative mouse model for studying Notch1 clonal competition and its effects on tumorigenesis. The summary is clear and overall the manuscript is well written. An appropriate review of the literature is included. However, the major question in the study is to determine how Notch1^{-/-} clones spread throughout the esophagus yet are rarely found in tumors. Interestingly, the group's 2020 Nature Genetics paper describes a very elegant model in which cell fitness is relative to neighboring clones (as cartooned in Figure 6f, 2020). From this perspective Notch1 mutations could inhibit tumorigenesis through two likely mechanisms, Notch1^{-/-} could intrinsically inhibit tumorigenic potential or Notch1^{-/-} could extrinsically inhibit tumorigenic potential. The current manuscript seems to suggest an extrinsic mechanism whereby increased cell fitness of Notch1^{-/-} cells prevents the growth of tumors. This is a very attractive hypothesis and extremely exciting. If proven, it would be a significant contribution to the field and warrant reconsideration. This concern is the most important to address and listed as priority 1 below. Several additional concerns are also provided – that are likely more easily rectified through textual revision and/or additional analysis/reanalysis of available data.

Major Experimental Concerns

1. The current data/experiments do not address the paradoxical abundance of human Notch1^{+/+} esophageal cancers despite the field/clonal expansion of Notch1^{-/-} in aged epithelium. The tumor modeling using the high induction model seems to test the intrinsic hypothesis because most cells will be Notch1^{-/-} and receive secondary hits in the same cell type. However, the extrinsic hypothesis is left

untested. Extrinsic forces of Notch1^{-/-} super-fit cells could inhibit neighboring mutant cells and reduce deterministic tumor outcomes and similar again to previous competition experiments in the esophagus (<https://www.nature.com/articles/s41588-020-0624-3> [nature.com]). This is a very interesting concept, particularly given the Adaptive Oncogenesis model proposed by DeGregori or Winton's competition measurements in the colon.

A very important experiment will be to induce infrequent clonal Notch1^{-/-} mutations, then chemically incite tumors. This would determine whether tumor incidence decreases in a mosaic epithelium with Notch ^{+/+}, ^{+/-}, and ^{-/-} clones already present. In this manner, Notch^{-/-} clones would be placed in direct competition with tumorigenic clones – a more naturalistic model of the human condition. This would also enable the direct demonstration that somatic mutations capable of driving clonal expansions could act as tumor suppressors.

2. The overall enthusiasm for the human data set (figure 1) is somewhat tempered by the limited number of patients in the study – albeit the concern is somewhat mitigated by the inclusion of several technical replicates. The authors are to be commended for the attempt at coregistering the imaging data with genomics mutation data. However, the presentation in 1c is very difficult to follow and poorly labelled (i.e. x axis). It is also difficult to draw conclusions without knowledge of intrapatient and interpatient variability. Can conclusive images for high or low notch be magnified and then “coregistered” in panel c with appropriate callouts/highlights etc. Most interesting would be an intrapatient analysis of the same section, with clear annotations of high/low notch correlating with VAF/CNLOH.

3. In figure 2, three recombination events are shown: Mutant Notch1, YFP+ Mutant Notch1, and YFP Notch^{+/+}. What is the frequency of these events where recombination of only YFP or only Notch occur? Previously these single recombination events have been reported to be highly differential depending on the transgene (PMID: 24355609). Interpreting the majority of the lineage tracing data is predicated on being able to identify NOTCH^{+/+}, Notch^{+/-}, and Notch^{-/-}. It is unclear from the current data if IF alone is a reasonable solution. Additional in situ proof is likely necessary (mRNA FISH etc.).

4. Similar to 2 above, overall basal patch size is provided in figure 2 (basal cell number per clone) and similar patches provided in figure 3c. For the clonal experiments it would very important to show whether or not overall clone number changes and how competition proceeds over time (i.e. ^{+/+}, ^{+/-}, ^{-/-} clone numbers over time and clone sizes over time).

5. The EdU experiments (instantaneous labelling – 1hr and pulse-chasing – 48 hrs) are important for understanding how clones grow larger in the Notch1^{-/-} epithelium. Given that scRNASeq did not reveal changes in the cell number (see below if more analysis is warranted). If there are no differences in proliferation then there must be differences in the rate of differentiation, thus the importance of the 48 hour chase. The 48 hours timepoint is poorly described. I think the data support the conclusion that the reduction in EdU positive suprabasal ^{-/-} is indicative of less differentiation and/or the increase in

suprbasal +/- is indicative of forced differentiation to provide “space” for -/- cells. If this is the case, then do the scRNAseq data support this conclusion? (Fig 2 “l, j, m, n” could all use appropriate figure headings). Please consider clarifying the description of the data and the presentation of the data.

6. Figure 4 genomics data is not overly helpful and could be moved to the supplement. The scRNAseq is really underutilized, particularly understated is the potential for difference in cell states between the genotypes. Was a pseudotime trajectory analysis performed / could it be useful to understand how homeostasis is maintained in Notch1 -/- ? Does this data support the overall lineage training / EdU analysis conclusions?

Minor Comments

VAF1 and VAF2 are not well defined, how do these relate to one another? Which mutation is being analyzed?

In Extended Data 3g, the text remarks that Notch+/- could be distinguished from Notch-/- based on intensity of the signal. An inset demonstrating the clear difference in intensity would be helpful to boost the confidence in these results.

The imaging is rather difficult to interpret due to the size of the presented images, consider revising accordingly.

In the Tamoxifen experiments for Figure 5 Notchflox/flox mice were injected with a quarter of the dose of the Notch+/+ and Notch+/flox mice. This lower induction rate is then compared to a higher induction rate to demonstrate no differences in tissue morphology and function.

The genetic nomenclature for the mice is very difficult to follow and sometimes not uniform throughout the legends.

Figure 7i denotes 16 weeks of antibody treatment, but the text reports 6 weeks of antibody treatment.

The manuscript is well-written but has a few Grammatical errors:

- ED Figure 3a legend: allowiNotch1
- ED Figure 3f: non uniform italics used
- No callout for Figure 4b in text
- Page 31: “short-term analysis the revealed”
- “recombination of in...” poor grammar p7
- Figure 2b, add figure label for Notch mRNA
- “excision of p 9”

Reviewer #4:

Remarks to the Author:

The manuscript by Abby et al describes a comprehensive work focuses on understanding how Notch1 mutations drive clonal expansion in normal esophageal epithelium but (somewhat surprisingly) impairs tumor growth. The work encompasses remarkably wide set assays ranging from sequencing of tissues from human donors, clonal analysis via imaging and sequencing of mouse models, and comparative analysis of tumors vs healthy tissues. These provide a clear understanding of a highly intriguing phenomenon: That large fraction of the normal esophageal epithelium accumulates mutations in Notch1 as we age. These mutations in Notch1 promote clonal expansion allowing mutant clones to take over neighboring wildtype tissue. The growth advantage is conferred by Notch1 heterozygous mutants (N1+/-) and even more so by homozygous mutants (N1-/-). A computational model based on analysis of clone size distribution largely captures the observed dynamics. Surprisingly, these Notch1 mutants do not affect the normal tissue morphology or characteristics (but do show specific changes in gene expression). Moreover, experiments inducing tumors in the tissue showed that tumors are less likely to accumulate Notch1 mutations compared to normal tissue and that deletion of Notch1 or inhibiting it using anti-Notch1 blocking antibody reduces tumor growth. This suggests a potential treatment path for esophagus cancer. Additional analysis such as identifying the specific types of Notch1 mutations (either in the ligand binding domain or in the NRR domain) and their effect on Notch activation, and quantitative analysis of growth rates in clones provide further support for the conclusions and details on the mechanisms the behind enhanced clonal expansion.

The manuscript is written in an exceptionally clear manner, and the data is well presented and overall convincing. While there are a few questions that require clarifications, I do not have any major issues with the scientific claims and the interpretation of the experimental results. I therefore recommend the manuscript for publication in Nature genetics after addressing the small concerns below.

We are most grateful to the reviewer for their careful reading and positive assessment of the manuscript.

Minor issues:

1. I am a little confused by the results of the EDU experiments shown in Figure 2g-n. I understand why the short EDU experiments show that proliferation in the basal levels is not affected by Notch1 mutation. This means that the differences observed in the longer EDU experiment are due to differences in relative turnover rates, namely, when competing Notch1 mutant and WT cells, the latter turn over faster. How does this fit the clonal expansion model?

We apologize for the lack of clarity in the explanation of the 1 hour and 48 hour EdU experiments in the first version of the paper. We now include additional explanation of these protocols (new Fig. 2e and Fig. 2i). The 48 hour EdU experiment labels S phase cells and tracks the fate of the daughter cells generated at the following mitosis over the following 48 hours. The pair of labelled cells may remain in the basal

layer, or one or both may differentiate and exit the basal layer (**Fig. 2i**). The ratio of the EdU labelled suprabasal cells to the total EdU labelled cells reflects the production of differentiating cells in the basal layer and their stratification into the suprabasal cell layers. In *Notch1*^{+/-} and *Notch1*^{-/-} clones, there is no increase in the proportion of S phase cells as assessed by 1 hour EdU labelling, but the proportion of EdU positive suprabasal cells 48h after labelling is significantly lower than in wild type areas (**Fig 2k,l**). This is consistent with mutant cells generating a lower proportion of differentiating than progenitor cell daughters as a mechanism of clonal expansion within normal epithelium.

Also, how do the results showing higher EDU positive cells at the WT boundary support the clonal expansion model? Moreover, since it is later claimed that *Notch1*^{-/-} is more efficient in clonal expansion than *Notch1*^{+/-}, shouldn't you expect to see even lower EDU levels in the former?

In normal epithelium, the density of cells in the basal layer is constant, cell division is linked to stratification of nearby differentiating cells (Mesa et al 2018). Basal cell density (cells/area) is similar in wild type epithelium and in expanding *Notch1*^{+/-} and *-/-* clones (**Fig.2. m-n**). This constraint on proliferation of both wild type and mutant cells and is why mutants gain a proliferative advantage through generating progressively more progenitor daughters rather than dividing at a faster rate. The increased wild type EdU+ suprabasal cells at the edge of *Notch1*^{-/-} clones in the 48 hour EdU cell tracking assay indicates that the differentiation and stratification of wild type cells in contact with *Notch1* null cells is increased, consistent with other studies of *Notch* pathway mutants (**Fig. 2l, j**) (Alcolea et al 2014, Lowell et al 2000). The exit of wild type cells from the basal layer at the mutant clone edge accelerates *Notch1*^{-/-} clone expansion by allowing mutant cell division.

We now explain these issues in the main text as follows.

Page 11:

We first counted the proportion of basal cells positive for EdU at 1 hour after labelling, which measures the fraction of basal cells in S phase, to assess whether an increase in the rate of cell division contributed to mutant clone expansion (Fig. 2e,f). This value was similar for cells within *Notch1*^{+/-} clones and wild type cells distant from clones (Fig. 2g). Within *Notch1*^{-/-} mutant clones the proportion of EdU positive basal cells was marginally lower than in wild type cells (Fig. 2h). We conclude neither *Notch1*^{+/-} nor *Notch1*^{-/-} clonal expansion results from an increase in mutant cell division rate compared with wild type cells.

The 48 hour EdU experiment labels S phase cells and tracks the fate of the two cells generated by the subsequent mitosis over the following 48 hours. The pair of labelled cells may remain in the basal layer, or one or both may differentiate and exit the basal layer (Fig. 2i). The ratio of EdU labelled suprabasal cells to the total EdU labelled cells reflects the rate of production of differentiating cells in the basal layer and their stratification into the suprabasal cell layers. In *Notch1*^{+/-} and *Notch1*^{-/-} clones, this ratio is decreased, consistent with a tilt in mutant progenitor cell fate so that more progenitor and fewer differentiating daughters are produced per average cell division (Fig. 2k). Strikingly, adjacent to *Notch1*^{-/-} clones there was an increase in the suprabasal EdU+: total EdU+ cell ratio in the *wild type* cells at the clone margin compared with wild type cells further from the mutant clone (Fig. 2j,l). This, along with a small decrease in the proportion of wild type S phase cells at the clone edge, indicates that wild type cells adjacent to the clone exit the cell cycle, differentiate and stratify out of the basal layer at an increased rate, a phenomenon also reported in previous studies of Notch inhibited keratinocytes interacting with wild type cells (Fig. 2h)(Alcolea et al 2014, Lowell et al 2000). These observations explain the increased fitness of *Notch1*^{-/-} over *Notch1*^{+/-} clones. Cell density was similar in both mutant genotypes and wild type areas suggesting that the linkage between cell division and the exit of a nearby differentiating cell from the basal layer is maintained (Fig. 2m,n). Within this constraint, the driving of wild type cell differentiation and stratification permits *Notch1*^{-/-} cell division at the clone edge, accelerating clonal expansion (Fig. 2o).

-'. How does this fit the clonal expansion model?'

The clonal expansion model used is based on models used in a previous paper (Colom et al 2020), which demonstrated that clones in the mouse esophagus (largely *Notch1* mutant clones) competed in a spatial manner, with local competition between neighbors determining cell fate. The 2D Wright–Fisher model used in this paper is essentially a more computationally efficient version of the 2D Moran model used in (Colom et al 2020). This computational efficiency allowed the fitting of the model to the clone size data.

In the model, the fitness of a cell relative to its neighbors determines the cell fate. The fitter the cells, the more likely they are to divide, and the less likely they are to differentiate. Hence the finding that the *Notch1* mutant cells in the model are fitter than the WT cells is consistent with the EdU data showing that the WT cells on the edge of the *Notch1* clones have a biased cell fate towards differentiation/stratification and the *Notch1* mutant cells have a bias towards renewal. The larger EdU imbalance between the *Notch1*^{-/-} clones and WT cells is consistent with the larger fitness difference between these genotypes found by the model.

We have added the following paragraph to the supplementary note to explain this:

Page 63:

“The fitting results are consistent with the EdU 48h data shown in Fig. 2 k,l. In the model, interaction between highly fit mutant cells and adjacent wild type cells biases the wild type cells towards differentiation and the mutant cells towards division. The higher the fitness imbalance, the stronger the fate bias. The stronger differentiation bias in the wild type cells adjacent to *Notch1*^{-/-} clones (Fig.2l) is therefore consistent with the higher fitness inferred for the *Notch1*^{-/-} clones.”

2. In the computational model you estimate the fitness advantage of the *Notch1*^{+/-} and *Notch1*^{-/-} cells over WT. First, it is unclear why in ED Figure 4b, the best fit for Fitness are obtained only in specific ranges of induction levels. Since the induction ranges for *N1*^{+/-} and *N1*^{-/-} do not overlap how can a single preferred parameter set be deduced? (I'm probably just missing some explanation).

At later stages of the lineage tracing experiments the clones collide. The model (supported by the EdU data and previous studies) is based on local competition between cells – with cell fate decisions based on the difference in fitness between neighbors (Colom et al 2020). Therefore, if a *Notch1* mutant cell is surrounded by other *Notch1* mutant cells, it will have a different cell fate than if it were surrounded by the less fit WT cells.

The induction rate determines how many *Notch1* mutant clones we start with. At early time points, the induction rate makes little difference because the clones are rarely colliding, and the fitness difference between the *Notch1* cells and the WT cells is the key parameter for determining clonal expansion. At later time points, the growth of mutant clones will be slowed by any other mutant clones they encounter. The more the clones collide, the slower the expansion and the smaller the clone sizes. The inferred induction of the *Notch1*^{+/-} clones is higher than for the *Notch1*^{-/-} because at later time points the *Notch1*^{+/-} expansion slows more (relative to what would occur without any clonal collision). This can be seen by comparing the expansion curves in **ED Fig. 4**.

We discuss in the supplementary text that this is likely a deviation of the true clonal expansion from the simple model we have used. The tissue in the *Notch1*^{+/-} experiment is not as heavily mutated as the model infers, meaning that some of the slowing of the clonal expansion of *Notch1*^{+/-} clones observed is due to some factor other than clonal collision. We have observed behavior like this with *Trp53* mutant clones expanding in mouse skin – the initial fast expansion slowed at later timepoints, and in that case it was not due to clonal collision with other *Trp53* mutant cells (Murai et al 2018). Such adaptive behavior that is not captured by this model, and is unlikely to be simply characterized and explained by data we have available (e.g. EdU).

However, the purpose of the model is to firstly confirm that the *Notch1*^{-/-} clones are substantially fitter than the *Notch1*^{+/-} clones, and secondly, to help explain how the tissue becomes occupied by *Notch1* null clones because of the haploinsufficiency of *Notch1* and the strong additional advantage of a mutation to the second *Notch1* allele. We are therefore predominantly interested in the inferred fitness parameter from the model. The inclusion of the induction rate in the model is to account for the experimental design (which due to the YFP allows us to still measure the sizes of clones after some clonal collision has occurred), but we are less interested in the inferred values.

The deficiency of the model we used is possibly overestimating *Notch1*^{+/-} fitness at later time points/larger clone sizes. However, this has no impact on the overall conclusions of the modelling.

We have added some text in the Supplementary note addressing this. **The additional text is in bold**, the rest of the paragraph is included for clarity.

Pages 63/64:

“Although mutant fitness is the key parameter we are interested in, the inferred induction proportion can also provide some useful information. It can suggest where the simple model we are using might be less able to replicate the experimental results, and provide reassurance that the conclusions we are drawing from the inferred fitness values are robust. In the cases of neutral clones (*Notch1*^{+/+} and wild type intensity clones from induced mutant mice) the induction proportion was not constrained by the fitting (Extended data Fig. 4b, Table 2), because these clones have the same fitness as the surrounding wild type cells and therefore do not impact the growth of any other clones. The inferred induction proportion for the *Notch1*^{+/-} clones was larger than for the *Notch1*^{-/-} clones (Extended data Fig. 4b, Table 2). In simulations of the best fitting *Notch1*^{+/-} parameters, the mutant clones were colliding at the later time points, reducing the clone sizes (Video 1). This high density of clones and extensive clonal collision was not occurring to such an extent in the experimental data (Extended data Fig. 3 j,o), suggesting that the model is not fully capturing all details of the *Notch1*^{+/-} clonal dynamics and may be overestimating the fitness of large *Notch1*^{+/-} clones. However, the conclusions of the inferred fitness comparison are still valid: that the *Notch1*^{+/-} clones have a clear growth advantage over wild type cells, and that the *Notch1*^{-/-} clones have much larger growth advantage than the *Notch1*^{+/-} clones.”

3. It would be useful to relate the values of fitness advantages you get in the model to the experimentally measured changes in EdU positive cells. Do these match the observed values.

The model does not explicitly model each cell division so we cannot directly compare the EdU data to the simulations. However, as discussed above (in the answer to minor issue 1), the data is qualitatively consistent with the fitness differences inferred from the modelling. A fitness difference in the model creates larger cell fate biases, consistent with the larger difference in suprabasal/basal EdU ratio seen in the wild type cells adjacent to the *Notch1*^{-/-} clones.

We have added the following paragraph to the supplementary note:

Page 63:

“The fitting results are consistent with the EdU 48h data shown in Fig. 2k,l. In the model, interaction between highly fit mutant cells and adjacent wild type cells biases the wild type cells towards differentiation and the mutant cells towards division. The higher the fitness imbalance, the stronger the fate bias. The stronger differentiation bias in the wild type cells adjacent to *Notch1*^{-/-} clones (Fig.2l) is therefore consistent with the higher fitness inferred for the *Notch1*^{-/-} clones.”

4. In page 3, the sentence ‘94% (15/16) of negative staining areas in the two oldest donors carried nonsense or essential splice...’. Shouldn’t it be ‘missense’ instead of ‘nonsense’?

Nonsense mutations lead to the appearance of stop codons (truncating mutation) which is consistent with the lack of detection of a NOTCH1 protein. Missense mutations with a substitution by a different amino acid, may still lead to the expression and detection of the protein. The quoted sentence matches our data and is also consistent with the hypothetical effect of these mutations on protein expression.

5. In the second to last paragraph in the discussion you should replace 'pleiotropic effects' with 'pleiotropic effects'

We thank the Reviewer for picking up this mistake. We have corrected this point.

References

- Alcolea MP, Greulich P, Wabik A, Frede J, Simons BD, Jones PH. 2014. Differentiation imbalance in single oesophageal progenitor cells causes clonal immortalization and field change. *Nat Cell Biol* 16: 615-22
- Bray SJ, Gomez-Lamarca M. 2018. Notch after cleavage. *Current opinion in cell biology* 51: 103-09
- Celli A, Mackenzie DS, Zhai Y, Tu C-L, Bikle DD, et al. 2012. SERCA2-Controlled Ca²⁺-Dependent Keratinocyte Adhesion and Differentiation Is Mediated via the Sphingolipid Pathway: A Therapeutic Target for Darier's Disease. *Journal of Investigative Dermatology* 132: 1188-95
- Clayton E, Doupe DP, Klein AM, Winton DJ, Simons BD, Jones PH. 2007. A single type of progenitor cell maintains normal epidermis. *Nature* 446: 185-89
- Colom B, Alcolea MP, Piedrafita G, Hall MW, Wabik A, et al. 2020. Spatial competition shapes the dynamic mutational landscape of normal esophageal epithelium. *Nature Genetics* 52: 604-14
- Colom B, Herms A, Hall MWJ, Dentre SC, King C, et al. 2021. Mutant clones in normal epithelium outcompete and eliminate emerging tumours. *Nature* 598: 510-14
- Consortium T. 2017. Integrated genomic characterization of oesophageal carcinoma. *Nature* 541: 169-75
- Dann E, Henderson NC, Teichmann SA, Morgan MD, Marioni JC. 2022. Differential abundance testing on single-cell data using k-nearest neighbor graphs. *Nat Biotechnol* 40: 245-53
- Doupe DP, Alcolea MP, Roshan A, Zhang G, Klein AM, et al. 2012. A single progenitor population switches behavior to maintain and repair esophageal epithelium. *Science* 337: 1091-3
- Fernandez-Antoran D, Piedrafita G, Murai K, Ong SH, Herms A, et al. 2019. Outcompeting p53-Mutant Cells in the Normal Esophagus by Redox Manipulation. *Cell Stem Cell* 25: 329-41
- Frede J, Greulich P, Nagy T, Simons BD, Jones PH. 2016. A single dividing cell population with imbalanced fate drives oesophageal tumour growth. *Nat Cell Biol* 18: 967-78

- Gilpin AR. 1993. Table for Conversion of Kendall'S Tau to Spearman'S Rho Within the Context of Measures of Magnitude of Effect for Meta-Analysis. *Educational and Psychological Measurement* 53: 87-92
- Herms A, Colom B, Piedrafita G, Murai K, Ong SH, et al. 2021. Levelling out differences in aerobic glycolysis neutralizes the competitive advantage of oncogenic *PIK3CA* mutant progenitors in the esophagus. *bioRxiv*: 2021.05.28.446104
- Hong JH, Yang Y-M, Kim HS, Lee S-I, Muallem S, Shin DM. 2010. Markers of squamous cell carcinoma in sarco/endoplasmic reticulum Ca²⁺ ATPase 2 heterozygote mice keratinocytes. *Progress in Biophysics and Molecular Biology* 103: 81-87
- La Manno G, Soldatov R, Zeisel A, Braun E, Hochgerner H, et al. 2018. RNA velocity of single cells. *Nature* 560: 494-98
- Liu LH, Boivin GP, Prasad V, Periasamy M, Shull GE. 2001. Squamous Cell Tumors in Mice Heterozygous for a Null Allele of *Atp2a2*, Encoding the Sarco(endo)plasmic Reticulum Ca²⁺-ATPase Isoform 2 Ca²⁺Pump. *Journal of Biological Chemistry* 276: 26737-40
- Lowell S, Jones P, Le Roux I, Dunne J, Watt FM. 2000. Stimulation of human epidermal differentiation by delta-notch signalling at the boundaries of stem-cell clusters. *Curr Biol* 10: 491-500
- Lubin DJ, Mick R, Shroff SG, Stashek K, Furth EE. 2018. The notch pathway is activated in neoplastic progression in esophageal squamous cell carcinoma. *Human Pathology* 72: 66-70
- Martincorena I, Fowler JC, Wabik A, Lawson ARJ, Abascal F, et al. 2018. Somatic mutant clones colonize the human esophagus with age. *Science* 362: 911-17
- Martincorena I, Raine KM, Gerstung M, Dawson KJ, Haase K, et al. 2017. Universal Patterns of Selection in Cancer and Somatic Tissues. *Cell* 171: 1029-41.e21
- Martincorena I, Roshan A, Gerstung M, Ellis P, Van Loo P, et al. 2015. Tumor evolution. High burden and pervasive positive selection of somatic mutations in normal human skin. *Science* 348: 880-6
- McGinn J, Hallou A, Han S, Krizic K, Ulyanchenko S, et al. 2021. A biomechanical switch regulates the transition towards homeostasis in oesophageal epithelium. *Nat Cell Biol* 23: 511-25
- Mesa KR, Kawaguchi K, Cockburn K, Gonzalez D, Boucher J, et al. 2018. Homeostatic Epidermal Stem Cell Self-Renewal Is Driven by Local Differentiation. *Cell Stem Cell* 23: 677-86.e4
- Movahedan A, Afsharkhamseh N, Sagha HM, Shah JR, Milani BY, et al. 2013. Loss of Notch1 disrupts the barrier repair in the corneal epithelium. *PLoS One* 8: e69113
- Murai K, Skrupskelyte G, Piedrafita G, Hall M, Kostiou V, et al. 2018. Epidermal Tissue Adapts to Restrain Progenitors Carrying Clonal p53 Mutations. *Cell Stem Cell* 23: 687-99.e8

- Natsuizaka M, Whelan KA, Kagawa S, Tanaka K, Giroux V, et al. 2017. Interplay between Notch1 and Notch3 promotes EMT and tumor initiation in squamous cell carcinoma. *Nature Communications* 8: 1758
- Piedrafita G, Kostiou V, Wabik A, Colom B, Fernandez-Antoran D, et al. 2020. A single progenitor model as the unifying paradigm of epidermal and esophageal epithelial maintenance in mice. *Nature Communications* 11: 1429
- Prasad V, Boivin GP, Miller ML, Liu LH, Erwin CR, et al. 2005. Haploinsufficiency of *Atp2a2*, Encoding the Sarco(endo)plasmic Reticulum Ca²⁺-ATPase Isoform 2 Ca²⁺ Pump, Predisposes Mice to Squamous Cell Tumors via a Novel Mode of Cancer Susceptibility. *Cancer Research* 65: 8655-61
- Sawangarun W, Mandasari M, Aida J, Morita K-i, Kayamori K, et al. 2018. Loss of Notch1 predisposes oro-esophageal epithelium to tumorigenesis. *Experimental Cell Research* 372: 129-40
- Trapnell C, Cacchiarelli D, Grimsby J, Pokharel P, Li S, et al. 2014. The dynamics and regulators of cell fate decisions are revealed by pseudotemporal ordering of single cells. *Nature Biotechnology* 32: 381-86
- Yokoyama A, Kakiuchi N, Yoshizato T, Nannya Y, Suzuki H, et al. 2019. Age-related remodelling of oesophageal epithelia by mutated cancer drivers. *Nature* 565: 312-17

Decision Letter, first revision:

17th Aug 2022

Dear Professor Jones,

Your Article, "Notch1 mutation drives clonal expansion in normal esophageal epithelium but impairs tumor growth" has now been seen by 4 referees (as before, Reviewers #2 and #3 reviewed together and uploaded the same report). You will see from their comments below that while they find your work of interest, some points have been raised. We remain interested in the possibility of publishing your study in Nature Genetics, but would like to consider your response to these concerns in the form of a revised manuscript before we make a final decision on publication.

Our hope at this point would be to assess your revisions in-house once you resubmit. However, we might have to return to one or more reviewers depending on your response. Please be assured that we'll only do this if absolutely necessary, we are keen to avoid incurring delays.

We therefore invite you to revise your manuscript taking into account all reviewer and editor comments. Please highlight all changes in the manuscript text file. At this stage we will need you to upload a copy of the manuscript in MS Word .docx or similar editable format.

We are committed to providing a fair and constructive peer-review process. Do not hesitate to contact

us if there are specific requests from the reviewers that you believe are technically impossible or unlikely to yield a meaningful outcome.

*2) If you have not done so already please begin to revise your manuscript so that it conforms to our Article format instructions, available [here](http://www.nature.com/ng/authors/article_types/index.html). Refer also to any guidelines provided in this letter.

[redacted]

We hope to receive your revised manuscript within four to eight weeks. If you cannot send it within this time, please let us know.

Sincerely,

Safia Danovi
Editor
Nature Genetics

Reviewers' Comments:

Reviewer #1:

Remarks to the Author:

The revised version of the manuscript by Emilie Abby, Phil Jones and co-authors addresses most comments and constitutes a substantial improvement over the previous version.

The authors now better clarify one of the key experiments, referred to as 'the 8 hour EdU experiment' that leads them to conclude that neither Notch1+/- nor Notch1-/- have increased fitness compared to wild type. Rather, they induce the differentiation and stratification of adjacent wild type cells. This mechanism sounds similar to that recently described in the case of APC mutant cells by van Nerveen et al Nature 2021 and this reference should be cited.

If this is the mechanism by which mutant Notch1 clones colonise normal epithelium, it is unclear whether the 3D in vitro experiment shown in Reviewer Fig.2 allows to reproduce the induced differentiation and stratification of wild type cells?

Another point that in my opinion remains not sufficiently explained in the text is the apparent paradox of ~10% of TCGA ESCC with mutations in Notch1. In these cases, the authors hypothesise that Notch1 mutations pre-existed tumour formation and that the tumour could form because of other driver alterations co-occurring in the same cells. At this stage this however remains a hypothesis. As a consequence, the paragraph added at p.41 results cryptic and misleading.

"That over 90% of human ESCC retain one or more wild type copies of NOTCH1 while arising from an epithelium where the majority of cells have biallelic NOTCH1 disruption argues wild type NOTCH1 does promote cancer development"

This is, at best, an indirect interpretation and may result misleading in this context. Similarly, the following sentence is unclear:

"The complexity of genomic alterations in human ESCC may explain the ability of the minority of tumors to bypass the benefit of retaining wild type NOTCH1(Consortium 2017)."

Most cancer types have complex landscape of genomic alterations and ESCC is no exception. Moreover, it usually shows an average of 5 drivers/sample, one of which can be indeed Notch1.

This paragraph should be rewritten to describe the data currently available in human cancer in a more objective way.

Please note that Reviewers #2 and #3 reviewed the paper together and have uploaded the same report

Reviewer #2:

Remarks to the Author:

Comments for Author

Thank you for your thorough response to the concerns raised during the first review of the manuscript. Many of the changes have yielded considerable clarity in the revised manuscript that highlight the mechanistic insight by which Notch1 deficient clones spread through the epithelium and inhibit tumorigenesis. This finding is likely to have a major impact on how we interpret cell competition, clonal fitness, and tumorigenesis in other cancers as well. A few minor textual / figure layout concerns remain as well as the need to temper a few statements in the discussion section.

Page 2 – change “migrates to the nucleus” to “translocates to the nucleus”

Figure 1g – consider color blind guidelines for color figure presentation. ITGA6 is also difficult to interpret because magenta/green colocalization would also present as “white / greyscale”.

Page 8 – Sentence reading, “To follow the behavior of Notch1 mutant clones.....with a genetic labeling sentence” Please revise the sentence for grammar. As written the sentence is unclear.

Figure 2 ghkl – The y-axis scales are set too high – especially gh, which makes the data points difficult to compare across groups.

Figure 3e – Please reannotate the image to include dotted outlines on both images to help guide the reader. Also, if DAPI and GFP are not included on the channel on the right please annotate accordingly.

Page 24 – Callout missing in the “supplementary note” bottom right of page.

Figure 4fgh, The three main cell types “cycling basal, resting basal, and differentiating cells” are difficult to view in panel g, which is labelled according to cluster identity (i.e. cluster number). Consider adding a second UMAP with three main colors for the three major cell types with color code as per “f” y-axis. Similar colors for h would also be helpful. Typo also “diffrentiating”

Figure 6e – type y-axis “pojected”

Discussion – I found added discussion sections reasonable but in need of tempering.

For instance, “Once an area has been colonized.....mutant progenitors revert towards that of wild type progenitors”. I agree that the epithelium is histologically normal. However, the question is whether or not the epithelium is actually normally behaved. While homeostasis is normal, could it also not be the case that all cells turnover less or all cells are turning over more compared to WT. As long as rates of turnover / renewal are equivalent then clonal neutrality is achieved.

Again, "Notch1 disruption argues wild type NOTCH1 does promote cancer development." The current data do not definitively demonstrate whether NOTCH1 promotes cancer development or if instead NOTCH1 WT is permissive to cancer development.

Reviewer #3:

Remarks to the Author:

Thank you for your thorough response to the concerns raised during the first review of the manuscript. Many of the changes have yielded considerable clarity in the revised manuscript that highlight the mechanistic insight by which Notch1 deficient clones spread through the epithelium and inhibit tumorigenesis. This finding is likely to have a major impact on how we interpret cell competition, clonal fitness, and tumorigenesis in other cancers as well. A few minor textual / figure layout concerns remain as well as the need to temper a few statements in the discussion section.

Page 2 – change "migrates to the nucleus" to "translocates to the nucleus"

Figure 1g – consider color blind guidelines for color figure presentation. ITGA6 is also difficult to interpret because magenta/green colocalization would also present as "white / greyscale".

Page 8 – Sentence reading, "To follow the behavior of Notch1 mutant clones.....with a genetic labeling sentence" Please revise the sentence for grammar. As written the sentence is unclear.

Figure 2 ghkl – The y-axis scales are set too high – especially gh, which makes the data points difficult to compare across groups.

Figure 3e – Please reannotate the image to include dotted outlines on both images to help guide the reader. Also, if DAPI and GFP are not included on the channel on the right please annotate accordingly.

Page 24 – Callout missing in the "supplementary note" bottom right of page.

Figure 4fgh, The three main cell types "cycling basal, resting basal, and differentiating cells" are difficult to view in panel g, which is labelled according to cluster identity (i.e. cluster number). Consider adding a second UMAP with three main colors for the three major cell types with color code as per "f" y-axis. Similar colors for h would also be helpful. Typo also "diffrentiating"

Figure 6e – type y-axis "pojected"

Discussion – I found added discussion sections reasonable but in need of tempering.

For instance, "Once an area has been colonized.....mutant progenitors revert towards that of wild type progenitors". I agree that the epithelium is histologically normal. However, the question is whether or not the epithelium is actually normally behaved. While homeostasis is normal, could it also not be the case that all cells turnover less or all cells are turning over more compared to WT. As long as rates of turnover / renewal are equivalent then clonal neutrality is achieved.

Again, "Notch1 disruption argues wild type NOTCH1 does promote cancer development." The current data do not definitively demonstrate whether NOTCH1 promotes cancer development or if instead NOTCH1 WT is permissive to cancer development.

Reviewer #4:

Remarks to the Author:

In the revised version, the authors addressed my comments adequately. The new explanation (and figure) on the EDU assay has considerably clarified the results and conclusions. I also found the response to the other reviewers thorough and convincing. Overall, I think the results presented are important and are suitable for publication in Nature Genetics.

Author Rebuttal, first revision:

We are again most grateful to the reviewers for their careful reading of the revised manuscript and their positive suggestions which have further improved the text. We respond to these below, our comments are in blue.

Reviewer #1:

Remarks to the Author:

The revised version of the manuscript by Emilie Abby, Phil Jones and co-authors addresses most comments and constitutes a substantial improvement over the previous version.

The authors now better clarify one of the key experiments, referred to as 'the 48 hour EdU experiment' that leads them to conclude that neither Notch1^{+/-} nor Notch1^{-/-} have increased fitness compared to wild type. Rather, they induce the differentiation and stratification of adjacent wild type cells. This mechanism sounds similar to that recently described in the case of APC mutant cells by van Nerveen et al Nature 2021 and this reference should be cited.

The work by van Nerveen relates to *Apc* mutant clones competing in the intestinal crypt and is indeed a very nice parallel of a mutant clone driving wild type cell differentiation (van Neerven et al., 2021). We now cite this and compare this to *Notch1*^{-/-} clones as helpfully suggested by the reviewer as follows:

Page 40

'By driving wild type cell differentiation, *Notch1* null cells at the clone margins can divide within the homeostatic constraint of cell division being linked to differentiation, resulting in extensive colonization of the epithelium as wild type and heterozygous cells are displaced (**Fig.**

2o). This mechanism would explain the selection of the clones with biallelic *NOTCH1*

disruption that come to dominate normal human esophagus. Such 'supercompetition' has also been seen in the intestine where *Apc* mutant intestinal stem cells drive the differentiation of their wild type neighbors to colonize the intestinal crypt (van Neerven *et al.*, 2021).

If this is the mechanism by which mutant Notch1 clones colonise normal epithelium, it is unclear whether the 3D in vitro experiment shown in Reviewer Fig.2 allows to reproduce the induced differentiation and stratification of wild type cells?

The organotypic cultures shown in Reviewer Figure 2 recapitulate the 3D organization of the epithelium with a proliferating basal layer overlaid by non-dividing, differentiating supra-basal cells (Fernandez-Antoran *et al.*, 2019; Herms *et al.*, 2021). They are thus capable of replicating the differentiation and stratification of wild type cells induced at the margin of *Notch1*^{-/-} clones *in vivo*.

Another point that in my opinion remains not sufficiently explained in the text is the apparent paradox of ~10% of TCGA ESCC with mutations in Notch1. In these cases, the authors hypothesise that Notch1 mutations pre-existed tumour formation and that the tumour could form because of other driver alterations co-occurring in the same cells. At this stage this however remains a hypothesis. As a consequence, the paragraph added at p.41 results cryptic and misleading.

"That over 90% of human ESCC retain one or more wild type copies of NOTCH1 while arising from an epithelium where the majority of cells have biallelic NOTCH1 disruption argues wild type NOTCH1 does promote cancer development"

This is, at best, an indirect interpretation and may result misleading in this context. Similarly, the following sentence is unclear:

"The complexity of genomic alterations in human ESCC may explain the ability of the minority of tumors to bypass the benefit of retaining wild type NOTCH1(Consortium 2017)."

Most cancer types have complex landscape of genomic alterations and ESCC is no exception. Moreover, it usually shows an average of 5 drivers/sample, one of which can be indeed Notch1.

This paragraph should be rewritten to describe the data currently available in human cancer in a more objective way.

We agree that while our data show NOTCH1 signaling sustains tumor growth, we cannot exclude the possibility of *NOTCH1* mutants having a positive role within the small sub ESCC that carry them. However, neither can we know if *NOTCH1* is a 'driver' in these tumors, as mutant gene appear positively selected in a tumor by virtue of being positively selected in normal tissue and being merely a 'passenger' in the

transformed clone. In the revised version of the text, we have sought to clarify that the functional significance of the *NOTCH1* mutants in 8% of ESCC is unknown, as follows:

Page 41:

'Might these findings be of relevance in humans? Over 90% of human ESCC retain one or more wild type copies of *NOTCH1* while arising from an epithelium where most cells have biallelic *NOTCH1* disruption. This argues that retention of wild type *NOTCH1* favors the development of ESCC. What of the remaining ESCC cases that do have biallelic *NOTCH1* disruption (Consortium, 2017)? One possibility is that *NOTCH1* loss, in association with multiple other genomic alterations, promotes transformation in this subset of tumors. However, it is also plausible that the *NOTCH1* alterations in these cases are 'passengers', carried over from normal tissue with the requirement for wildtype *NOTCH1* in carcinogenesis bypassed by the other genome changes in the cancer clone.

We would be happy to revise this text if required, but we feel it reflects the uncertainty of the position, embodying the reviewer's corrective comments on our previous version.

Please note that Reviewers #2 and #3 reviewed the paper together and have uploaded the same report
Reviewer #2:

Remarks to the Author:

Comments for Author

Thank you for your thorough response to the concerns raised during the first review of the manuscript. Many of the changes have yielded considerable clarity in the revised manuscript that highlight the mechanistic insight by which Notch1 deficient clones spread through the epithelium and inhibit tumorigenesis. This finding is likely to have a major impact on how we interpret cell competition, clonal fitness, and tumorigenesis in other cancers as well. A few minor textual / figure layout concerns remain as well as the need to temper a few statements in the discussion section.

Page 2 – change “migrates to the nucleus” to “translocates to the nucleus” This has been changed.

Figure 1g – consider color blind guidelines for color figure presentation. ITGA6 is also difficult to interpret because magenta/green colocalization would also present as “white / greyscale”.

In order to improve the interpretation of Figure 1 for all, including color-blind readers, we have modified Figure 1g along with Figure 1a, that presented with the same color combination. In Fig. 1a, ITGA6 has been removed to avoid confusion. We have instead used a white dotted to delineate the basement membrane separating the epithelium from the submucosa.

Fig.1g has now been clarified by removing KRT4 staining and by replacing the original figure with newly acquired images from the same area but a few sections deeper into the block which have an improved signal/background ratio.

We have made the relevant changes in the figure legends.

Page 8 – Sentence reading, “To follow the behavior of Notch1 mutant clones.....with a genetic labeling sentence” Please revise the sentence for grammar. As written the sentence is unclear. We have amended this sentence to

‘To follow the behavior of *Notch1* mutant clones we generated *AhCre^{ERT} Rosa26^{flox}YFP Notch1^{flox}* triple transgenic (*YFP^{Cre}Notch1*) mice. These animals carry a conditional *Notch1* allele and a genetic labeling system.’

Figure 2 ghkl – The y-axis scales are set too high – especially gh, which makes the data points difficult to compare across groups.

This has been corrected, with y-axis in plots g and h now set to 0.5.

Figure 3e – Please reannotate the image to include dotted outlines on both images to help guide the reader. Also, if DAPI and GFP are not included on the channel on the right please annotate accordingly.

Thank you for pointing out the lack of clarity in Fig3e annotations. We have now made the suggested changes.

Page 24 – Callout missing in the “supplementary note” bottom right of page. We have added Fig. 4e here.

Figure 4fgh, The three main cell types “cycling basal, resting basal, and differentiating cells” are difficult to view in panel g, which is labelled according to cluster identity (i.e. cluster number). Consider adding a second UMAP with three main colors for the three major cell types with color code as per “f” y-axis. Similar colors for h would also be helpful.

We have included a new UMAP plot as panel h as suggested in order to display the three major cell types. Panels f to i were amended accordingly using consistent color codes for the three main cell types in panel i (previous panel h) as recommended. We have made the relevant changes in the figure legends.

Typo also “differentiating” This has been corrected.

Figure 6e – type y-axis “pojected” This has been corrected.

Discussion – I found added discussion sections reasonable but in need of tempering.

For instance, “Once an area has been colonized.....mutant progenitors revert towards that of wild type progenitors”. I agree that the epithelium is histologically normal. However, the question is whether or not the epithelium is actually normally behaved. While homeostasis is normal, could it also not be the case that all cells turnover less or all cells are turning over more compared to WT. As long as rates of turnover / renewal are equivalent then clonal neutrality is achieved.

We agree that we need to be moderate in our statements when dealing with the reversion of mutant cell phenotype. The multiple assays we performed *in vivo* did not reveal any significant phenotypic difference between colonized mutant and wild type epithelium. Nevertheless, we still sought to be conservative by writing ‘revert towards’ rather than ‘returned to’ normal, seeking to imply that full normality may not be attained.

On the matter of cell turnover, we specifically addressed mutant tissue turnover by measuring basal cell density, % of proliferating cells (using EdU or BrdU, Ki67, and pH3 proliferation markers), rate of stratification in two different 48-hour EdU assays and tissue thickness (Fig.5 and Extended data Fig.6q-w). None of these parameters differed significantly between wild type and *Notch1* ^{-/-} epithelium, arguing against any substantive difference in cell turnover rate. We conclude that the requirement for clonal neutrality suggested by the reviewer is thus achieved.

In addition, there were no significant differences in single cell RNAseq analysis and no qualitative differences in histology or immunostaining for differentiation markers (Fig.4 and Fig.5). We would certainly not argue against the existence of any phenotypic differences, indeed as we state in the discussion, we observed transcriptional changes in bulk RNAseq. It is the case however that no phenotypic differences were detectable with the range of assays performed in the paper.

For now, we have stuck with ‘revert towards’ but are open to alternative suggestions of how to phrase this to both be appropriately cautious but reflect the fact that multiple assays show no significant difference. We have simplified the text to read (page 40), no longer referring specifically to progenitors but rather more generally to cells and commenting on the histology of the epithelium remaining normal. This is a robust aspect of the phenotype on which we can comment with confidence.

‘Once an area has been colonized by biallelic *Notch1* mutants, the phenotype of mutant cells reverts towards that of wild type cells so that the mutant epithelium remains histologically normal. *Notch1* does impact the transcription of some genes involved in DNA replication, but scRNAseq, cell tracking and histological analyses indicate that cell proliferation and differentiation and epithelial structure and histology are not significantly perturbed. Such reversion towards a near normal cell state

explains the normal appearance of aged human esophageal epithelium despite NOTCH1 signaling being disrupted in the majority of the tissue.'

Again, "Notch1 disruption argues wild type NOTCH1 does promote cancer development." The current data do not definitively demonstrate whether NOTCH1 promotes cancer development or if instead NOTCH1 WT is permissive to cancer development.

We agree that 'promote' is ambiguous in the context of the mouse cancer literature. However, the data on Notch1 being required for optimal tumor growth seem to us to make 'permissive' too weak. We have opted for 'favors' as term that avoids ambiguity but would be happy to review this if required. The text, in the discussion, now reads

Page 41

'Might these findings be of relevance in humans? Over 90% of human ESCC retain one or more wild type copies of *NOTCH1* while arising from an epithelium where most cells have biallelic *NOTCH1* disruption. This argues that retention of wild type *NOTCH1* favors the development of ESCC.'

As this same expression was used in the Abstract, we have amended the abstract accordingly:

Page 2:

'Notch1 null tumors showed reduced proliferation. We conclude that Notch1 mutations in normal epithelium are beneficial as wild type Notch1 favors tumor expansion. NOTCH1 blockade may have therapeutic potential in preventing esophageal squamous cancer.'

Reviewer #3:

Remarks to the Author:

Thank you for your thorough response to the concerns raised during the first review of the manuscript. Many of the changes have yielded considerable clarity in the revised manuscript that highlight the mechanistic insight by which Notch1 deficient clones spread through the epithelium and inhibit tumorigenesis. This finding is likely to have a major impact on how we interpret cell competition, clonal fitness, and tumorigenesis in other cancers as well. A few minor textual / figure layout concerns remain as well as the need to temper a few statements in the discussion section.

Page 2 – change "migrates to the nucleus" to "translocates to the nucleus"

Figure 1g – consider color blind guidelines for color figure presentation. ITGA6 is also difficult to interpret because magenta/green colocalization would also present as “white / greyscale”.

Page 8 – Sentence reading, “To follow the behavior of Notch1 mutant clones.....with a genetic labeling sentence” Please revise the sentence for grammar. As written the sentence is unclear.

Figure 2 ghkl – The y-axis scales are set too high – especially gh, which makes the data points difficult to compare across groups.

Figure 3e – Please reannotate the image to include dotted outlines on both images to help guide the reader. Also, if DAPI and GFP are not included on the channel on the right please annotate accordingly.

Page 24 – Callout missing in the “supplementary note” bottom right of page.

Figure 4fgh, The three main cell types “cycling basal, resting basal, and differentiating cells” are difficult to view in panel g, which is labelled according to cluster identity (i.e. cluster number). Consider adding a second UMAP with three main colors for the three major cell types with color code as per “f” y-axis. Similar colors for h would also be helpful. Typo also “differentiating”

Figure 6e – type y-axis “projected”

Discussion – I found added discussion sections reasonable but in need of tempering.

For instance, “Once an area has been colonized.....mutant progenitors revert towards that of wild type progenitors”. I agree that the epithelium is histologically normal. However, the question is whether or not the epithelium is actually normally behaved. While homeostasis is normal, could it also not be the case that all cells turnover less or all cells are turning over more compared to WT. As long as rates of turnover / renewal are equivalent then clonal neutrality is achieved.

Again, “Notch1 disruption argues wild type NOTCH1 does promote cancer development.” The current data do not definitively demonstrate whether NOTCH1 promotes cancer development or if instead NOTCH1 WT is permissive to cancer development.

Reviewer #4:

Remarks to the Author:

In the revised version, the authors addressed my comments adequately. The new explanation (and figure) on the EDU assay has considerably clarified the results and conclusions. I also found the response to the other reviewers thorough and convincing. Overall, I think the results presented are important and are suitable for publication in Nature Genetics.

We are most grateful to the reviewer for their positive comments and help with improving the manuscript.

Response references

- Consortium, T. (2017). Integrated genomic characterization of oesophageal carcinoma. *Nature* 541, 169-175. 10.1038/nature20805.
- Fernandez-Antoran, D., Piedrafita, G., Murai, K., Ong, S.H., Herms, A., Frezza, C., and Jones, P.H. (2019). Outcompeting p53-Mutant Cells in the Normal Esophagus by Redox Manipulation. *Cell stem cell* 25, 329-341. 10.1016/j.stem.2019.06.011.
- Herms, A., Colom, B., Piedrafita, G., Murai, K., Ong, S.H., Fernandez-Antoran, D., Bryant, C., Frezza, C., Vanhaesebroeck, B., and Jones, P.H. (2021). Levelling out differences in aerobic glycolysis neutralizes the competitive advantage of oncogenic *PIK3CA* mutant progenitors in the esophagus. *bioRxiv*, 2021.2005.2028.446104. 10.1101/2021.05.28.446104.
- Murai, K., Skrupskelyte, G., Piedrafita, G., Hall, M., Kostiou, V., Ong, S.H., Nagy, T., Cagan, A., Goulding, D., Klein, A.M., et al. (2018). Epidermal Tissue Adapts to Restrain Progenitors Carrying Clonal p53 Mutations. *Cell stem cell* 23, 687-699.e688. 10.1016/j.stem.2018.08.017.
- van Neerven, S.M., de Groot, N.E., Nijman, L.E., Scicluna, B.P., van Driel, M.S., Lecca, M.C., Warmerdam, D.O., Kakkar, V., Moreno, L.F., Vieira Braga, F.A., et al. (2021). Apc-mutant cells act as supercompetitors in intestinal tumour initiation. *Nature* 594, 436-441. 10.1038/s41586-021-03558-4.

Decision Letter, second revision:
--

26th Sep 2022

Dear Dr. Jones,

I hope you're well.

Thank you for submitting your revised manuscript "Notch1 mutation drives clonal expansion in normal esophageal epithelium but impairs tumor growth" (NG-A57207R1). We assessed your revisions in-house and I'm delighted to say that we'll be happy in principle to publish it in *Nature Genetics*, pending minor revisions to satisfy the referees' final requests and to comply with our editorial and formatting guidelines.

Sincerely,

Safia

Safia Danovi
Editor
Nature Genetics

Final Decision Letter:

7th Dec 2022

Dear Dr. Jones,

I am delighted to say that your manuscript "Notch1 mutations drive clonal expansion in normal esophageal epithelium but impair tumor growth" has been accepted for publication in an upcoming issue of Nature Genetics.

Your paper will be published online after we receive your corrections and will appear in print in the next available issue. You can find out your date of online publication by contacting the Nature Press Office (press@nature.com) after sending your e-proof corrections. Now is the time to inform your Public Relations or Press Office about your paper, as they might be interested in promoting its publication. This will allow them time to prepare an accurate and satisfactory press release. Include your manuscript tracking number (NG-A57207R2) and the name of the journal, which they will need when they contact our Press Office.

Before your paper is published online, we shall be distributing a press release to news organizations worldwide, which may very well include details of your work. We are happy for your institution or

funding agency to prepare its own press release, but it must mention the embargo date and Nature Genetics. Our Press Office may contact you closer to the time of publication, but if you or your Press Office have any enquiries in the meantime, please contact press@nature.com.

Please note that *Nature Genetics* is a Transformative Journal (TJ). Authors may publish their research with us through the traditional subscription access route or make their paper immediately open access through payment of an article-processing charge (APC). Authors will not be required to make a final decision about access to their article until it has been accepted. [Find out more about Transformative Journals](https://www.springernature.com/gp/open-research/transformative-journals)

Authors may need to take specific actions to achieve [compliance](https://www.springernature.com/gp/open-research/funding/policy-compliance-faqs) with funder and institutional open access mandates. If your research is supported by a funder that requires immediate open access (e.g. according to [Plan S principles](https://www.springernature.com/gp/open-research/plan-s-compliance)) then you should select the gold OA route, and we will direct you to the compliant route where possible. For authors selecting the subscription publication route, the journal's standard licensing terms will need to be accepted, including [self-archiving-and-license-to-publish](https://www.nature.com/nature-portfolio/editorial-policies/self-archiving-and-license-to-publish). Those licensing terms will supersede any other terms that the author or any third party may assert apply to any version of the manuscript.

Please note that Nature Portfolio offers an immediate open access option only for papers that were first submitted after 1 January, 2021.

If you have not already done so, we invite you to upload the step-by-step protocols used in this manuscript to the Protocols Exchange, part of our on-line web resource, natureprotocols.com. If you complete the upload by the time you receive your manuscript proofs, we can insert links in your article that lead directly to the protocol details. Your protocol will be made freely available upon publication of your paper. By participating in natureprotocols.com, you are enabling researchers to more readily reproduce or adapt the methodology you use. [Natureprotocols.com](http://natureprotocols.com) is fully searchable, providing your protocols and paper with increased utility and visibility. Please submit your protocol to <https://protocolexchange.researchsquare.com/>. After entering your nature.com username and password you will need to enter your manuscript number (NG-A57207R2). Further information can be found at <https://www.nature.com/nature-portfolio/editorial-policies/reporting-standards#protocols>

Sincerely,

Safia Danovi
Editor
Nature Genetics

Click here if you would like to recommend Nature Genetics to your librarian
<http://www.nature.com/subscriptions/recommend.html#forms>